# The Bridge-Garden Dilemma in LLM Distillation:
# Why Mixing Hard and Soft Labels Works

Guanghui Wang [1]   Kaiwen Lv Kacuila [2]   Zhiyong Yang [1]   Zitai Wang [3]   Jin-Wen Wu [2]   Longtao Huang [2]
Qianqian Xu [3 4]   Qingming Huang [1 5 3]

## Abstract

Knowledge distillation (KD) transfers knowledge from a large teacher model to a smaller student. In language modeling, the student is trained either on tokens sampled from the teacher (**hard labels**) or the teacher's full next-token distribution (**soft labels**). Despite soft labels appear strictly richer, we find that mixing hard and soft labels consistently yields better results. Crucially, we show that this gain cannot be explained by closer teacher matching during training. Instead, it comes from reduced exposure bias—the mismatch between training and inference distributions. To explain this phenomenon, we introduce the Bridge–Garden Decomposition theory, which categorizes generation steps into two types: *Bridges*, where the next token must be *exact*, and *Gardens*, where it can be *flexible*. We show that hard-only KD excels in Bridges by avoiding risky deviations, while soft-only KD preserves diversity in Gardens. A hybrid strategy handles both cases and, as a result, reduces exposure bias across the sequence. Guided by this theory, we develop a family of Bridge–Garden hybrid supervision methods that adaptively balance hard and soft labels. Across a primary suite of seven teacher–student pairs (including Qwen, Llama, Gemma, and DeepSeek) and benchmarks in reasoning and coding, our approach outperforms divergence-based and on-policy KD baselines while reducing training cost by **9.7×**, en-

[1]School of Computer Science and Technology, University of Chinese Academy of Sciences, Beijing, China [2]Alibaba Group, Hangzhou, China [3]State Key Laboratory of AI Safety, Institute of Computing Technology, Chinese Academy of Sciences, Beijing, China [4]Beijing Academy of Artificial Intelligence, Beijing, China [5]Key Laboratory of Big Data Mining and Knowledge Management (BDKM), University of Chinese Academy of Sciences, Beijing, China. Correspondence to: Zhiyong Yang <yangzhiyong21@ucas.ac.cn>, Qingming Huang <qmhuang@ucas.ac.cn>.

*Proceedings of the $43^{rd}$ International Conference on Machine Learning*, Seoul, South Korea. PMLR 306, 2026. Copyright 2026 by the author(s).

abling efficient model compression. Code is available at https://github.com/ghwang-s/bridge_garden_hybrid_kd_release.

## 1. Introduction

Recent progress in large language models (LLMs, Achiam et al. 2023; Grattafiori et al. 2024; Guo et al. 2025; Yang et al. 2025) has largely been driven by scaling up model sizes (Hoffmann et al., 2022), yet this leads to high inference costs for deployment. Knowledge distillation (KD, Hinton et al. 2015) offers a practical way to alleviate this cost by transferring capabilities from a *powerful* teacher model to a *compact* student. The central question is how to design a distillation objective that makes the student match the teacher's generative behavior as closely as possible.

Conventional wisdom favors soft-label distillation (Wen et al., 2023; Gu et al., 2024; Agarwal et al., 2024; Xu et al., 2025; Wang et al., 2025b; Ko et al., 2025), where the student is trained to match the teacher's full predictive distribution over the next token. This approach is intuitively appealing because it captures richer information, including the teacher's confidence over alternative tokens. In contrast, hard-label distillation (Kim & Rush, 2016; Wang et al., 2023; Taori et al., 2023; Peng et al., 2023; Guo et al., 2025) relies on a single token sampled from the teacher's distribution as the training target, which loses much of the distributional information present in the soft labels.

Surprisingly, our empirical investigation reveals a consistent counter-intuitive trend: a simple linear combination of hard and soft losses systematically outperforms pure soft distillation (Fig.2a). This presents a puzzle: *If hard labels provide less information, why do they help?* At first glance, it seems that using hard labels can make optimization much easier, enabling the student to imitate the teacher more closely during training. However, we discover that the performance gain does not come from better training imitation. In fact, our experiments show that adding hard labels even worsens this training fit, as shown in Fig.2(b).

Instead, we show that the gain arises from a different source:

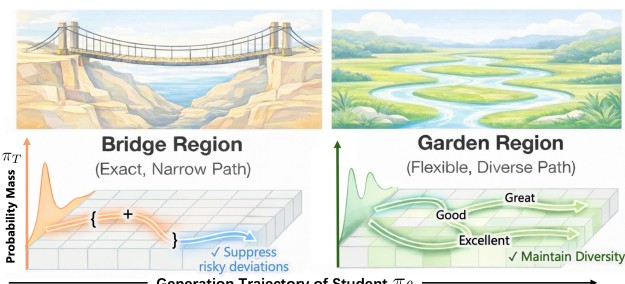

*Figure 1. The Bridge–Garden Dilemma.* Bridges require exact tokens to prevent error cascades, favoring Hard KD for risk suppression. Gardens allow flexible choices, favoring Soft KD for preserving diversity. Hybrid KD balances both for superior generative performance.

reduced exposure bias. Exposure bias (Bengio et al., 2015) refers to the performance gap that emerges when the student, conditioned on its own previously generated tokens, deviates from the teacher's generation trajectory. This autoregressive distribution shift is a core challenge in sequence-level KD (Ko et al., 2024; Gu et al., 2024; Agarwal et al., 2024). Our analysis shows that adding hard labels is particularly effective at suppressing this bias, leading to better final performance despite the weaker training fit (Fig.2c).

To explain this phenomenon, we introduce a novel conceptual and theoretical framework based on a core measure of local risk sensitivity. Within this framework, we find that the autoregressive generation process can be partitioned into two distinct regions: Bridges and Gardens (see Fig.1). **Bridge regions** exhibit **high** local risk sensitivity; here, the next token must be *exact*, as a single error can propagate and ruin the entire sequence. Conversely, **Garden regions** exhibit **low** local risk sensitivity, where token choices are more flexible and multiple alternatives can preserve both meaning and coherence. On top of this, we derive an upper bound for exposure bias that decomposes into contributions from these regions, which reveals a **Bridge–Garden Dilemma**. Our theory shows that hard-label matching excels in Bridges by concentrating probability mass on the teacher's chosen token, thereby averting cascading errors. In contrast, soft-label matching excels in Gardens by faithfully preserving the teacher's full distribution, maintaining output diversity. In this way, neither of the two pure strategies is optimal for both regions simultaneously. This insight naturally leads to a hybrid distillation objective that selectively blends hard and soft supervision based on the prefix context.

Based on this analysis, we propose a family of Bridge–Garden hybrid supervision methods that adaptively mix hard and soft labels, using confidence-, entropy-, curriculum-, and risk-guided strategies. We study a **primary suite of seven** teacher–student pairs across multiple families and scales, including **Qwen (7B→3B)**, **Llama (8B→1B)**, **Gemma (4B→1B)**, and **DeepSeek-Coder (6.7B→1.3B)**.

Extended evaluations further cover a larger Qwen2.5 capacity gap, an additional Qwen2.5-Coder pair, and open-ended generation. Across commonsense, math, and coding benchmarks, the proposed methods outperform divergence-based and on-policy baselines, while reducing training cost by **9.7×**, making our approach more practical for industrial applications.

## 2. Preliminaries

**Autoregressive Generation.** We consider sequence generation over a vocabulary $\mathcal{V}$. At each step $t$, the language model predicts the next token $y_t \in \mathcal{V}$ given an input prompt $x$ and the preceding sequence $y_{<t} := (y_1, \ldots, y_{t-1})$. We denote the **prefix** at step $t$ by $s := (x, y_{<t})$ and the **next token** by $a := y_t$. The model's behavior is defined by a **policy** $\pi(a \mid s)$, a conditional probability distribution over $\mathcal{V}$.

**Knowledge Distillation (KD)** seeks to align a *parameterized* **student** $\pi_\theta$ with a *fixed* **teacher** $\pi_T$ by minimizing the discrepancy in their predictions. Formally, the objective is to minimize the expected divergence $\mathbb{D}$ (such as KL divergence; see App.A.1) between their next-token distributions under the teacher-generated **prefix distribution** $d_T$:

$$\mathcal{L}_{d_T}(\pi_\theta) := \mathbb{E}_{s \sim d_T}\big[\mathbb{D}\big(\pi_T(\cdot \mid s) \,\|\, \pi_\theta(\cdot \mid s)\big)\big]. \quad (1)$$

This objective is empirically optimized via two approaches:

**Soft KD.** Here the student directly matches the teacher's full output distribution for every prefix $s$ by minimizing

$$\ell_{\text{soft}}(s; \theta) := \mathbb{D}\big(\pi_T(\cdot \mid s) \,\|\, \pi_\theta(\cdot \mid s)\big). \quad (2)$$

**Hard KD.** When the full distribution $\pi_T(\cdot \mid s)$ is unavailable (*e.g.,* with black-box teacher APIs (Achiam et al., 2023)) or the divergence $\mathbb{D}$ is costly to compute (Carlini et al., 2024), one can instead sample a token $a^* \sim \pi_T(\cdot \mid s)$ and train the student to maximize its log-likelihood:

$$\ell_{\text{hard}}(s; \theta) := -\log \pi_\theta(a^* \mid s). \quad (3)$$

This objective provides an unbiased estimate of $\mathbb{E}_{a \sim \pi_T(\cdot|s)}[-\log \pi_\theta(a|s)]$. When $\mathbb{D}$ is the forward KL divergence, minimizing Eq.(3) is equivalent to minimizing $\mathcal{L}_{d_T}(\pi_\theta)$, up to a constant teacher entropy. This equivalence concerns the population objective; with finite data and limited student capacity, hard and soft supervision can still lead to different training trajectories and student distributions. ***See App.A for further related work and comparisons with prior arts.***

## 3. The Hard-Label Paradox in Distillation

### 3.1. An Empirical Training–Inference Puzzle

Conventional wisdom (Hinton et al., 2015; Cho & Hariharan, 2019; Zhao et al., 2022) holds that soft labels (Eq.2),

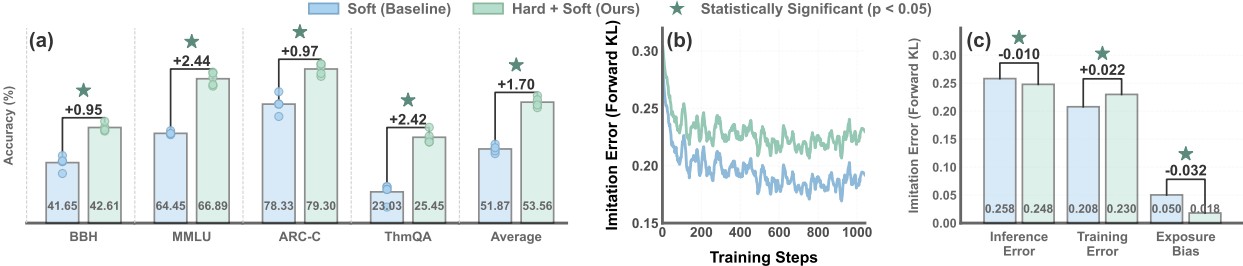

*Figure 2.* Comparative analysis (Qwen2.5-7B → 3B) of Hybrid KD ($\lambda\ell_{\text{soft}}+(1-\lambda)\ell_{\text{hard}}$) vs. Soft KD ($\ell_{\text{soft}}$). (a) benchmark performance gains, (b) student-teacher imitation error during training (quantified by Forward KL), and (c) inference imitation error decomposition based on the same metric. The experiments provided in *App.F further demonstrate the consistency* of hard-soft paradox *across various architectures, tasks, and distillation divergences.*

which encapsulate the teacher's full predictive distribution, should offer richer guidance than hard one-hot labels (Eq.3). Intuitively, matching the teacher's probabilities across all tokens ought to produce a more accurate student.

Our experiments, however, reveal the opposite trend. As shown in Fig. 2(a), a simple linear hybrid loss, termed Hybrid KD, consistently surpasses pure soft distillation across architectures, tasks, and divergence measures.

> **OBSERVATION 1.** *A linear interpolation ($\lambda\ell_{\text{soft}} + (1-\lambda)\ell_{\text{hard}}$) significantly outperforms pure Soft KD across models, benchmarks, and divergence choices ($p < 0.05$). This consistent advantage is further shown in App.F.*

Despite these robust gains, the underlying mechanism remains unclear. Without a principled explanation, the improvement appears incidental, hindering the systematic design of distillation objectives.

One natural hypothesis points to optimization: hard labels may yield sharper gradients and ease optimization. If this were the primary cause, we would expect hybrid training to help the student better imitate the teacher than pure soft distillation. However, Fig. 2(b) shows the opposite:

> **OBSERVATION 2.** *Adding hard labels worsens the training fit: the student mimics the teacher worse and yields a larger distribution discrepancy $\mathbb{D}(\pi_T \parallel \pi_\theta)$ than Soft KD. This trend is further observed in App.F.*

Since optimization difficulty cannot explain the gains, we must look elsewhere. This motivates a deeper problem:

*"If the performance gain is not from better* teacher-imitation during training*, where does it come from?"*

### 3.2. Locating the Missing Gain

To solve this, we first need to examine the total inference error of the student more carefully. During training, the student learns from teacher-generated prefixes, but at inference time it must generate tokens conditioned on its own past

outputs, a fundamentally different distribution $d_\theta$.

On top of this, we can decompose the student's total inference error into two interpretable parts. Let $\mathcal{L}_{d_T}(\pi_\theta)$ denote the training-fit error (how well the student imitates the teacher on teacher prefixes), and let $\mathcal{L}_{d_\theta}(\pi_\theta)$ be the inference imitation error (how well it performs on its own generated sequences). A simple identity relates them:

$$\mathcal{L}_{d_\theta}(\pi_\theta) = \mathcal{L}_{d_T}(\pi_\theta) + \underbrace{\left(\mathcal{L}_{d_\theta}(\pi_\theta) - \mathcal{L}_{d_T}(\pi_\theta)\right)}_{\text{Residual}}. \quad (4)$$

The first term is the familiar training-fit error. The second term captures the extra loss incurred when the student deviates from the teacher's prefix distribution. Notably, this residual has a precise and well-known meaning in sequence generation: it coincides with the exposure-bias term in KD

> **DEFINITION 3.1** *(Distillation Exposure Bias, Bengio et al. 2015) In autoregressive KD, exposure bias is defined as*
>
> $$\mathsf{EB}(\pi_\theta) := \underbrace{\mathbb{E}_{s\sim d_\theta}\left[\ell_{\text{soft}}(s)\right]}_{\text{Inference Error } \mathcal{L}_{d_\theta}(\pi_\theta)} - \underbrace{\mathbb{E}_{s\sim d_T}\left[\ell_{\text{soft}}(s)\right]}_{\text{Training Fit } \mathcal{L}_{d_T}(\pi_\theta)}.$$
>
> *It quantifies the performance gap caused by the shift from teacher prefixes $d_T$ to student-generated prefixes $d_\theta$.*

**Remark.** Unlike the classical *i.i.d.* generalization gap (Mohri et al., 2018), exposure bias stems specifically from the *autoregressive* distribution shift $d_T \to d_\theta$.

Now our earlier observations fall into place. Hard labels improve final performance (Obs.1) despite they hurt training fit (Obs.2). Eq.(4) shows the only way this can happen: hard labels must reduce exposure bias enough to outweigh their worse training fit. Fig.2(c) and results for more models and tasks in App.F confirms this directly. The drop in exposure bias indeed compensates for the rise in training error.

We now know that hard labels suppress exposure bias. But

why does this happen? We turn next to uncover the mechanism hidden behind.

# 4. The Bridge–Garden Decomposition of Exposure Bias

To answer this question, we revisit the core of exposure bias: it emerges when the student chooses a different token than the teacher, and such mistakes can accumulate over subsequent generation steps. Understanding why this bias is reduced requires us to identify where these deviations occur and how they influence future tokens.

## 4.1. The Bridge–Garden Decomposition

Our analysis starts with a simple insight: not every generation step is equally sensitive to a local deviation. In some contexts, the next token must be exact. For example, changing an operator in a mathematical derivation (*e.g.*, "+" to "−") can break the entire reasoning chain. In others, the next token can be flexible, such as replacing "excellent" with "great" in open-ended dialogue often preserves the meaning.

Motivated by this, we conceptually partition the generation process into two types of regions:

> **BRIDGES:** These are regions where *the next token must be exact*. A mistake here often invalidates the subsequent generation.
>
> **GARDENS:** These are regions where *the next token can be flexible*. A small deviation usually does not affect the subsequent generation.

Although intuitive, quantifying this distinction is nontrivial. This is because, under the student's own autoregressive generation, a single token deviation can alter all following predictions and trigger error accumulation, making it difficult to isolate token-wise effects from the overall loss.

## 4.2. Quantifying Local Risk Sensitivity via Single-Override Policy

Fortunately, we can derive a structured upper bound on exposure bias (Thm.4.1), which decomposes the global bias into a sum of local, token-level terms. Each term captures the effect of the student deviating from the teacher at a single generation step, providing a natural measure of local risk.

> **THEOREM 4.1** *(A $\kappa$-Weighted Bound on Exposure Bias) Under mild conditions (bounded loss, non-vanishing probabilities, and prefix concentrability), let $\Delta\pi_\theta := \pi_\theta - \pi_T$. Then the exposure bias $\mathsf{EB}(\pi_\theta)$ satisfies*
>
> $$\mathsf{EB}(\pi_\theta) \leq \mathbb{E}_{s \sim d_T}\Big[ F(s, \pi_\theta) \Big],$$

> *where for a constant $C_2 > 0$,*
>
> $$F(s, \pi_\theta) := \underbrace{\sum_a \kappa(a\,|\,s) \cdot |\Delta\pi_\theta(a\,|\,s)|}_{\kappa(a|s)\text{-weighted deviation}} + C_2 \|\Delta\pi_\theta(\cdot\,|\,s)\|_1^2.$$

The proof of Thm.4.1 is in App.B. To interpret $\kappa(a \mid s)$, we first introduce the notion of a single-override policy.

> **(SINGLE-OVERRIDE POLICY)** *Fix a prefix $s \in \mathcal{S}$ and token $a \in \mathcal{V}$. Define the override policy $\pi^{(s,a)}$ by*
>
> $$\pi^{(s,a)}(\cdot \mid s') := \begin{cases} \delta_a(\cdot), & s' = s, \\ \pi_T(\cdot \mid s'), & s' \neq s, \end{cases}$$
>
> *with $\delta_a$ the Dirac distribution on $a$. Let $d^{(s,a)} := d_{\pi^{(s,a)}}$ be the prefix distribution induced by $\pi^{(s,a)}$.*

> **DEFINITION 4.1** *(**Local-Risk Sensitivity**) For a prefix $s$ with its visiting probability $d_T(s) > 0$, the sensitivity $\kappa(a|s)$ measures the extra loss incurred per visit to $s$ when choosing $a$ instead of following the teacher:*
>
> $$\kappa(a|s) := \frac{\mathsf{EB}(\pi^{(s,a)})}{d_T(s)} = \frac{\mathcal{L}_{d^{(s,a)}}(\pi_\theta) - \mathcal{L}_{d_T}(\pi_\theta)}{d_T(s)},$$
>
> *where $\mathcal{L}_d(\pi_\theta) := \mathbb{E}_{s' \sim d}[\mathbb{D}(\pi_T(\cdot \mid s') \| \pi_\theta(\cdot \mid s'))]$.*

In the bound above, the contribution of a local deviation at $(s, a)$ is the product $\kappa(a|s)\,|\Delta\pi_\theta(a|s)|$. Here $|\Delta\pi_\theta(a|s)|$ measures the probability difference between student and teacher for token $a$ given $s$, and $\kappa(a|s)$ weights that difference in the upper bound.

The override policy $\pi^{(s,a)}$ acts as a prefix-conditioned token intervention for the teacher: at exactly prefix $s$ it outputs token $a$ with probability one, while following the teacher everywhere else. The difference $\mathcal{L}_{d^{(s,a)}}(\pi_\theta) - \mathcal{L}_{d_T}(\pi_\theta)$ is precisely the loss increase caused by changing the current token to $a$ at $s$. The quantity $\kappa(a|s)$ discounts this increase by the visiting probability $d_T(s)$. Thus, **a small $\kappa(a|s)$ means that deviating to $a$ at $s$ has minimal impact on exposure bias; a large $\kappa(a|s)$ signals that this single-token change can strongly affect the bias.**

Interestingly, this construction aligns with classical algorithmic stability theory (Bousquet & Elisseeff, 2002), which measures how the loss changes when a single training example is perturbed. Here the "example" is the prefix $s$, and the perturbation is forcing token $a$ whenever $s$ is reached during generation. The value $\kappa(a|s)$ quantifies the sensitivity of the loss to this token-level change. Because $\kappa$ is evaluated with the current student policy, its value can evolve during training as the student distribution changes.

So far, we have analyzed the effect of a single deviation at a given step. But what happens when we consider all possible

deviations at that step?

### 4.3. The Bridge–Garden Bound on Exposure Bias

To capture how sensitive a given step $s$ is to **all** possible token deviations, we aggregate the local sensitivities by summing over the vocabulary:

$$\kappa(s) := \sum_{a \in \mathcal{V}} \kappa(a \mid s).$$

This quantity summarizes the token-level risk at step $s$. When $\kappa(s)$ is large, many tokens have large $\kappa(a|s)$; changing the teacher's token at $s$ to one of them can substantially increase the loss. When $\kappa(s)$ is small, most $\kappa(a|s)$ are small, so for the majority of tokens a deviation at $s$ has only a mild effect. Accordingly, high- and low-sensitivity $\kappa(s)$ regions correspond naturally to the notions of Bridges and Gardens introduced earlier (Sec.4.1).

> **DEFINITION 4.2 (The Bridge–Garden Partition)**
> Given a task- and model-dependent threshold $\tau$, partition the prefix space $\mathcal{S}$ into
>
> **Bridge:** $\quad \mathcal{B} := \{s \in \mathcal{S} : \kappa(s) > \tau\},$
>
> **Garden:** $\quad \mathcal{G} := \{s \in \mathcal{S} : \kappa(s) \leq \tau\}.$

This partition provides a structured lens for analyzing the bound of Thm.4.1, as formalized next (proof in App.C).

> **PROPOSITION 4.1 (The Bridge–Garden Upper Bound on Exposure Bias)** Using the partition from Def.4.2, the bound from Thm.4.1 decomposes as follows:
>
> $$F(\pi_\theta) := F_{\mathcal{B}}(\pi_\theta) + F_{\mathcal{G}}(\pi_\theta),$$
>
> where $F_{\mathcal{X}}(\pi_\theta) := \mathbb{E}_{s \sim d_T}[\mathbb{1}_{\mathcal{X}}(s)F(s, \pi_\theta)]$ for $\mathcal{X} \in \{\mathcal{B}, \mathcal{G}\}$, and the term $F(s, \pi_\theta)$ is defined in Thm.4.1.

The decomposition reveals an asymmetry in how a student should learn from the teacher:

In Bridges $\mathcal{B}$, only a few tokens are safe; most alternatives carry high risk. Hence $F_{\mathcal{B}}$ is dominated by deviations on tokens with large $\kappa(a|s)$. To minimize $F_{\mathcal{B}}$, the student must avoid these high-risk tokens and concentrate probability on the few safe options supported by the teacher.

In Gardens $\mathcal{G}$, many next-token choices are acceptable and their $\kappa(a|s)$ are uniformly small. Because no single deviation incurs a large penalty, the bound behaves like a standard distribution-matching distance. Minimizing $F_{\mathcal{G}}$ encourages the student to match the teacher's broad distribution, preserving the diversity inherent in Gardens.

### 4.4. Mechanism of Hybrid Improvement

*NOTE: Hard KD and Soft KD refer to Eq.(3) and Eq.(2), respectively. Hybrid KD is defined in Obs.1.*

To reduce total exposure bias, we must keep it low in both Bridge and Garden regions. We next show that neither Hard nor Soft KD alone can achieve this goal.

Hard KD employs one-hot teacher targets, pushing the student to place most probability on the teacher's single preferred token. This is effective in Bridges, but fails in Gardens because it suppresses distributional diversity. Soft KD matches the teacher's full distribution, thereby keeping probability spread over many teacher-supported tokens. This works well in Gardens. But in Bridges, it may assign non-negligible probability to high-risk alternatives.

If, however, we adopt a Hybrid KD regime, we can reduce the loss in both regions by choosing an appropriate mixing coefficient $\lambda$, as empirically observed. The following one-step theorem makes this intuition precise (proof in App.D).

> **THEOREM 4.2 (Complementarity Gain) (Informal)**
> Let $\pi_{hard}$ and $\pi_{soft}$ be (near-)minimizers of $\ell_{hard}$ and $\ell_{soft}$, respectively; in late-stage distillation,
>
> $$F_{\mathcal{G}}(\pi_{soft}) < F_{\mathcal{G}}(\pi_{hard}), \quad F_{\mathcal{B}}(\pi_{hard}) < F_{\mathcal{B}}(\pi_{soft}).$$
>
> This suggests that for some $\lambda \in (0, 1)$, the minimizer $\pi_{hyb}$ of the hybrid objective $(\lambda \ell_{soft} + (1-\lambda)\ell_{hard})$ can achieve
>
> $$F(\pi_{hyb}) < \min\{F(\pi_{hard}), F(\pi_{soft})\}.$$

Thm.4.2 formalizes the complementarity: the hard-KD solution $\pi_{hard}$ has smaller $F_{\mathcal{B}}$ but larger $F_{\mathcal{G}}$ than the soft-KD solution $\pi_{soft}$, and vice versa. No single objective can keep both $F_{\mathcal{B}}$ and $F_{\mathcal{G}}$ small simultaneously. Blending the losses produces a hybrid model with a strictly smaller $F(\pi)$ than either extreme, tightening the exposure-bias bound and aligning with the empirical gains of hybrid KD reported in Sec.3.

## 5. Practical Algorithms for Bridge-Garden Hybrid Supervision

Guided by the Bridge–Garden analysis in Sec.4, we now develop practical algorithms that seek hard-label supervision in Bridges while preserving distributional diversity in Gardens. A natural starting point is to combine the two losses with a fixed mixing coefficient $\lambda$:

$$\ell_{hyb}(s; \theta) = \lambda \cdot \ell_{soft}(s; \theta) + (1 - \lambda) \cdot \ell_{hard}(s; \theta). \quad (5)$$

Although this static hybrid already outperforms pure soft or hard KD (Obs.1), it does not fully reflect the distinct needs of Bridges and Gardens. We next explore two different strategies: **(i)** dynamically tuning the weight $\lambda$ (Methods 1–3) and **(ii)** modifying the hard-loss term itself (Method 4).

**1) Confidence-based weighting.** Teacher's confidence in its top prediction serves as a simple proxy for distinguishing

*Table 1.* Accuracy (↑) and the mean score (Avg.) on general reasoning benchmarks for Qwen2.5 and Llama3 models under different hybrid strategies (with Forward KL as soft supervision). Best and second-best results are **bolded** and underlined.

| Method | Llama3.1-8B → Llama3.2-1B | | | | | Qwen2.5-7B → Qwen2.5-3B | | | | |
|---|---|---|---|---|---|---|---|---|---|---|
| | BBH | MMLU | ARC-C | ThmQA | Avg. | BBH | MMLU | ARC-C | ThmQA | Avg. |
| *Teacher* | $57.72_{\pm0.07}$ | $70.90_{\pm0.03}$ | $83.58_{\pm0.10}$ | $18.10_{\pm0.40}$ | 57.58 | $64.66_{\pm0.69}$ | $78.22_{\pm0.22}$ | $89.90_{\pm0.24}$ | $32.47_{\pm0.32}$ | 66.31 |
| *Student (No Distill)* | $14.01_{\pm4.41}$ | $19.78_{\pm9.89}$ | $21.57_{\pm9.94}$ | $2.22_{\pm0.44}$ | 14.40 | $22.34_{\pm0.06}$ | $64.61_{\pm0.06}$ | $78.40_{\pm0.03}$ | $12.22_{\pm0.25}$ | 44.39 |
| Hard KD (Kim & Rush, 2016) | $15.29_{\pm2.75}$ | $22.54_{\pm1.38}$ | $23.98_{\pm1.96}$ | $3.88_{\pm0.73}$ | 16.42 | $41.52_{\pm0.33}$ | $65.76_{\pm0.18}$ | $78.75_{\pm0.71}$ | $23.75_{\pm0.40}$ | 52.45 |
| Soft KD (Hinton et al., 2015) | $22.07_{\pm2.11}$ | $33.13_{\pm1.69}$ | $33.41_{\pm1.78}$ | $4.37_{\pm0.43}$ | 23.25 | $41.65_{\pm0.16}$ | $64.45_{\pm0.04}$ | $78.33_{\pm0.22}$ | $23.02_{\pm0.33}$ | 51.87 |
| Static Weighting (Ours) | $23.03_{\pm1.26}$ | $34.12_{\pm1.80}$ | $34.35_{\pm1.53}$ | $5.42_{\pm0.73}$ | 24.23 | $42.61_{\pm0.09}$ | $66.89_{\pm0.19}$ | $79.30_{\pm0.14}$ | $\mathbf{25.45}_{\pm0.23}$ | 53.56 |
| Confidence-based Weighting (Ours) | $25.64_{\pm0.80}$ | $35.32_{\pm1.45}$ | $34.68_{\pm2.24}$ | $3.97_{\pm0.78}$ | 24.90 | $44.07_{\pm0.19}$ | $67.50_{\pm0.19}$ | $80.77_{\pm0.27}$ | $22.78_{\pm0.28}$ | 53.78 |
| Entropy-based Weighting (Ours) | $24.40_{\pm0.88}$ | $35.93_{\pm1.56}$ | $34.78_{\pm2.61}$ | $5.83_{\pm0.53}$ | 25.23 | $\mathbf{46.83}_{\pm0.16}$ | $67.06_{\pm0.14}$ | $79.81_{\pm0.79}$ | $23.65_{\pm0.20}$ | 54.34 |
| Curriculum Schedule (Ours) | $25.06_{\pm1.19}$ | $\mathbf{36.13}_{\pm1.54}$ | $35.29_{\pm1.16}$ | $\mathbf{6.18}_{\pm0.68}$ | 25.67 | $44.39_{\pm0.18}$ | $67.32_{\pm0.20}$ | $80.51_{\pm0.10}$ | $22.13_{\pm0.47}$ | 53.59 |
| Risk-Guided Hybrid (Ours) | $\mathbf{27.44}_{\pm2.85}$ | $35.64_{\pm3.22}$ | $\mathbf{37.01}_{\pm3.98}$ | $5.00_{\pm1.25}$ | **26.27** | $46.53_{\pm0.05}$ | $\mathbf{69.05}_{\pm0.07}$ | $\mathbf{81.23}_{\pm0.00}$ | $23.82_{\pm0.58}$ | **55.16** |

Bridges from Gardens. In Bridges, a well-trained teacher typically places most probability mass on a small set of low-risk next tokens, since high-risk deviations can trigger error accumulation. Motivated by this, we reduce the soft-label weight when the teacher is confident:

$$\lambda_{\mathrm{conf}}(s) = 1 - \max_a \pi_T(a \mid s),$$

**2) Entropy-based weighting.** Alternatively, we can use the teacher's predictive entropy as a global uncertainty signal: higher entropy indicates that the teacher considers the next-token choice flexible, aligning with the behavior of Gardens. We thus set the soft-loss weight to the normalized entropy:

$$\lambda_{\mathrm{ent}}(s) = \frac{H_T(s)}{\log |\mathcal{V}|}, \quad H_T(s) = -\sum_{a\in\mathcal{V}} \pi_T(a|s) \log \pi_T(a|s).$$

**3) Curriculum schedule.** Since mistakes in Bridges can lead to error accumulation, it can be useful to emphasize hard labels early in training, letting the student first learn these critical regions reliably. The weight on soft labels is then gradually increased to capture the diversity present in Gardens. We implement this with a linear warm-up:

$$\lambda(t) = \min\left(\frac{t}{T}, \, 1\right) \cdot \lambda_{\mathrm{max}},$$

where $\lambda_{\mathrm{max}}$ is the target soft-label weight, $t$ is the current training step, and $T$ is the number of warm up steps.

**4) Risk-Guided Hybrid.** Beyond adjusting the mixing coefficient, we can also revise the hard loss from a reinforcement learning perspective: the risk $\kappa(a|s)$ can be related to reward, where higher risk implies lower reward. From this perspective, reward is concentrated in Bridges but more uniform in Gardens. Recall that maximizing reward incentivizes the consistency between the policy (student distribution) and the reward distribution. In this way, the student

tends to behave sharply in Bridges and preserve diversity in Gardens. This directly extends recent reward-optimization advances for LMs (Cundy & Ermon, 2023) to KD, and thus we have the following objective:

$$\ell'_{\mathrm{hard}}(s; \theta) \coloneqq \ell_{\mathrm{hard}}(s; \theta) + \frac{\alpha}{4} \Delta_\theta(s, a^*)^2,$$

where $\alpha > 0$, $a^*$ is the hard target in $\ell_{\mathrm{hard}}$ (Sec.2), and $\Delta_\theta(s, a^*) \coloneqq \log\sum_{a'\in\mathcal{V}} \exp\big(f_\theta(a' \mid [s, a^*]) - f_\theta(a^* \mid s)\big)$, with $f_\theta(\cdot \mid s)$ the student logits. Here $[s, a^*]$ denotes the prefix obtained by appending the hard target $a^*$ to $s$. In all experiments, we fix $\alpha = 0.1$; the additional term reuses the student logits and only adds one log-sum-exp computation per token, so its cost remains close to standard soft KD (see App.F.1).

## 6. Experiments

We now conduct empirical experiments to validate our theoretical analysis and to assess the practical effectiveness of the proposed hybrid distillation framework. Specifically, we seek to answer three key questions:

**Q1 (Hybrid Strategy):** How do different hybrid KD variants impact distillation performance?

**Q2 (Universality):** Are the gains from hybrid distillation consistent across different divergence measures?

**Q3 (vs. On-Policy Methods):** Compared to recent on-policy distillation techniques, does our approach improve final accuracy while being more efficient?

In App. F, we extend the primary benchmark study in three directions. We first test whether the same hard–soft pattern holds under larger teacher–student capacity gaps, open-ended generation, and on-policy prefixes. We then measure training cost and compare against simpler alternatives, including reverse-proxy rules, regularization, and temperature

*Table 2.* Performance comparison of Qwen2.5 on reasoning benchmarks. Avg. is the mean across all benchmarks. We evaluate our hybrid KD (Forward KL) against recent soft KD methods. ***Notably, our approach is orthogonal to these works (see Fig. 3 for gains when combined)***. Best and second-best results are highlighted, also denoted by **bold** and underlined values.

| Method | Qwen2.5-7B → Qwen2.5-0.5B | | | | | Qwen2.5-7B → Qwen2.5-3B | | | | |
|---|---|---|---|---|---|---|---|---|---|---|
| | BBH | MMLU | ARC-C | ThmQA | Avg. | BBH | MMLU | ARC-C | ThmQA | Avg. |
| Reverse KL (Gu et al., 2024) | $24.91_{\pm0.01}$ | $44.72_{\pm0.01}$ | $47.44_{\pm0.00}$ | $11.00_{\pm0.00}$ | 32.02 | $44.07_{\pm0.10}$ | $65.67_{\pm0.19}$ | $77.68_{\pm0.12}$ | **$24.20_{\pm0.49}$** | 52.90 |
| Total Variation (Wen et al., 2023) | $26.74_{\pm0.38}$ | $44.35_{\pm0.12}$ | $46.76_{\pm0.51}$ | $10.20_{\pm0.56}$ | 32.01 | $40.50_{\pm0.16}$ | $64.52_{\pm0.03}$ | $78.11_{\pm0.18}$ | $22.83_{\pm0.56}$ | 51.49 |
| JS divergence (Agarwal et al., 2024) | $24.55_{\pm0.13}$ | $43.31_{\pm0.10}$ | $44.78_{\pm0.05}$ | **$11.82_{\pm0.37}$** | 31.12 | $45.50_{\pm0.08}$ | $64.68_{\pm0.17}$ | $78.85_{\pm0.14}$ | $22.27_{\pm0.41}$ | 52.83 |
| Adaptive KL (Wu et al., 2025) | $26.04_{\pm0.23}$ | $41.85_{\pm0.11}$ | $45.89_{\pm0.32}$ | $11.33_{\pm0.61}$ | 31.28 | $44.71_{\pm0.13}$ | $64.69_{\pm0.07}$ | $79.23_{\pm0.22}$ | $22.25_{\pm0.33}$ | 52.72 |
| Skew FKL (Ko et al., 2024; 2025) | $25.87_{\pm0.24}$ | $44.12_{\pm0.24}$ | $47.12_{\pm0.65}$ | $10.65_{\pm0.22}$ | 31.94 | $41.39_{\pm0.17}$ | $64.67_{\pm0.15}$ | $77.75_{\pm0.23}$ | $23.77_{\pm0.71}$ | 51.89 |
| Skew RKL (Ko et al., 2024; 2025) | **$28.16_{\pm0.16}$** | $45.03_{\pm0.05}$ | $47.51_{\pm0.20}$ | $11.37_{\pm0.66}$ | **33.02** | $41.22_{\pm0.08}$ | $63.95_{\pm0.08}$ | $76.91_{\pm0.18}$ | $23.67_{\pm0.20}$ | 51.44 |
| $\alpha$-$\beta$ divergence (Wang et al., 2025b) | $26.18_{\pm0.15}$ | $42.59_{\pm0.18}$ | $46.70_{\pm0.37}$ | $11.07_{\pm0.43}$ | 31.63 | $45.12_{\pm0.23}$ | $64.95_{\pm0.17}$ | $79.81_{\pm0.09}$ | $22.94_{\pm0.54}$ | 53.21 |
| **HybKD (Ours)** | $26.58_{\pm0.16}$ | **$49.08_{\pm0.22}$** | **$51.69_{\pm0.62}$** | $10.50_{\pm0.54}$ | **34.46** | **$46.53_{\pm0.05}$** | **$69.05_{\pm0.07}$** | **$81.23_{\pm0.00}$** | $23.82_{\pm0.58}$ | **55.16** |

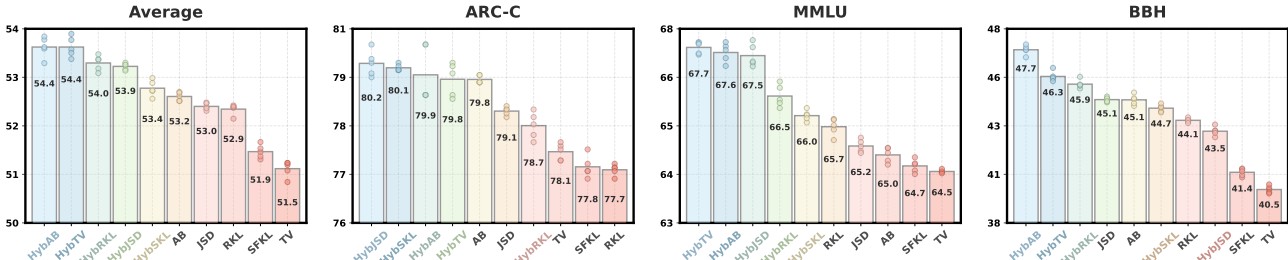

*Figure 3.* Performance achieved by integrating various divergences with hard-label supervision (via static weighting). HybRKL stands for the hybrid of Reverse KL and hard labels, with other hybrid variants named similarly. More numerical results are available in App. F.

*Table 3.* Distillation results on math benchmarks (↑).

| Method | Qwen2.5-Math-7B → 1.5B | | | |
|---|---|---|---|---|
| | GSM8K | MATH | Gaokao23 | Avg. |
| Hard KD | $65.75_{\pm4.43}$ | $50.44_{\pm1.78}$ | $41.04_{\pm3.15}$ | 52.41 |
| Forward KL | $68.72_{\pm0.92}$ | $49.89_{\pm0.82}$ | $41.77_{\pm1.11}$ | 53.46 |
| Reverse KL | $58.98_{\pm4.44}$ | $46.78_{\pm4.02}$ | $37.35_{\pm4.68}$ | 47.70 |
| Skew F(R)KL | $69.13_{\pm1.62}$ | $49.59_{\pm1.63}$ | $42.23_{\pm1.92}$ | 53.65 |
| $\alpha$-$\beta$ divergence | $70.86_{\pm2.29}$ | $50.68_{\pm1.30}$ | $42.88_{\pm2.34}$ | 54.81 |
| **HybKD (Ours)** | **$71.65_{\pm1.20}$** | **$51.37_{\pm0.71}$** | **$44.10_{\pm2.09}$** | **55.71** |

*Table 4.* DeepSeek-Coder distillation results (↑). **HE**: HumanEval.

| Method | DeepSeek-Coder-6.7B → 1.3B | | | | |
|---|---|---|---|---|---|
| | HE | HE+ | MBPP | MBPP+ | Avg. |
| Hard KD | 35.37 | 32.93 | 60.32 | 50.00 | 44.66 |
| Forward KL | 38.41 | 33.54 | **63.49** | **51.59** | 46.76 |
| Reverse KL | 39.02 | 34.15 | 62.17 | 50.00 | 46.34 |
| Skew F(R)KL | **42.07** | 35.98 | 60.85 | 50.26 | 47.29 |
| $\alpha$-$\beta$ divergence | 41.46 | **36.59** | 60.05 | 50.26 | 47.09 |
| **HybKD (Ours)** | 41.46 | **36.59** | 63.12 | 50.39 | **47.89** |

changes. Finally, we use controlled synthetic domains where exact token-level $\kappa$ is computable, so the Bridge–Garden decomposition can be tested directly rather than only inferred from LLM benchmarks.

### 6.1. Experimental Setup

**Models and Benchmarks.** Our primary benchmark suite evaluates seven teacher–student pairs from **Qwen2.5** (Qwen et al., 2025) (7B→{0.5, 1.5, 3}B), **Llama** (Grattafiori et al., 2024) (8B→1B), **Gemma-3** (Gemma Team et al., 2025) (4B→1B), **Qwen-Math** (Yang et al., 2024) (7B→1.5B), and **DeepSeek-Coder** (Guo et al., 2024) (6.7B→1.3B). Evaluation spans commonsense (MMLU (Hendrycks et al., 2020),

BBH (Suzgun et al., 2022), ARC-C (Clark et al., 2018), ThmQA (Chen et al., 2023)), math (GSM8K (Cobbe et al., 2021), MATH (Hendrycks et al., 2021), Gaokao23 (Liao et al., 2024)), and code (HumanEval (Chen, 2021), MBPP (Austin et al., 2021)). App. F adds two further teacher–student settings and open-ended evaluation. We report five-seed mean accuracy. Training and evaluation details are in App. E.

**Baselines.** We compare against three baseline categories: **(1) Standard off-policy KD.** Hard KD (Kim & Rush, 2016), which trains on teacher-generated tokens, and Forward KL distillation (Hinton et al., 2015). **(2) Alternative divergence objectives.** Method using Reverse KL (Gu et al., 2024),

*Table 5.* Accuracy (↑) and mean score (Avg.) on Llama and Gemma models. Best and second-best results are highlighted.

| Method | Llama3.1-8B → Llama3.2-1B | | | | | Gemma3-4B → Gemma3-1B | | | | |
|---|---|---|---|---|---|---|---|---|---|---|
| | BBH | MMLU | ARC-C | ThmQA | Avg. | BBH | MMLU | ARC-C | ThmQA | Avg. |
| Reverse KL (Gu et al., 2024) | $23.69_{\pm0.77}$ | $31.63_{\pm1.88}$ | $32.08_{\pm1.20}$ | $3.60_{\pm0.09}$ | 22.75 | $9.52_{\pm1.03}$ | $24.77_{\pm0.68}$ | $27.58_{\pm0.60}$ | $1.26_{\pm0.32}$ | 15.78 |
| Total Variation (Wen et al., 2023) | $23.55_{\pm1.51}$ | $27.41_{\pm4.96}$ | $28.99_{\pm4.47}$ | $2.68_{\pm0.56}$ | 20.66 | $6.29_{\pm0.86}$ | $25.07_{\pm0.31}$ | $25.83_{\pm0.43}$ | $1.29_{\pm0.41}$ | 14.62 |
| JS divergence (Agarwal et al., 2024) | $25.16_{\pm2.03}$ | $34.08_{\pm2.15}$ | $34.42_{\pm1.81}$ | $5.70_{\pm0.13}$ | 24.84 | $7.14_{\pm0.74}$ | $25.27_{\pm0.56}$ | $26.03_{\pm0.33}$ | $4.57_{\pm0.97}$ | 15.75 |
| Adaptive KL (Wu et al., 2025) | $25.18_{\pm1.86}$ | $33.86_{\pm2.34}$ | $33.74_{\pm1.77}$ | $4.85_{\pm0.44}$ | 24.41 | $5.47_{\pm0.63}$ | $24.76_{\pm1.56}$ | $24.07_{\pm1.12}$ | $4.23_{\pm0.41}$ | 14.63 |
| Skew FKL (Ko et al., 2024; 2025) | $25.05_{\pm1.40}$ | $31.39_{\pm2.84}$ | $32.34_{\pm2.99}$ | $4.73_{\pm0.86}$ | 23.37 | $7.01_{\pm0.82}$ | $25.08_{\pm0.88}$ | $26.14_{\pm0.70}$ | $5.01_{\pm0.29}$ | 15.81 |
| Skew RKL (Ko et al., 2024; 2025) | $25.24_{\pm0.79}$ | $33.00_{\pm1.40}$ | $32.24_{\pm1.36}$ | $4.27_{\pm0.44}$ | 23.69 | $5.83_{\pm0.25}$ | $25.17_{\pm0.22}$ | $25.93_{\pm0.29}$ | $3.38_{\pm0.26}$ | 15.08 |
| $\alpha$-$\beta$ divergence (Wang et al., 2025b) | $25.07_{\pm1.36}$ | $34.78_{\pm2.04}$ | $34.90_{\pm1.45}$ | $5.27_{\pm0.44}$ | 25.01 | $7.26_{\pm0.78}$ | $25.49_{\pm1.00}$ | $26.07_{\pm0.66}$ | $4.34_{\pm0.89}$ | 15.79 |
| **HybKD (Ours)** | $27.44_{\pm2.85}$ | $35.64_{\pm3.22}$ | $37.01_{\pm3.98}$ | $5.00_{\pm1.25}$ | **26.27** | $7.78_{\pm1.19}$ | $25.94_{\pm0.55}$ | $26.07_{\pm0.85}$ | $5.24_{\pm0.79}$ | **16.26** |

JS divergence (Agarwal et al., 2024), Total Variation (TV) (Wen et al., 2023), Adaptive KL (Wu et al., 2025), Skew FKL/RKL (Ko et al., 2024; 2025), and the $\alpha$–$\beta$ divergence framework (Wang et al., 2025b). **(3) On-policy techniques.** Recent methods that train the student on its own generated sequences (Gu et al., 2024; Agarwal et al., 2024; Ko et al., 2025), which directly mitigate exposure bias by aligning training and inference distributions.

## 6.2. Q1: Effectiveness of Hybrid Strategy

We evaluate various hybrid strategies against pure Hard and Soft KD to answer **Q1**. From Tab.1, we observe: **1) Neither pure baseline is consistently superior.** Soft KD leads on Llama-3.2-1B (+6.83 average score), while Hard KD leads on Qwen2.5-3B (+0.59). This suggests their complementary potential. **2) Hybrids consistently outperform pure KD.** Even simple static weighting improve average scores (+1.69 on Qwen2.5-3B), with adaptive methods (*e.g.,* confidence-, entropy-, risk-guided) providing further gains. **3) Curriculum scheduling becomes highly competitive for large capacity gaps** (Llama3.1-8B → 3.2-1B). Its success in this setting supports that prioritizing Bridge regions is essential for stable learning when student capacity is limited.

## 6.3. Q2: Universality across Divergence Measures

We evaluate our method across various divergences to answer **Q2** and find: 1) Our method excels with a fixed Forward KL. It **outperforms the best non-hybrid baseline** by +1.95 average points on Qwen2.5-3B (Tab.2). Consistent gains are also observed across more models (*e.g.,* Llama, Gemma) and domains (*e.g.,* math, coding), as shown in Tabs.3, 4, and 5. **2) The hybrid principle is broadly applicable.** It consistently improves other measures like Reverse or Skew KL (Fig.3). These results show that the benefits of hybrid supervision come from the adaptive mechanism rather than a specific divergence choice.

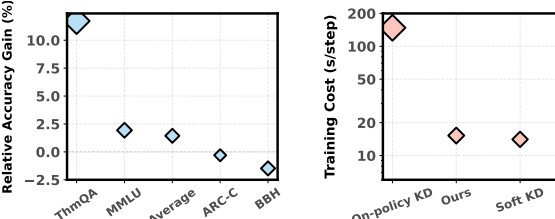

*Figure 4.* Relative accuracy gain and training cost (s/step) versus On-policy KD under *identical experimental settings* (see App.E.2).

## 6.4. Q3: Comparison with On-Policy Approach

To answer **Q3**, we compare our method with on-policy KD in the Qwen2.5-7B → 3B setting. As shown in Fig.4, our approach (with only static weighting) achieves a 1.43% average performance gain over this baseline. Crucially, while our method maintains training costs comparable to standard Hard/Soft KD, it is **9.7x more efficient** than on-policy KD, making it more practical for large-scale deployment. Because hybrid supervision modifies the target distribution rather than the prefix source, it can also be combined with on-policy prefix sampling; App. F.1 shows that this combination consistently improves over on-policy soft KD across reasoning and code tasks.

## 7. Additional Analysis

To better understand the behavior of hybrid distillation, we examine its properties from the following two aspects.

**Impact of Mixing Weight $\lambda$.** We evaluate Hybrid KD across a range of fixed weights. As shown in Fig.5, intermediate values of $\lambda \in (0, 1)$ consistently outperform both pure hard ($\lambda = 0$) and soft ($\lambda = 1$) distillation. We also observe that the optimal weight varies across tasks and models. This finding highlights the importance of the adaptive strategies introduced in Sec.5 based on our Bridge–Garden theory.

**Entropy Analysis.** We further examine the student's next-

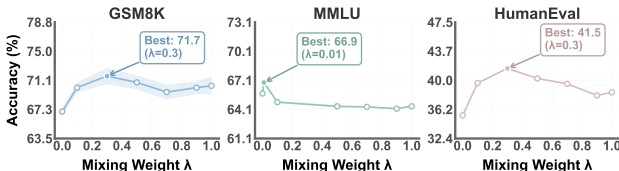

Figure 5. Effect of mixing weight $\lambda$ on student performance.

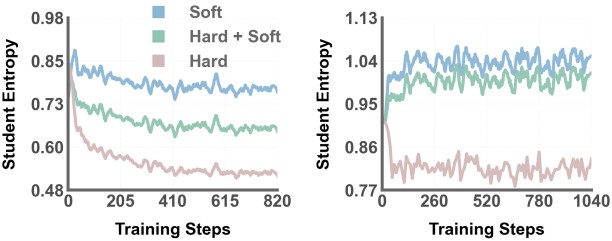

Figure 6. Student entropy during training. Left: DeepSeek-Coder-1.3B on code generation; Right: Qwen2.5-3B on general reasoning.

token entropy. As illustrated in Fig. 6, Hard KD induces sharper (low-entropy) distributions while Soft KD yields flatter ones, aligning with their roles in Bridge and Garden regions. Hybrid KD maintains an intermediate entropy, balancing Bridge precision with Garden diversity to achieve superior performance over single-label baselines. App. F checks the main alternative explanations directly: reversing the confidence or entropy rule hurts performance, global regularization and temperature changes do not consistently match Hybrid KD, and controlled synthetic experiments show that the largest exact $\kappa$ values concentrate on semantic decision states.

## 8. Conclusion

In this work, we identify a hard-label paradox where hard supervision enhances student inference by reducing exposure bias, despite degrading teacher imitation during training. We capture this with a Bridge–Garden partition of generation into exact-prediction regions (Bridges) and flexible regions (Gardens). Hard KD excels at suppressing risky deviations in Bridges, whereas Soft KD specializes in preserving diversity in Gardens. Based on this insight, we develop hybrid strategies that adaptively combine their behaviors. Empirical results show that our methods consistent improvement over existing baselines, with up to $9.7\times$ lower training cost.

## Impact Statement

This work aims to improve the efficiency and reliability of knowledge distillation for language models, potentially reducing inference costs and enabling deployment under tighter compute and energy budgets. As with most advances in model training and compression, these techniques could be used for beneficial applications (*e.g.,* accessibility, on-device assistance) as well as for misuse. We encourage future work to evaluate downstream safety implications and to pair efficiency gains with robust safeguards.

## Acknowledgements

This work was supported in part by the National Natural Science Foundation of China under Grants 62525212, U23B2051, 62236008, 62441232, 62521007, U21B2038, 62576332, and 62502500, in part by the Youth Innovation Promotion Association CAS, in part by the Strategic Priority Research Program of the Chinese Academy of Sciences under Grant XDB0680201, in part by the project ZR2025ZD01 supported by Shandong Provincial Natural Science Foundation, in part by the China National Postdoctoral Program for Innovative Talents under Grant BX20240384, in part by Beijing Natural Science Foundation under Grant L252144, in part by General Program of the Chinese Postdoctoral Science Foundation under Grant 2025M771558, in part by the Beijing Major Science and Technology Project under Contract Z251100008125059, and in part by Beijing Academy of Artificial Intelligence (BAAI).

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

# Appendix

## Table of Contents

# A. Related Work

In this section, we review related work and discuss our contributions in the context of recent studies.

## A.1. Soft-Label Knowledge Distillation

The seminal work by (Hinton et al., 2015) first proposed leveraging the output distribution of a more powerful teacher model to supervise the training of a student model, minimizing the predictive discrepancy between them via Forward KL divergence. This methodology achieved marked progress across diverse domains (Li et al., 2023b; Wang et al., 2021; Tian et al., 2025; Li et al., 2024; Wang et al., 2024a;b; Zhao et al., 2024; 2025; Yang et al., 2026; Wang et al., 2026) and established a foundational framework for extensive subsequent research (Cho & Hariharan, 2019; Zhao et al., 2022; Huang et al., 2022; Li et al., 2023a; Beyer et al., 2022; Chen et al., 2022; Hao et al., 2023; Zheng & YANG, 2024; Sun et al., 2024; Busbridge et al., 2025; Harutyunyan et al., 2023; Peng et al., 2025). While highly effective, recent studies have noted that Forward KL divergence tends to induce mode-seeking behavior, which can undermine the student's focused learning of critical categories (Gu et al., 2024; Ko et al., 2024; Wen et al., 2023; Wu et al., 2025; Wang et al., 2025b; Gu et al., 2025). Therefore, some studies have turned to alternative distillation divergence objectives, such as Reverse KL divergence (Lee et al., 2023; Kim et al., 2024; Gu et al., 2024), Maximum Mean Discrepancy (Huang & Wang, 2017), Wasserstein Distance (Lv et al., 2024), Total Variation distance (Wen et al., 2023), Jensen-Shannon divergence (Binici et al., 2022; Agarwal et al., 2024), Skew Forward/Reverse divergence (Ko et al., 2024; 2025; Shin et al., 2025), and $\alpha$-$\beta$-divergence (Wang et al., 2025b). There are also approaches that optimize a weighted sum of multiple divergences during training (Amara et al., 2022; Wu et al., 2025). These methods have achieved marked empirical progress by enabling the student to better match the overall teacher's predictive distribution.

Crucially, the technique we propose in this work is complementary to these promising methods. As shown in Fig.3, our hybrid supervision approach can be seamlessly integrated with various distillation divergences outlined above to achieve further performance gains.

## A.2. Hard-Label Knowledge Distillation

Despite the success of soft-label distillation, applying it to large language models can face practical constraints: top-tier teacher models are often closed-source (*e.g.,* ChatGPT (Achiam et al., 2023)), and using them continuously during training remains prohibitively expensive (Gu et al., 2025). To overcome these barriers, hard-label distillation (Kim & Rush, 2016) uses the teacher model to generate corresponding output sequences for a set of input prompts, thereby constructing a labeled dataset on which the student is trained via standard maximum likelihood estimation (Chiang et al., 2023; Wang et al., 2023; Taori et al., 2023; Peng et al., 2023; Wang et al., 2025a). In practice, this direct approach has yielded solid advances (Guo et al., 2025).

In this work, we revisit the effect of hard labels in LLM distillation. We start from the empirical observation that blending hard and soft labels improves performance, then formalize the cause through our Bridge–Garden framework. Our theory indicates that using only hard or soft labels can result in suboptimal performance, whereas combining both can resolve this limitation.

## A.3. Hard-Soft Hybrid Supervision in Knowledge Distillation

Several studies in image classification have also investigated strategies to combine hard and soft labels for student training. However, a fundamental distinction exists regarding the nature of the hard labels. Unlike in LLM distillation, where hard labels are typically generated by the teacher (representing merely a subset of the teacher's distribution, as discussed in Sec. 3.1) (Kim & Rush, 2016), image classification tasks generally utilize ground-truth labels from the training dataset. In that context, hard labels provide accurate supervision signals even when the teacher makes incorrect predictions. Despite this structural difference, prior research has still achieved significant progress by effectively combining hard and soft labels (Zhou et al., 2021; Zheng & YANG, 2024; Ren et al., 2023; Ham et al., 2025).

For instance, (Zhou et al., 2021) decomposes the knowledge distillation objective via bias-variance analysis and weights the hard and KL losses based on sample difficulty, ensuring the teacher provides better guidance for samples that are difficult for the student. (Zheng & YANG, 2024) proposes using teacher entropy as a weighting factor to focus on samples where the teacher's predictions are more diverse. Additionally, (Li et al., 2023a; Jafari et al., 2021) balances the contributions of hard and KL losses by scaling the KL loss with a curriculum-based temperature coefficient. (Ganguly et al., 2024)

uses a teacher-loss-based decay function to prioritize easy samples and progressively shift focus to harder instances. (Ren et al., 2023) assigns higher loss weights to samples that yield greater gradient similarity between the student's training update and the validation set performance, ensuring the teacher prioritizes teaching what helps the student generalize. (Ham et al., 2025) employs soft-label distillation on refusal responses to smooth the loss surface and hard-label supervision on filtered benign samples to mitigate gradient conflicts between safety and downstream tasks. Recent token-adaptive methods for autoregressive LMs also adjust the teaching mode or divergence at each token (Zhong et al., 2024; Jung et al., 2025). Our work studies a different axis of adaptation: mixing the teacher-sampled hard token with the teacher's full next-token distribution.

Despite these successful and mature developments in image classification, **the application of hybrid supervision in LLM distillation remains largely unexplored, and theoretical analysis explaining its effectiveness is still lacking**. In this work, we formally demonstrate that this approach can be effectively applied to LLM distillation. More importantly, our Bridge-Garden framework reveals that its efficacy stems from better mitigating the complex issue of exposure bias.

### A.4. Exposure Bias in Knowledge Distillation

A key challenge in distilling autoregressive models is exposure bias (Bengio et al., 2015), which arises from a fundamental distribution mismatch: during training, the student learns from sequences produced by the teacher, yet at inference time it must generate from its own previous outputs. This discrepancy can cause significant degradation in performance.

To address this, on-policy distillation methods (Agarwal et al., 2024; Gu et al., 2024; Ko et al., 2024; Rosset et al., 2024) train the student on its own generated outputs, with the teacher providing corrective feedback. This directly aligns the training and inference distributions and yields effective improvements in practice (Yang et al., 2025). (Xu et al., 2025) further leverages speculative decoding to use the teacher's distribution to correct the noise introduced by low-quality early student generations.

Although prior work has proposed various heuristics to mitigate exposure bias, a systematic and principled theoretical understanding of exposure bias in KD has remained largely absent. We fill this gap by introducing the Bridge–Garden framework (Theorem 4.1) to characterize how exposure bias arises. Grounded in this theory, we develop a family of principled distillation methods for reducing exposure bias effectively. Moreover, a key advantage of our approach is its efficiency: it requires no expensive sampling from the student model during training. This makes it particularly suitable for large-scale industrial applications.

## B. A $\kappa$-Weighted Bound on Exposure Bias (Proof of Thm. 4.1)

This section proves Thm.4.1, showing that the exposure bias $\mathsf{EB}(\pi_\theta)$ is upper bounded by a $d_T$-expectation of a pointwise $\kappa$-weighted deviation term plus a quadratic term.

**Setup.** We analyze length-$T$ autoregressive generation (variable-length can be handled by standard padding and masking). For any policy $\pi$, let $\xi = (s_1, a_1, \dots, s_T, a_T)$ denote the induced random autoregressive trajectory, where $s_t = (x, y_{<t})$ and $a_t \in \mathcal{V}$, and write $\mathbb{E}_\pi[\cdot]$ for expectation with respect to this randomness. Let $d_\pi$ denote the induced prefix distribution, *i.e.*, the distribution of $s_t$ when $t$ is chosen uniformly from $\{1, \dots, T\}$. We will use the identity: for any measurable $f$,

$$\mathbb{E}_{s \sim d_\pi}[f(s)] = \frac{1}{T} \sum_{t=1}^{T} \mathbb{E}_\pi[f(s_t)]. \tag{6}$$

Recall

$$\ell_{\text{soft}}(s; \theta) := \mathbb{D}\big(\pi_T(\cdot \mid s) \,\|\, \pi_\theta(\cdot \mid s)\big), \qquad \mathcal{L}_d(\pi_\theta) := \mathbb{E}_{s \sim d}\big[\ell_{\text{soft}}(s; \theta)\big],$$

and the exposure bias (Def.3.1)

$$\mathsf{EB}(\pi_\theta) = \mathcal{L}_{d_\theta}(\pi_\theta) - \mathcal{L}_{d_T}(\pi_\theta).$$

We use the following standard conditions (only to control the quadratic term):

1. (*bounded loss*) $0 \leq \ell_{\text{soft}}(s; \theta) \leq L_{\max}$ for all relevant prefixes $s$;

2. (*minimum probability along the interpolation*) for $\pi_u := (1 - u)\pi_T + u\pi_\theta$ and all $u \in [0, 1]$, $\pi_u(a \mid s) \geq \beta$ for all $a \in \mathcal{V}$ and relevant prefixes $s$;

3. (*prefix concentrability*) there exists $C_{\text{conc}} \geq 1$ such that $d_{\pi_u}(s) \leq C_{\text{conc}} d_T(s)$ for all $u \in [0, 1]$ and prefixes $s$.

The first two conditions are regularity conditions for finite-vocabulary softmax models: bounded logits give finite losses and nonzero token probabilities on the prefixes under consideration. The third condition is the usual overlap requirement used when an off-policy distribution is used to reason about another rollout distribution. Here it requires the teacher-prefix distribution to cover the prefixes reached along the teacher–student interpolation. This is the relevant regime for distillation, where teacher and student share the tokenizer and are trained to keep the student close to the teacher.

**Restate of Theorem 4.1** *Under mild conditions (bounded loss and non-vanishing probabilities), let $\Delta\pi_\theta := \pi_\theta - \pi_T$. Then the exposure bias* $\mathsf{EB}(\pi_\theta)$ *satisfies*

$$\mathsf{EB}(\pi_\theta) \ \leq \ \mathbb{E}_{s \sim d_T}\Big[ F(s, \pi_\theta) \Big],$$

*where for a constant $C_2 > 0$,*

$$F(s, \pi_\theta) := \underbrace{\sum_a \kappa(a \,|\, s) \cdot |\Delta\pi_\theta(a \,|\, s)|}_{\kappa(a|s)\text{-}\textbf{weighted deviation}} \ + C_2 \|\Delta\pi_\theta(\cdot \,|\, s)\|_1^2.$$

*Proof.* The proof proceeds in several steps:

**Step 1: A one-dimensional path identity.** For $u \in [0, 1]$, define the linear interpolation

$$\pi_u := (1 - u)\pi_T + u\pi_\theta, \qquad g(u) := \mathsf{EB}(\pi_u) = \mathcal{L}_{d_{\pi_u}}(\pi_\theta) - \mathcal{L}_{d_T}(\pi_\theta).$$

By construction, $g(0) = \mathsf{EB}(\pi_T) = 0$ and $g(1) = \mathsf{EB}(\pi_\theta)$. Under the above regularity conditions, $g$ is twice differentiable on $[0, 1]$. Applying the fundamental theorem of calculus twice yields

$$g(1) = g(0) + \int_0^1 g'(u) \, du = g(0) + g'(0) + \int_0^1 \int_0^u g''(t) \, dt \, du.$$

Since $g(0) = 0$, it remains to simplify the double integral. By swapping the order of integration,

$$\int_0^1 \int_0^u g''(t) \, dt \, du = \int_0^1 \left( \int_t^1 du \right) g''(t) \, dt = \int_0^1 (1 - t) \, g''(t) \, dt.$$

Renaming $t$ as $u$ gives the identity

$$\mathsf{EB}(\pi_\theta) = g'(0) + \int_0^1 (1 - u) \, g''(u) \, du, \tag{7}$$

.

**Step 2: Rewrite $g(u)$ using a trajectory functional.** Define

$$A(\xi) := \sum_{t=1}^T \ell_{\text{soft}}(s_t; \theta), \qquad J(u) := \mathbb{E}_{\pi_u}[A(\xi)].$$

By Eq.(6),

$$\mathcal{L}_{d_{\pi_u}}(\pi_\theta) = \mathbb{E}_{s \sim d_{\pi_u}}\big[\ell_{\text{soft}}(s; \theta)\big] = \frac{1}{T}\mathbb{E}_{\pi_u}\left[ \sum_{t=1}^T \ell_{\text{soft}}(s_t; \theta) \right] = \frac{1}{T} J(u),$$

and similarly $\mathcal{L}_{d_T}(\pi_\theta) = \frac{1}{T} J(0)$. Therefore,

$$g(u) = \frac{1}{T}\big(J(u) - J(0)\big), \qquad g'(u) = \frac{1}{T}J'(u), \qquad g''(u) = \frac{1}{T}J''(u). \tag{8}$$

**Step 3: Compute $g'(0)$ and express it using $\kappa(a \mid s)$.** By Eq.(8), it suffices to compute $J'(0)$.

*(3.1) Score-function derivative of $J(u)$.* Let $\Delta\pi_\theta := \pi_\theta - \pi_T$. Let $\pi_u(\xi)$ denote the probability mass of trajectory $\xi$ induced by $\pi_u$. By autoregressive factorization,

$$\log \pi_u(\xi) = \sum_{t=1}^{T} \log \pi_u(a_t \mid s_t), \qquad \frac{d}{du} \log \pi_u(\xi) = \sum_{t=1}^{T} \frac{\Delta\pi_\theta(a_t \mid s_t)}{\pi_u(a_t \mid s_t)}.$$

Since $A(\xi)$ does not depend on $u$, the score-function identity yields

$$J'(u) = \mathbb{E}_{\pi_u}\left[ A(\xi) \sum_{t=1}^{T} \frac{\Delta\pi_\theta(a_t \mid s_t)}{\pi_u(a_t \mid s_t)} \right].$$

Evaluating at $u = 0$ gives

$$J'(0) = \mathbb{E}_{\pi_T}\left[ A(\xi) \sum_{t=1}^{T} \frac{\Delta\pi_\theta(a_t \mid s_t)}{\pi_T(a_t \mid s_t)} \right], \qquad g'(0) = \frac{1}{T} J'(0). \tag{9}$$

*(3.2) Regroup by prefixes.* For each prefix $s$ and token $a \in \mathcal{V}$, define

$$Q(s,a) := \mathbb{E}_{\pi_T}\left[ A(\xi) \mid s_t = s, \, a_t = a \right], \qquad \bar{Q}(s) := \sum_{a \in \mathcal{V}} \pi_T(a \mid s) \, Q(s,a).$$

Conditioning on $s_t = s$ and using $a_t \sim \pi_T(\cdot \mid s)$, we have

$$\mathbb{E}_{\pi_T}\left[ A(\xi) \frac{\Delta\pi_\theta(a_t \mid s_t)}{\pi_T(a_t \mid s_t)} \,\middle|\, s_t = s \right] = \sum_{a \in \mathcal{V}} \Delta\pi_\theta(a \mid s) \, Q(s,a).$$

Plugging this into Eq.(9), summing over $t$, and using Eq.(6) yields

$$g'(0) = \mathbb{E}_{s \sim d_T}\left[ \sum_{a \in \mathcal{V}} \Delta\pi_\theta(a \mid s) \, Q(s,a) \right]. \tag{10}$$

Moreover, since $\sum_{a \in \mathcal{V}} \Delta\pi_\theta(a \mid s) = 0$, we can center $Q$:

$$\sum_{a \in \mathcal{V}} \Delta\pi_\theta(a \mid s) \, Q(s,a) = \sum_{a \in \mathcal{V}} \Delta\pi_\theta(a \mid s) \left( Q(s,a) - \bar{Q}(s) \right). \tag{11}$$

*(3.3) Identify $Q(s,a) - \bar{Q}(s)$ with $\kappa(a \mid s)$.* Fix $(s,a)$ with $d_T(s) > 0$ and consider the single-override policy $\pi^{(s,a)}$ (Def. 4.1). It coincides with $\pi_T$ except that it deterministically outputs $a$ at prefix $s$. Thus, conditioned on $s_t = s$, the continuation under $\pi^{(s,a)}$ matches the teacher continuation conditioned on $a_t = a$, so

$$\mathbb{E}_{\pi^{(s,a)}}[A(\xi) \mid s_t = s] = Q(s,a), \qquad \mathbb{E}_{\pi_T}[A(\xi) \mid s_t = s] = \bar{Q}(s).$$

The visitation frequency of prefix $s$ is unchanged by the override before reaching $s$, hence the difference in total loss satisfies

$$\mathbb{E}_{\pi^{(s,a)}}[A(\xi)] - \mathbb{E}_{\pi_T}[A(\xi)] = T \, d_T(s) \left( Q(s,a) - \bar{Q}(s) \right).$$

Dividing by $T$ and using $\mathsf{EB}(\pi) = \mathcal{L}_{d_\pi}(\pi_\theta) - \mathcal{L}_{d_T}(\pi_\theta) = \frac{1}{T}\left( \mathbb{E}_\pi[A(\xi)] - \mathbb{E}_{\pi_T}[A(\xi)] \right)$ gives

$$\mathsf{EB}(\pi^{(s,a)}) = d_T(s) \left( Q(s,a) - \bar{Q}(s) \right).$$

Therefore, by Def. 4.1,

$$\kappa(a \mid s) = \frac{\mathsf{EB}(\pi^{(s,a)})}{d_T(s)} = Q(s,a) - \bar{Q}(s).$$

Substituting into Eq.(10)–(11) yields

$$g'(0) = \mathbb{E}_{s \sim d_T}\left[ \sum_{a \in \mathcal{V}} \kappa(a \mid s) \, \Delta\pi_\theta(a \mid s) \right]. \tag{12}$$

**Step 4: Upper bound the first-order term.** Recall that we analyze the typical regime where student deviations from the teacher exacerbate exposure bias, so $\mathsf{EB}(\pi^{(s,a)}) \geq 0$ and the associated sensitivity weight satisfies $\kappa(a \mid s) \geq 0$. Using the triangle inequality, we obtain (12),

$$g'(0) \leq \mathbb{E}_{s \sim d_T}\Big[\sum_{a \in \mathcal{V}} \kappa(a \mid s)\,|\Delta\pi_\theta(a \mid s)|\Big]. \tag{13}$$

**Step 5: Bound the remainder by a quadratic term.** From Eq.(8) and (3.1), define

$$H_u(\xi) := \sum_{t=1}^{T} \frac{\Delta\pi_\theta(a_t \mid s_t)}{\pi_u(a_t \mid s_t)}, \qquad \text{so that} \qquad J'(u) = \mathbb{E}_{\pi_u}\big[A(\xi)\,H_u(\xi)\big].$$

Differentiating once more gives

$$J''(u) = \mathbb{E}_{\pi_u}\Big[A(\xi)\big(H_u(\xi)^2 + \partial_u H_u(\xi)\big)\Big].$$

Since $\partial_u\Big(\frac{\Delta\pi_\theta(a_t \mid s_t)}{\pi_u(a_t \mid s_t)}\Big) = -\Big(\frac{\Delta\pi_\theta(a_t \mid s_t)}{\pi_u(a_t \mid s_t)}\Big)^2$, we obtain

$$H_u(\xi)^2 + \partial_u H_u(\xi) = \sum_{t_1 \neq t_2} \frac{\Delta\pi_\theta(a_{t_1} \mid s_{t_1})}{\pi_u(a_{t_1} \mid s_{t_1})} \cdot \frac{\Delta\pi_\theta(a_{t_2} \mid s_{t_2})}{\pi_u(a_{t_2} \mid s_{t_2})}.$$

Therefore, using $g''(u) = \frac{1}{T}J''(u)$,

$$g''(u) = \frac{1}{T}\mathbb{E}_{\pi_u}\Big[A(\xi) \sum_{t_1 \neq t_2} \frac{\Delta\pi_\theta(a_{t_1} \mid s_{t_1})}{\pi_u(a_{t_1} \mid s_{t_1})} \cdot \frac{\Delta\pi_\theta(a_{t_2} \mid s_{t_2})}{\pi_u(a_{t_2} \mid s_{t_2})}\Big]. \tag{14}$$

Let $r(s) := \|\Delta\pi_\theta(\cdot \mid s)\|_1$. By Assumption (ii), $\pi_u(a \mid s) \geq \beta$, hence

$$\Big|\frac{\Delta\pi_\theta(a_t \mid s_t)}{\pi_u(a_t \mid s_t)}\Big| \leq \frac{|\Delta\pi_\theta(a_t \mid s_t)|}{\beta} \leq \frac{r(s_t)}{\beta}.$$

Thus

$$\Big|\sum_{t_1 \neq t_2} \frac{\Delta\pi_\theta(a_{t_1} \mid s_{t_1})}{\pi_u(a_{t_1} \mid s_{t_1})} \cdot \frac{\Delta\pi_\theta(a_{t_2} \mid s_{t_2})}{\pi_u(a_{t_2} \mid s_{t_2})}\Big| \leq \frac{1}{\beta^2}\sum_{t_1 \neq t_2} r(s_{t_1})r(s_{t_2}) \leq \frac{T-1}{\beta^2}\sum_{t=1}^{T} r(s_t)^2,$$

where we used $\sum_{i \neq j} b_i b_j \leq (T-1)\sum_i b_i^2$ for $b_i \geq 0$. Also $0 \leq A(\xi) \leq T L_{\max}$ by Assumption (i). Plugging these into Eq.(14) yields

$$|g''(u)| \leq \frac{1}{T} \cdot (T L_{\max}) \cdot \frac{T-1}{\beta^2}\mathbb{E}_{\pi_u}\Big[\sum_{t=1}^{T} r(s_t)^2\Big] = \frac{L_{\max}}{\beta^2} T(T-1)\,\mathbb{E}_{s \sim d_{\pi_u}}\big[r(s)^2\big],$$

where the last equality follows from Eq.(6). By Assumption (iii), $\mathbb{E}_{s \sim d_{\pi_u}}[r(s)^2] \leq C_{\mathrm{conc}}\mathbb{E}_{s \sim d_T}[r(s)^2]$. Therefore,

$$\sup_{u \in [0,1]} |g''(u)| \leq \frac{L_{\max}}{\beta^2} C_{\mathrm{conc}} T(T-1)\,\mathbb{E}_{s \sim d_T}\big[\|\Delta\pi_\theta(\cdot \mid s)\|_1^2\big].$$

Finally, by Eq.(7),

$$\Big|\int_0^1 (1-u)g''(u)\,du\Big| \leq \frac{1}{2}\sup_{u \in [0,1]} |g''(u)| \leq C_2 \cdot \mathbb{E}_{s \sim d_T}\big[\|\Delta\pi_\theta(\cdot \mid s)\|_1^2\big],$$

with

$$C_2 := \frac{L_{\max}}{2\beta^2} C_{\mathrm{conc}} T(T-1). \tag{15}$$

**Step 6: Conclude.** Combining Eq.(7), Eq.(13), and Step 5 yields

$$\mathsf{EB}(\pi_\theta) \leq \mathbb{E}_{s \sim d_T}\Big[\sum_{a \in \mathcal{V}} \kappa(a \mid s)|\Delta\pi_\theta(a \mid s)| \;+\; C_2 \,\|\Delta\pi_\theta(\cdot \mid s)\|_1^2\Big].$$

Defining

$$F(s, \pi_\theta) := \sum_{a \in \mathcal{V}} \kappa(a \mid s)|\Delta\pi_\theta(a \mid s)| \;+\; C_2 \,\|\Delta\pi_\theta(\cdot \mid s)\|_1^2,$$

we obtain $\mathsf{EB}(\pi_\theta) \leq \mathbb{E}_{s \sim d_T}[F(s, \pi_\theta)]$, which is exactly Thm. 4.1. □

## C. The Bridge–Garden Upper Bound on Exposure Bias (Proof of Prop. 4.1)

**Restate of Proposition 4.1.** *Using the partition from Def.4.2, the bound from Thm.4.1 decomposes as follows:*

$$F(\pi_\theta) := F_{\mathcal{B}}(\pi_\theta) + F_{\mathcal{G}}(\pi_\theta),$$

*where $F_{\mathcal{X}}(\pi_\theta) := \mathbb{E}_{s \sim d_T}[\mathbf{1}_{\mathcal{X}}(s)F(s, \pi_\theta)]$ for $\mathcal{X} \in \{\mathcal{B}, \mathcal{G}\}$, and the term $F(s, \pi_\theta)$ is defined in Thm.4.1.*

*Proof.* By Def. 4.2, $\mathcal{B}$ and $\mathcal{G}$ form a partition of the prefix space $\mathcal{S}$. Hence, for every prefix $s$,

$$\mathbf{1}_{\mathcal{B}}(s) + \mathbf{1}_{\mathcal{G}}(s) = 1. \tag{16}$$

Thm. 4.1 gives

$$\mathsf{EB}(\pi_\theta) \;\leq\; \mathbb{E}_{s \sim d_T}\big[F(s, \pi_\theta)\big]. \tag{17}$$

Insert Eq.(16) into the integrand:

$$F(s, \pi_\theta) = \big(\mathbf{1}_{\mathcal{B}}(s) + \mathbf{1}_{\mathcal{G}}(s)\big)F(s, \pi_\theta) = \mathbf{1}_{\mathcal{B}}(s)F(s, \pi_\theta) + \mathbf{1}_{\mathcal{G}}(s)F(s, \pi_\theta).$$

Taking expectation under $d_T$ and using linearity,

$$\mathbb{E}_{s \sim d_T}\big[F(s, \pi_\theta)\big] = \mathbb{E}_{s \sim d_T}\big[\mathbf{1}_{\mathcal{B}}(s)F(s, \pi_\theta)\big] + \mathbb{E}_{s \sim d_T}\big[\mathbf{1}_{\mathcal{G}}(s)F(s, \pi_\theta)\big].$$

By definition, for $\mathcal{X} \in \{\mathcal{B}, \mathcal{G}\}$,

$$F_{\mathcal{X}}(\pi_\theta) \;:=\; \mathbb{E}_{s \sim d_T}\big[\mathbf{1}_{\mathcal{X}}(s)F(s, \pi_\theta)\big],$$

so

$$\mathbb{E}_{s \sim d_T}\big[F(s, \pi_\theta)\big] = F_{\mathcal{B}}(\pi_\theta) + F_{\mathcal{G}}(\pi_\theta).$$

Let $F(\pi_\theta) := \mathbb{E}_{s \sim d_T}\big[F(s, \pi_\theta)\big]$. Substituting into Eq.(17) yields

$$\mathsf{EB}(\pi_\theta) \;\leq\; F(\pi_\theta) = F_{\mathcal{B}}(\pi_\theta) + F_{\mathcal{G}}(\pi_\theta),$$

which proves the proposition. □

## D. Complementarity Gain (Proof of Thm.4.2)

**Restate of Theorem 4.2.** *Let $\pi_{hard}$ and $\pi_{soft}$ be (near-)minimizers of $\ell_{hard}$ and $\ell_{soft}$, respectively; in late-stage distillation, often,*

$$F_{\mathcal{G}}(\pi_{soft}) \;<\; F_{\mathcal{G}}(\pi_{hard}), \quad F_{\mathcal{B}}(\pi_{hard}) \;<\; F_{\mathcal{B}}(\pi_{soft}).$$

*This suggests that for some $\lambda \in (0, 1)$, an optimizer $\pi_{hyb} \in \arg\min\big(\lambda\ell_{soft} + (1-\lambda)\ell_{hard}\big)$ can achieve*

$$F(\pi_{hyb}) \;<\; \min\{F(\pi_{hard}), F(\pi_{soft})\}.$$

### D.1. Late-KD Region-wise Residual Profiles and Complementarity

This section first establishes the region-wise complementarity relations

$$F_{\mathcal{G}}(\pi_{\text{soft}}) \;<\; F_{\mathcal{G}}(\pi_{\text{hard}}), \qquad F_{\mathcal{B}}(\pi_{\text{hard}}) \;<\; F_{\mathcal{B}}(\pi_{\text{soft}}),$$

under verifiable *late-stage* KD structural conditions (Corollary D.2). Finally, Appendix D.2 shows how these region-wise inequalities yield a strict improvement guarantee for the Hybrid update.

D.1.1. AUXILIARY QUANTITIES (CF. THM. 4.1)

Recall the Bridge–Garden partition $(\mathcal{B}, \mathcal{G})$ and indicators $1_{\mathcal{B}}, 1_{\mathcal{G}}$ from Def. 4.2. For $\mathcal{X} \in \{\mathcal{B}, \mathcal{G}\}$, the region-wise bound is

$$F_{\mathcal{X}}(\pi) := \mathbb{E}_{s \sim d_T}\left[1_{\mathcal{X}}(s)\, F(s, \pi)\right],$$

where $F(s, \pi)$ and $\Delta_{\pi}(\cdot \mid s)$ are as in Thm. 4.1.

We denote the region masses

$$p_{\mathcal{B}} := \mathbb{E}[1_{\mathcal{B}}(s)], \qquad p_{\mathcal{G}} := \mathbb{E}[1_{\mathcal{G}}(s)], \tag{18}$$

and assume $p_{\mathcal{B}} > 0$ and $p_{\mathcal{G}} > 0$.

For later bounds, we use the decomposition

$$K_{\mathcal{X}}(\pi) := \mathbb{E}\left[1_{\mathcal{X}}(s) \sum_{a \in \mathcal{V}} \kappa(a \mid s)\, |\Delta_{\pi}(a \mid s)|\right], \tag{19}$$

$$\delta_{2,\mathcal{X}}(\pi)^2 := \mathbb{E}\left[1_{\mathcal{X}}(s)\, \|\Delta_{\pi}(\cdot \mid s)\|_1^2\right], \tag{20}$$

so that

$$F_{\mathcal{X}}(\pi) = K_{\mathcal{X}}(\pi) + C_2\, \delta_{2,\mathcal{X}}(\pi)^2. \tag{21}$$

We will also use the basic inequality for distributions:

$$0 \leq \|\Delta_{\pi}(\cdot \mid s)\|_1 \leq 2 \implies \|\Delta_{\pi}(\cdot \mid s)\|_1^2 \leq 2\, \|\Delta_{\pi}(\cdot \mid s)\|_1. \tag{22}$$

Finally, define the teacher top token

$$b(s) := \arg\max_{a \in \mathcal{V}} \pi_T(a \mid s). \tag{23}$$

D.1.2. LATE-KD STRUCTURAL CONDITIONS

**(D1) Bridges: teacher nearly deterministic.** Define the teacher tail mass

$$\eta(s) := 1 - \pi_T(b(s) \mid s) = \sum_{a \neq b(s)} \pi_T(a \mid s). \tag{24}$$

Assume that there exists $\varepsilon_{\mathcal{B}} \in (0, 1/2)$ such that

$$\eta(s) \leq \varepsilon_{\mathcal{B}}, \qquad \forall s \text{ with } 1_{\mathcal{B}}(s) = 1. \tag{25}$$

**(D2) Gardens: teacher far from any one-hot.** Assume there exists $\eta_{\mathcal{G}} \in (0, 1/2]$ such that for all $s$ with $1_{\mathcal{G}}(s) = 1$,

$$\max_{a \in \mathcal{V}} \pi_T(a \mid s) \leq 1 - \eta_{\mathcal{G}} \iff \|\delta_a - \pi_T(\cdot \mid s)\|_1 \geq 2\eta_{\mathcal{G}} \ \forall a \in \mathcal{V}. \tag{26}$$

**(D3) Optimization effectiveness (late KD).** Assume there exist $\eta_H, \eta_S > 0$ such that

$$\mathbb{E}[\|\pi_{\text{hard}}(\cdot \mid s) - \delta_{a^*}\|_1] \leq \eta_H, \tag{27}$$

$$\mathbb{E}[\|\pi_{\text{soft}}(\cdot \mid s) - \pi_T(\cdot \mid s)\|_1] \leq \eta_S. \tag{28}$$

**(D4) Soft is under-confident on Bridges on average.** Define

$$g(s) := \left[\pi_T(a^* \mid s) - \pi_{\text{soft}}(a^* \mid s)\right]_+, \tag{29}$$

where $a^*$ is the hard-KD training target sampled from the teacher model, *i.e.*, $a^* \sim \pi_T(\cdot \mid s)$. Assume

$$\mathbb{E}[1_{\mathcal{B}}(s)\, g(s)] \geq \Gamma_{\mathcal{B}}, \qquad \Gamma_{\mathcal{B}} > 0. \tag{30}$$

D.1.3. REGION-WISE BOUNDS AND COMPLEMENTARITY

**Theorem D.1** (Late-KD region-wise bounds for $F_{\mathcal{B}}$ and $F_{\mathcal{G}}$). *Under* (D1)–(D4)*, define*

$$A_{\mathcal{B}} := \eta_H + 4\,\mathbb{E}[1_{\mathcal{B}}(s)\,\eta(s)], \qquad \alpha_{\mathcal{G}} := \big(2\eta_{\mathcal{G}}\,p_{\mathcal{G}} - \eta_H\big)_+. \tag{31}$$

*Then:*

$$F_{\mathcal{B}}(\pi_{\text{hard}}) \leq (\kappa_{\max} + 2C_2)\,A_{\mathcal{B}} \;\leq\; (\kappa_{\max} + 2C_2)\big(\eta_H + 4p_{\mathcal{B}}\varepsilon_{\mathcal{B}}\big), \tag{32}$$

$$F_{\mathcal{G}}(\pi_{\text{soft}}) \leq (\kappa_{\max} + 2C_2)\,\eta_S, \tag{33}$$

$$F_{\mathcal{G}}(\pi_{\text{hard}}) \geq \frac{C_2}{p_{\mathcal{G}}}\,\alpha_{\mathcal{G}}^2, \tag{34}$$

$$F_{\mathcal{B}}(\pi_{\text{soft}}) \geq \frac{4C_2}{p_{\mathcal{B}}}\,\Gamma_{\mathcal{B}}^2. \tag{35}$$

*Proof.* We prove the four inequalities in turn.

**(1) Upper bound on $F_{\mathcal{B}}(\pi_{\text{hard}})$.** Fix $s$ with $1_{\mathcal{B}}(s) = 1$. By the triangle inequality,

$$\|\pi_{\text{hard}}(\cdot \mid s) - \pi_T(\cdot \mid s)\|_1 \leq \|\pi_{\text{hard}}(\cdot \mid s) - \delta_{a^*}\|_1 + \|\delta_{a^*} - \pi_T(\cdot \mid s)\|_1.$$

For the second term, conditioning on $s$ and using $a^* \sim \pi_T(\cdot \mid s)$,

$$\mathbb{E}[\|\delta_{a^*} - \pi_T(\cdot \mid s)\|_1 \mid s] = 2\Big(1 - \sum_{a \in \mathcal{V}} \pi_T(a \mid s)^2\Big) \leq 2\big(1 - \pi_T(b(s) \mid s)^2\big) = 4\eta(s) - 2\eta(s)^2 \leq 4\eta(s),$$

where we used Eq.(24) in the last two steps. Multiplying by $1_{\mathcal{B}}(s)$ and taking expectation gives

$$\mathbb{E}[1_{\mathcal{B}}(s)\,\|\pi_{\text{hard}}(\cdot \mid s) - \pi_T(\cdot \mid s)\|_1] \leq \mathbb{E}[\|\pi_{\text{hard}}(\cdot \mid s) - \delta_{a^*}\|_1] + 4\,\mathbb{E}[1_{\mathcal{B}}(s)\eta(s)] \leq A_{\mathcal{B}},$$

using Eq.(27) and Eq.(31),

$$K_{\mathcal{B}}(\pi_{\text{hard}}) \leq \kappa_{\max}\,\mathbb{E}[1_{\mathcal{B}}(s)\,\|\Delta_{\pi_{\text{hard}}}(\cdot \mid s)\|_1].$$

By Eq.(22),

$$\delta_{2,\mathcal{B}}(\pi_{\text{hard}})^2 = \mathbb{E}\big[1_{\mathcal{B}}(s)\,\|\Delta_{\pi_{\text{hard}}}(\cdot \mid s)\|_1^2\big] \leq 2\,\mathbb{E}[1_{\mathcal{B}}(s)\,\|\Delta_{\pi_{\text{hard}}}(\cdot \mid s)\|_1].$$

Combining with Eq.(21) yields

$$F_{\mathcal{B}}(\pi_{\text{hard}}) \leq (\kappa_{\max} + 2C_2)\,\mathbb{E}[1_{\mathcal{B}}(s)\,\|\Delta_{\pi_{\text{hard}}}(\cdot \mid s)\|_1] \leq (\kappa_{\max} + 2C_2)\,A_{\mathcal{B}},$$

which proves the first inequality in Eq.(32). The second inequality follows from (25), which implies $\mathbb{E}[1_{\mathcal{B}}(s)\eta(s)] \leq p_{\mathcal{B}}\varepsilon_{\mathcal{B}}$.

**(2) Upper bound on $F_{\mathcal{G}}(\pi_{\text{soft}})$.** By Eq.(28),

$$\mathbb{E}[1_{\mathcal{G}}(s)\,\|\Delta_{\pi_{\text{soft}}}(\cdot \mid s)\|_1] \leq \mathbb{E}[\|\pi_{\text{soft}}(\cdot \mid s) - \pi_T(\cdot \mid s)\|_1] \leq \eta_S.$$

Thus, using Eq.(22) exactly as above,

$$K_{\mathcal{G}}(\pi_{\text{soft}}) \leq \kappa_{\max}\eta_S, \qquad \delta_{2,\mathcal{G}}(\pi_{\text{soft}})^2 \leq 2\eta_S,$$

and Eq.(33) follows from Eq.(21).

**(3) Lower bound on $F_{\mathcal{G}}(\pi_{\text{hard}})$.** Fix $s$ with $1_{\mathcal{G}}(s) = 1$. By Eq.(26), $\|\delta_{a^*} - \pi_T(\cdot \mid s)\|_1 \geq 2\eta_{\mathcal{G}}$. By the reverse triangle inequality,

$$\|\Delta_{\pi_{\text{hard}}}(\cdot \mid s)\|_1 = \|\pi_{\text{hard}}(\cdot \mid s) - \pi_T(\cdot \mid s)\|_1 \geq \|\delta_{a^*} - \pi_T(\cdot \mid s)\|_1 - \|\pi_{\text{hard}}(\cdot \mid s) - \delta_{a^*}\|_1 \geq 2\eta_{\mathcal{G}} - \|\pi_{\text{hard}}(\cdot \mid s) - \delta_{a^*}\|_1.$$

Multiplying by $1_{\mathcal{G}}(s)$ and taking expectation yields

$$\mathbb{E}[1_{\mathcal{G}}(s)\,\|\Delta_{\pi_{\text{hard}}}(\cdot \mid s)\|_1] \geq 2\eta_{\mathcal{G}}p_{\mathcal{G}} - \mathbb{E}[\|\pi_{\text{hard}}(\cdot \mid s) - \delta_{a^*}\|_1] \geq 2\eta_{\mathcal{G}}p_{\mathcal{G}} - \eta_H,$$

hence $\mathbb{E}[1_\mathcal{G}\|\Delta_{\pi_{\text{hard}}}\|_1] \geq \alpha_\mathcal{G}$ by Eq.(31). Now apply Cauchy–Schwarz to $X := 1_\mathcal{G}(s)\|\Delta_{\pi_{\text{hard}}}(\cdot \mid s)\|_1 \geq 0$:

$$\mathbb{E}[X]^2 \leq \mathbb{E}[1_\mathcal{G}(s)] \cdot \mathbb{E}\big[1_\mathcal{G}(s)\|\Delta_{\pi_{\text{hard}}}(\cdot \mid s)\|_1^2\big] = p_\mathcal{G}\,\delta_{2,\mathcal{G}}(\pi_{\text{hard}})^2.$$

Therefore $\delta_{2,\mathcal{G}}(\pi_{\text{hard}})^2 \geq \alpha_\mathcal{G}^2/p_\mathcal{G}$. Finally, since $K_\mathcal{G}(\pi_{\text{hard}}) \geq 0$, Eq.(21) gives

$$F_\mathcal{G}(\pi_{\text{hard}}) \geq C_2\,\delta_{2,\mathcal{G}}(\pi_{\text{hard}})^2 \geq \frac{C_2}{p_\mathcal{G}}\,\alpha_\mathcal{G}^2,$$

which is Eq.(34).

**(4) Lower bound on $F_\mathcal{B}(\pi_{\text{soft}})$.** Fix $s$ with $1_\mathcal{B}(s) = 1$. If $g(s) > 0$, then $\Delta_{\pi_{\text{soft}}}(a^* \mid s) = \pi_{\text{soft}}(a^* \mid s) - \pi_T(a^* \mid s) = -g(s)$. Using $\sum_{a\in\mathcal{V}}\Delta_{\pi_{\text{soft}}}(a \mid s) = 0$, we have $\sum_{a\neq a^*}\Delta_{\pi_{\text{soft}}}(a \mid s) = g(s)$ and thus $\sum_{a\neq a^*}|\Delta_{\pi_{\text{soft}}}(a \mid s)| \geq g(s)$. Hence

$$\|\Delta_{\pi_{\text{soft}}}(\cdot \mid s)\|_1 = |\Delta_{\pi_{\text{soft}}}(a^* \mid s)| + \sum_{a\neq a^*}|\Delta_{\pi_{\text{soft}}}(a \mid s)| \geq g(s) + g(s) = 2g(s),$$

and the same inequality holds trivially if $g(s) = 0$. Consequently,

$$\delta_{2,\mathcal{B}}(\pi_{\text{soft}})^2 = \mathbb{E}\big[1_\mathcal{B}(s)\|\Delta_{\pi_{\text{soft}}}(\cdot \mid s)\|_1^2\big] \geq 4\,\mathbb{E}\big[1_\mathcal{B}(s)\,g(s)^2\big].$$

By Cauchy–Schwarz, $\mathbb{E}[1_\mathcal{B}g]^2 \leq \mathbb{E}[1_\mathcal{B}]\,\mathbb{E}[1_\mathcal{B}g^2] = p_\mathcal{B}\mathbb{E}[1_\mathcal{B}g^2]$, hence

$$\mathbb{E}[1_\mathcal{B}g^2] \geq \frac{\mathbb{E}[1_\mathcal{B}g]^2}{p_\mathcal{B}} \geq \frac{\Gamma_\mathcal{B}^2}{p_\mathcal{B}},$$

by Eq.(30). Therefore

$$\delta_{2,\mathcal{B}}(\pi_{\text{soft}})^2 \geq \frac{4}{p_\mathcal{B}}\,\Gamma_\mathcal{B}^2.$$

Since $K_\mathcal{B}(\pi_{\text{soft}}) \geq 0$, Eq.(21) implies

$$F_\mathcal{B}(\pi_{\text{soft}}) \geq C_2\,\delta_{2,\mathcal{B}}(\pi_{\text{soft}})^2 \geq \frac{4C_2}{p_\mathcal{B}}\,\Gamma_\mathcal{B}^2,$$

which is Eq.(35). $\qquad\square$

**Remark.** Thm. D.1 provides *complementary* upper/lower bounds on the region-wise objectives $F_\mathcal{B}$ and $F_\mathcal{G}$ for $\pi_{\text{hard}}$ and $\pi_{\text{soft}}$. To turn these bounds into a *strict* region-wise comparison, one needs a quantitative regime in which the late-stage optimization residuals are dominated by the region-structural terms.

Specifically, in late KD one typically observes that the Soft-only fit-to-teacher error $\eta_S$ becomes small, and the Hard-only Bridge mismatch term
$$A_\mathcal{B} = \eta_H + 4\,\mathbb{E}_{s\sim d_T}\big[1_\mathcal{B}(s)\eta(s)\big]$$
also becomes small as the hard objective converges and the teacher is nearly deterministic on Bridges. At the same time, the region structure does not vanish: Gardens remain far from one-hot (captured by $\alpha_\mathcal{G}$), and Soft-only retains a nontrivial average under-confidence on Bridges (captured by $\Gamma_\mathcal{B}$). Empirically, we also observe a consistent contrast between the two endpoints: the Hard-only solution $\pi_{\text{hard}}$ tends to produce more peaked output distributions, whereas the Soft-only solution $\pi_{\text{soft}}$ tends to produce smoother ones; see Fig.6.

In this domination regime, the strict ordering follows whenever the residual upper bounds fall below the structural lower bounds, namely,

$$\underbrace{(\kappa_{\max} + 2C_2)\,\eta_S}_{\text{Soft upper bound on } F_\mathcal{G}(\pi_{\text{soft}})} < \underbrace{\frac{C_2}{p_\mathcal{G}}\,\alpha_\mathcal{G}^2}_{\text{Hard lower bound on } F_\mathcal{G}(\pi_{\text{hard}})}, \qquad \underbrace{(\kappa_{\max} + 2C_2)\,A_\mathcal{B}}_{\text{Hard upper bound on } F_\mathcal{B}(\pi_{\text{hard}})} < \underbrace{\frac{4C_2}{p_\mathcal{B}}\,\Gamma_\mathcal{B}^2}_{\text{Soft lower bound on } F_\mathcal{B}(\pi_{\text{soft}})}. \tag{36}$$

**Corollary D.2** (Region-wise complementarity inequalities). *Under the conditions of Theorem D.1 and the domination regime Eq.(36),*
$$F_\mathcal{G}(\pi_{\text{soft}}) < F_\mathcal{G}(\pi_{\text{hard}}), \qquad F_\mathcal{B}(\pi_{\text{hard}}) < F_\mathcal{B}(\pi_{\text{soft}}).$$

*Proof.* Using Eq.(33) and Eq.(34),

$$F_{\mathcal{G}}(\pi_{\text{soft}}) \leq (\kappa_{\max} + 2C_2)\,\eta_S \ < \ \frac{C_2}{p_{\mathcal{G}}}\,\alpha_{\mathcal{G}}^2 \ \leq \ F_{\mathcal{G}}(\pi_{\text{hard}}),$$

where the strict inequality is the first condition in Eq.(36). Using Eq.(32) and Eq.(35),

$$F_{\mathcal{B}}(\pi_{\text{hard}}) \leq (\kappa_{\max} + 2C_2)\,A_{\mathcal{B}} \ < \ \frac{4C_2}{p_{\mathcal{B}}}\,\Gamma_{\mathcal{B}}^2 \ \leq \ F_{\mathcal{B}}(\pi_{\text{soft}}),$$

where the strict inequality is the second condition in Eq.(36). □

### D.2. Cross-direction Descent and Hybrid Improvement

This subsection proves the main-text claim (Theorem 4.2) that a one-step hybrid update can strictly reduce the bound $F$ compared with both pure KD solutions, $\pi_{\text{soft}}$ (Soft-only) and $\pi_{\text{hard}}$ (Hard-only). The idea is to show that $F$ has a strictly negative one-sided directional derivative along the cross direction at each of these two solutions, and then pick a convex combination of the Soft/Hard directions with a small step size to guarantee a strict decrease (**see Theorem D.11**).

D.2.1. SETUP AND STRUCTURAL ASSUMPTIONS

**Recall (exposure-bias bound).** From App.D.1, we use the exposure-bias bound

$$F(\pi) = F_{\mathcal{B}}(\pi) + F_{\mathcal{G}}(\pi), \qquad F_X(\pi) = K_X(\pi) + C_2\,\delta_{2,X}(\pi)^2, \quad X \in \{\mathcal{B}, \mathcal{G}\}.$$

**Feasible Soft/Hard interpolations.** For any policy $\pi$ and $t \in [0,1]$, define

$$(\mathsf{T}_S^t \pi)(\cdot \mid s) := (1-t)\pi(\cdot \mid s) + t\,\pi_T(\cdot \mid s), \tag{37}$$

$$(\mathsf{T}_H^t \pi)(\cdot \mid s) := (1-t)\pi(\cdot \mid s) + t\,\delta_{a^*}(\cdot), \tag{38}$$

where $a^* \sim \pi_T(\cdot \mid s)$ is the hard label token in Eq. (3). The corresponding feasible directions are

$$v_S(\pi) := \pi_T - \pi, \qquad v_H(\pi) := \delta_{a^*} - \pi. \tag{39}$$

**Right directional derivative.** For any functional $J[\pi]$ and any feasible operator $\mathsf{T}^t$,

$$D_{\mathsf{T}}^+ J[\pi] := \frac{d}{dt} J[\mathsf{T}^t \pi]\bigg|_{t=0^+}. \tag{40}$$

**Endpoint asymmetry assumptions.** These conditions formalize complementarity at $\pi_{\text{soft}}$ (Soft-only) and $\pi_{\text{hard}}$ (Hard-only). Empirically, we also find these endpoint asymmetries to be consistently observable across models and datasets: the Soft-only solution typically yields a smoother, less confident (higher-entropy) distribution, whereas the Hard-only solution yields a sharper (lower-entropy) distribution and can underfit the teacher's distributional diversity in the Garden region (see Sec.6).

**(E1) Oversmoothing at the Soft-only endpoint.** Assume $g(s) > 0$ almost surely, where $g(s)$ is defined in Eq.(29).

**(E2) Nondegenerate mean gap.**

$$\mathbb{E}_{s \sim d_T}[g(s)] \geq \Gamma_{\text{OS}} > 0. \tag{41}$$

**(E3) $\kappa$-shift at $\pi_{\text{soft}}$.** Under (E1)–(E2), $\pi_{\text{soft}}$ is oversmoothed and therefore assigns insufficient probability to the teacher-sampled token $a^*$, with the remaining mass necessarily redistributed over $a \neq a^*$. Assumption (E3) formalizes that, in $\kappa$-weighted terms, this mismatch is primarily attributable to tokens that $\pi_{\text{soft}}$ *over-assigns* relative to $\pi_T$.

Let $d_a(s) := \pi_{\text{soft}}(a \mid s) - \pi_T(a \mid s)$ and define

$$\mathcal{P}(s) := \{a \neq a^* : d_a(s) > 0\}, \qquad \mathcal{N}(s) := \{a \neq a^* : d_a(s) \leq 0\}.$$

Assume that almost surely,

$$\sum_{a \in \mathcal{N}(s)} \kappa(a \mid s)\,\pi_{\text{soft}}(a \mid s) \ \leq \ \kappa(a^* \mid s)\big(1 - \pi_{\text{soft}}(a^* \mid s)\big) + \sum_{a \in \mathcal{P}(s)} \kappa(a \mid s)\,\pi_{\text{soft}}(a \mid s). \tag{42}$$

*Remark* D.3. Eq. (42) captures a typical oversmoothing effect at $\pi_{\text{soft}}$: because $\pi_{\text{soft}}$ is overly smooth, it assigns nontrivial probability mass to some non-$a^*$ tokens with large $\kappa(a \mid s)$, and in particular over-assigns them relative to the teacher (i.e., $d_a(s) > 0, a \in \mathcal{P}(s)$). As a result, the $\kappa$-weighted mismatch is mainly driven by these high-$\kappa$ over-assigned tokens.

### (E4) Hard distillation leaves residual error in the Garden region (follows directly from Theorem D.1).

$$\delta_{2,\mathcal{G}}(\pi_{\text{hard}}) \geq \underline{\delta}_{\mathcal{G}} > 0. \tag{43}$$

### D.2.2. DIRECTIONAL LEMMAS

**Lemma D.4** (Right directional derivative of $|\cdot|$). *For any $u, w \in \mathbb{R}$, $\phi(t) := |u + tw|$ has a right derivative at $t = 0$ given by*

$$\phi'_+(0) = \begin{cases} \text{sign}(u)\, w, & u \neq 0, \\ |w|, & u = 0. \end{cases}$$

**Lemma D.5** (Soft interpolation contracts $F_X$). *Fix any policy $\pi$ and $X \in \{\mathcal{B}, \mathcal{G}\}$. Let $\pi_t := \mathsf{T}_S^t \pi$. Then for all $t \in [0,1]$,*

$$K_X(\pi_t) = (1-t)K_X(\pi), \qquad \delta_{2,X}(\pi_t)^2 = (1-t)^2 \delta_{2,X}(\pi)^2,$$

*and hence*

$$D_{\mathsf{T}_S}^+ F_X(\pi) = -K_X(\pi) - 2C_2 \delta_{2,X}(\pi)^2 \leq 0. \tag{44}$$

*Proof.* Under $\pi_t = (1-t)\pi + t\pi_T$, we have $\Delta_{\pi_t} = (1-t)\Delta_\pi$ and thus $r_{\pi_t}(s) = (1-t)r_\pi(s)$. Plugging into the definitions of $K_X$ and $\delta_{2,X}$ yields the claimed scalings; differentiating at $t = 0^+$ gives Eq.(44). $\square$

**Lemma D.6** (Hard step at $\pi_{\text{soft}}$ strictly decreases $F$). *Let $\pi_t := \mathsf{T}_H^t \pi_{\text{soft}}$. Under Eq.(42) and (E1)–(E2), for each $X \in \{\mathcal{B}, \mathcal{G}\}$,*

$$\left. \frac{d}{dt} \delta_{2,X}(\pi_t)^2 \right|_{t=0^+} \leq -8\, \mathbb{E}_{s \sim d_T} \left[ \mathbf{1}_X(s)\, g(s)^2 \right], \tag{45}$$

$$\left. \frac{d}{dt} K_X(\pi_t) \right|_{t=0^+} \leq 0. \tag{46}$$

*Consequently,*

$$D_{\mathsf{T}_H}^+ F(\pi_{\text{soft}}) \leq -8C_2\, \mathbb{E}_{s \sim d_T}[g(s)^2] \leq -8C_2 \Gamma_{\text{OS}}^2 < 0. \tag{47}$$

*Proof.* Fix a prefix $s$ and condition on the sampled hard label $a^* \sim \pi_T(\cdot \mid s)$. Define $q(a) = \pi_{\text{soft}}(a \mid s)$, $p(a) = \pi_T(a \mid s)$, and $d_a = q(a) - p(a)$. By (E1) and Eq.(29), we have $d_{a^*} = -g(s) < 0$. Under the Hard interpolation $q_t = (1-t)q + t\delta_{a^*}$,

$$d_{a^*}(t) = d_{a^*} + t(1 - q(a^*)), \qquad d_a(t) = d_a - tq(a) \ (a \neq a^*).$$

**Step 1.** By Lemma D.4,

$$\left. \frac{d}{dt} |d_{a^*}(t)| \right|_{0^+} = -(1 - q(a^*)), \qquad \left. \frac{d}{dt} |d_a(t)| \right|_{0^+} = \begin{cases} -q(a), & d_a > 0, \\ +q(a), & d_a \leq 0. \end{cases}$$

Thus

$$\left. \frac{d}{dt} r_{\pi_t}(s) \right|_{0^+} = -(1 - q(a^*)) - \sum_{a \in \mathcal{P}(s)} q(a) + \sum_{a \in \mathcal{N}(s)} q(a) = -2 \sum_{a \in \mathcal{P}(s)} q(a).$$

Moreover, since $\sum_{a \neq a^*} d_a = -d_{a^*} = g(s)$ and $d_a \leq 0$ on $\mathcal{N}(s)$, we have $g(s) \leq \sum_{a \in \mathcal{P}(s)} d_a$. For $a \in \mathcal{P}(s)$, $q(a) = p(a) + d_a \geq d_a$, hence $\sum_{a \in \mathcal{P}(s)} q(a) \geq g(s)$, so

$$\left. \frac{d}{dt} r_{\pi_t}(s) \right|_{0^+} \leq -2g(s). \tag{48}$$

**Step 2.** At $t = 0$, $|d_{a^*}| = g(s)$ and $\sum_{a \neq a^*} |d_a| \geq \left| \sum_{a \neq a^*} d_a \right| = g(s)$, hence $r_{\pi_{\text{soft}}}(s) \geq 2g(s)$. Therefore,

$$\frac{d}{dt} r_{\pi_t}(s)^2 \Big|_{0+} = 2 r_{\pi_{\text{soft}}}(s) \frac{d}{dt} r_{\pi_t}(s) \Big|_{0+} \leq -8g(s)^2,$$

and multiplying by $1_X(s)$ then taking $\mathbb{E}_{s \sim d_T}$ gives Eq.(45).

**Step 3.** Let $k(t) := \sum_a \kappa(a \mid s) |d_a(t)|$. The same case split yields

$$k'_+(0) = -\kappa(a^* \mid s)(1 - q(a^*)) - \sum_{a \in \mathcal{P}(s)} \kappa(a \mid s) q(a) + \sum_{a \in \mathcal{N}(s)} \kappa(a \mid s) q(a).$$

Assumption Eq.(42) implies $k'_+(0) \leq 0$. Averaging over $s \sim d_T$ and restricting with $1_X(s)$ yields Eq.(46). Summing over $X \in \{\mathcal{B}, \mathcal{G}\}$ proves Eq.(47). $\qquad\square$

**Lemma D.7** (Cross-direction strict descent at the two pure endpoints). *Under* (E1)–(E4), *there exists $\eta_0 > 0$ such that*

$$D^+_{\mathsf{T}_S} F(\pi_{\text{hard}}) \leq -\eta_0, \qquad D^+_{\mathsf{T}_H} F(\pi_{\text{soft}}) \leq -\eta_0.$$

*In particular, one may take $\eta_0 = \min\{ 2C_2\underline{\delta}_{\mathcal{G}}^2, 8C_2\Gamma_{\text{OS}}^2 \}$.*

*Proof.* Lemma D.5 yields

$$D^+_{\mathsf{T}_S} F(\pi_{\text{hard}}) = \sum_{X \in \{\mathcal{B}, \mathcal{G}\}} \left( -K_X(\pi_{\text{hard}}) - 2C_2\delta_{2,X}(\pi_{\text{hard}})^2 \right) \leq -2C_2\delta_{2,\mathcal{G}}(\pi_{\text{hard}})^2 \leq -2C_2\underline{\delta}_{\mathcal{G}}^2,$$

using $K_X(\cdot) \geq 0$ and Eq.(43). The second inequality follows from Lemma D.6, which gives $D^+_{\mathsf{T}_H} F(\pi_{\text{soft}}) \leq -8C_2\Gamma_{\text{OS}}^2$. $\qquad\square$

**Hybrid feasible update.** For $\lambda \in [0, 1]$, define

$$q_\lambda(\cdot \mid s) = (1 - \lambda)\delta_{a^*}(\cdot) + \lambda\pi_T(\cdot \mid s), \qquad (\mathsf{T}_\lambda^t \pi)(\cdot \mid s) = (1 - t)\pi(\cdot \mid s) + t\, q_\lambda(\cdot \mid s),$$

so that $\mathsf{T}_\lambda^t \pi = \pi + t\, v_\lambda(\pi)$ with

$$v_\lambda(\pi) = (1 - \lambda)v_H(\pi) + \lambda v_S(\pi). \tag{49}$$

**Policy metric (avoids notation clash).** Define the $d_T$-weighted $\ell_1$ norm

$$\|U\|_{1,d_T} := \mathbb{E}_{s \sim d_T}\Big[\|U(\cdot \mid s)\|_1\Big] = \mathbb{E}_{s \sim d_T}\Big[\sum_{a \in \mathcal{V}} |U(a \mid s)|\Big]. \tag{50}$$

For brevity, write $\|U\| := \|U\|_{1,d_T}$.

**(S1) Approximate endpoint fixed points (in $\|\cdot\|_{1,d_T}$).**

$$\|v_H(\pi_{\text{hard}})\| \leq \varepsilon, \qquad \|v_S(\pi_{\text{soft}})\| \leq \varepsilon. \tag{51}$$

**A convexity fact: hybrid is a convex combination of the two pure updates.** Note that for every $\pi, t, \lambda$ we have the identity

$$\mathsf{T}_\lambda^t \pi = (1 - \lambda)\, \mathsf{T}_H^t \pi + \lambda\, \mathsf{T}_S^t \pi, \tag{52}$$

since both sides equal $(1 - t)\pi + t\big((1 - \lambda)\delta_{a^*} + \lambda\pi_T\big)$ pointwise in $s$.

**Lemma D.8** (Convexity of $F$ and a hybrid directional-derivative inequality). *By construction of the bound $F$ defined in Thm. 4.1 (restated above), $F$ is convex in $\pi$. Consequently, for any policy $\pi$ and any $\lambda \in [0, 1]$,*

$$D^+_{\mathsf{T}_\lambda} F(\pi) \leq (1 - \lambda) D^+_{\mathsf{T}_H} F(\pi) + \lambda D^+_{\mathsf{T}_S} F(\pi). \tag{53}$$

*Proof.* Fix any prefix $s$. The pointwise integrand

$$\pi(\cdot \mid s) \mapsto \sum_{a \in \mathcal{V}} \kappa(a \mid s) \, |\pi(a \mid s) - \pi_T(a \mid s)| \; + \; C_2 \|\pi(\cdot \mid s) - \pi_T(\cdot \mid s)\|_1^2$$

is convex because it is a nonnegative weighted sum of absolute values plus a squared norm. Taking $\mathbb{E}_{s \sim d_T}[1_X(s)\cdot]$ preserves convexity, hence each $F_X$ and thus $F = F_{\mathcal{B}} + F_{\mathcal{G}}$ is convex.

Using Eq.(52) and convexity of $F$,

$$F(\mathsf{T}_\lambda^t \pi) \leq (1 - \lambda) F(\mathsf{T}_H^t \pi) + \lambda F(\mathsf{T}_S^t \pi).$$

Subtract $F(\pi) = (1 - \lambda)F(\pi) + \lambda F(\pi)$ from both sides, divide by $t > 0$, and take $t \to 0^+$. The right directional derivatives exist by Lemma D.4 (applied inside the definitions of $K_X$ and $\delta_{2,X}$), yielding Eq.(53). $\qquad\square$

**Lemma D.9** (A Lipschitz upper bound for $F$). *Let $\kappa_{\max} := \sup_{s,a} \kappa(a \mid s) < \infty$. Then for any policies $\pi, \pi'$,*

$$|F(\pi') - F(\pi)| \; \leq \; L_{\text{Lip}} \, \|\pi' - \pi\|, \qquad L_{\text{Lip}} := \kappa_{\max} + 4C_2. \tag{54}$$

*In particular, for any feasible operator $\mathsf{T}^t \pi = \pi + t \, v(\pi)$,*

$$D_{\mathsf{T}}^+ F(\pi) \; \leq \; L_{\text{Lip}} \, \|v(\pi)\|. \tag{55}$$

*Proof.* Fix $s$ and write $\Delta(\cdot \mid s) = \pi(\cdot \mid s) - \pi_T(\cdot \mid s)$ and $\Delta'(\cdot \mid s) = \pi'(\cdot \mid s) - \pi_T(\cdot \mid s)$. For the linear term, using $||x| - |y|| \leq |x - y|$,

$$\left| \sum_a \kappa(a \mid s)|\Delta'(a \mid s)| - \sum_a \kappa(a \mid s)|\Delta(a \mid s)| \right| \leq \sum_a \kappa(a \mid s)\,|\pi'(a \mid s) - \pi(a \mid s)| \leq \kappa_{\max} \|\pi'(\cdot \mid s) - \pi(\cdot \mid s)\|_1.$$

For the quadratic term, note $\|\Delta(\cdot \mid s)\|_1 \leq 2$ and $\|\Delta'(\cdot \mid s)\|_1 \leq 2$ (difference of two distributions), hence

$$\left| \|\Delta'\|_1^2 - \|\Delta\|_1^2 \right| = (\|\Delta'\|_1 + \|\Delta\|_1) \, \big| \|\Delta'\|_1 - \|\Delta\|_1 \big| \leq 4 \, \|\Delta' - \Delta\|_1 = 4 \, \|\pi'(\cdot \mid s) - \pi(\cdot \mid s)\|_1.$$

Multiply by $C_2$, sum the two bounds, multiply by $1_X(s) \leq 1$, and take $\mathbb{E}_{s \sim d_T}$. This yields Eq.(54).

For Eq.(55), apply Eq.(54) to $\pi' = \mathsf{T}^t \pi = \pi + tv(\pi)$:

$$\frac{F(\mathsf{T}^t \pi) - F(\pi)}{t} \leq \frac{L_{\text{Lip}} \|t \, v(\pi)\|}{t} = L_{\text{Lip}} \|v(\pi)\|.$$

Taking $t \to 0^+$ gives the claim. $\qquad\square$

**Lemma D.10** (Negative right directional derivative implies strict decrease). *Let $\phi : [0, 1] \to \mathbb{R}$ be right-differentiable at $0$ and suppose $\phi'_+(0) < 0$. Then there exists $t_0 > 0$ such that $\phi(t) < \phi(0)$ for all $t \in (0, t_0]$.*

*Proof.* By definition of the right derivative, $\lim_{t \to 0^+} \frac{\phi(t) - \phi(0)}{t} = \phi'_+(0) < 0$. Hence there exists $t_0 > 0$ such that for all $t \in (0, t_0]$,

$$\frac{\phi(t) - \phi(0)}{t} \; < \; \frac{1}{2} \phi'_+(0) \; < \; 0,$$

which implies $\phi(t) < \phi(0)$. $\qquad\square$

### D.2.3. STRICT IMPROVEMENT BY A ONE-STEP HYBRID UPDATE

**Theorem D.11** (Strict improvement by a one-step Hybrid update). *Assume Lemma D.7, Eq.(51), and $\kappa_{\max} < \infty$. Define*

$$\nu_S := 2C_2 \underline{\delta}_{\mathcal{G}}^2, \qquad \nu_H := 8C_2 \Gamma_{\text{OS}}^2, \qquad L_{\text{Lip}} := \kappa_{\max} + 4C_2.$$

*If*

$$(L_{\text{Lip}}\varepsilon)^2 \; < \; \nu_S \nu_H, \tag{56}$$

*then there exist $\lambda \in (0, 1)$ and $t \in (0, 1]$ such that, letting*

$$\pi_* \in \arg\min\{F(\pi_{\text{hard}}), F(\pi_{\text{soft}})\}, \qquad \pi_{\text{hyb}} := \mathsf{T}_\lambda^t \pi_*,$$

*we have*

$$F(\pi_{\text{hyb}}) \; < \; \min\{F(\pi_{\text{hard}}), F(\pi_{\text{soft}})\}.$$

*Proof.* **Step 1 (a uniform negative hybrid-direction derivative at both endpoints).** By Lemma D.8,

$$D^+_{\mathsf{T}_\lambda} F(\pi) \leq (1-\lambda) D^+_{\mathsf{T}_H} F(\pi) + \lambda D^+_{\mathsf{T}_S} F(\pi), \qquad \forall \pi.$$

At $\pi_{\text{hard}}$, Lemma D.7 gives $D^+_{\mathsf{T}_S} F(\pi_{\text{hard}}) \leq -\nu_S$. Also, by Lemma D.9 and Eq.(51),

$$D^+_{\mathsf{T}_H} F(\pi_{\text{hard}}) \leq L_{\text{Lip}} \|v_H(\pi_{\text{hard}})\| \leq L_{\text{Lip}}\varepsilon.$$

Therefore,

$$D^+_{\mathsf{T}_\lambda} F(\pi_{\text{hard}}) \leq (1-\lambda) L_{\text{Lip}}\varepsilon - \lambda \nu_S. \tag{57}$$

Similarly, at $\pi_{\text{soft}}$, Lemma D.7 gives $D^+_{\mathsf{T}_H} F(\pi_{\text{soft}}) \leq -\nu_H$ and Lemma D.9 plus Eq.(51) gives $D^+_{\mathsf{T}_S} F(\pi_{\text{soft}}) \leq L_{\text{Lip}}\varepsilon$. Hence,

$$D^+_{\mathsf{T}_\lambda} F(\pi_{\text{soft}}) \leq \lambda L_{\text{Lip}}\varepsilon - (1-\lambda)\nu_H. \tag{58}$$

Choose $\lambda$ such that

$$\lambda > \frac{L_{\text{Lip}}\varepsilon}{\nu_S + L_{\text{Lip}}\varepsilon}, \qquad \lambda < \frac{\nu_H}{\nu_H + L_{\text{Lip}}\varepsilon}.$$

This is possible iff Eq.(56) holds. Fix any such $\lambda$. Then Eq.(57)–(58) imply

$$D^+_{\mathsf{T}_\lambda} F(\pi_{\text{hard}}) < 0, \qquad D^+_{\mathsf{T}_\lambda} F(\pi_{\text{soft}}) < 0. \tag{59}$$

**Step 2 (convert negative derivative into a strict one-step decrease).** Let $\pi_* \in \arg\min\{F(\pi_{\text{hard}}), F(\pi_{\text{soft}})\}$ and define $\phi(t) := F(\mathsf{T}_\lambda^t \pi_*)$. By Eq.(59), we have $\phi'_+(0) = D^+_{\mathsf{T}_\lambda} F(\pi_*) < 0$. Lemma D.10 then yields a $t_0 > 0$ such that $\phi(t) < \phi(0)$ for all $t \in (0, t_0]$.

Pick any $t \in (0, \min\{1, t_0\}]$ and set $\pi_{\text{hyb}} := \mathsf{T}_\lambda^t \pi_*$. Then

$$F(\pi_{\text{hyb}}) = \phi(t) < \phi(0) = F(\pi_*) = \min\{F(\pi_{\text{hard}}), F(\pi_{\text{soft}})\},$$

which proves the theorem. $\qquad\square$

# E. Detailed Experimental Settings

In this section, we provide comprehensive details regarding the dataset construction, training hyperparameters, and evaluation protocols used in our experiments.

## E.1. Training Data Construction

We describe the data construction process across distinct training settings. A primary challenge lies in ensuring the diversity of input queries, where corresponding responses are generated by the respective teacher models for each teacher-student configuration.

**General Reasoning Data Construction** Our training data is constructed starting from Infinity Instruct (Li et al., 2025). To ensure a balanced capability profile for the student model, we curated a high-quality subset by applying a filtering pipeline based on the provided `topic` tags and quality scores. We specifically retained samples with high quality ratings across several core domains to maintain data diversity:

- **Instruction Following & General Chat:** Includes dialogue, literature, and reading comprehension. This subset maintains fundamental world knowledge and responsiveness to human instructions.

- **Mathematical & Logical Reasoning:** Focuses on multi-step mathematical problems. This partition preserves complex reasoning chains and logical consistency.

- **Academic & Scientific Expertise:** Covers specialized subjects from MMLU (Hendrycks et al., 2021) (e.g., STEM, humanities) and scientific inquiries from ARC-C (Clark et al., 2018). This targets expert-level factual knowledge and scientific precision.

Through strategic balancing of these domains, we curated a final dataset of approximately **66,000 samples**. This size was chosen empirically to balance computational efficiency with sufficient diversity for smooth knowledge transfer.

**Mathematical Reasoning Data Construction**    We constructed the mathematical reasoning dataset by sampling and aggregating instances from MetaMathQA (Yu et al., 2023), DAPO-Math-17K (Yu et al., 2025), and the DeepScaleR-Preview-Dataset (Luo et al., 2025). This composite corpus covers a broad spectrum of difficulty levels, resulting in a final dataset of approximately 50,000 samples.

**Coding Data Construction**    We leverage the coding dataset from (Meng et al., 2024). Derived from CodeFeedback (Zheng et al., 2024), this subset contains 156,526 samples selected to target Python function-generation tasks similar to HumanEval/MBPP.

### E.2. Training Configurations

All distillation experiments are implemented using PyTorch and the Hugging Face `transformers` and `trl` libraries. We leverage DeepSpeed ZeRO-2 for efficient distributed training across NVIDIA A100 GPUs. The specific hyperparameters are listed below:

*Table 6.* Hyperparameters for distillation. **Common settings:** Batch Size = 128, Sequence Length = 2048, LoRA Dropout = 0.05, Target Modules = All Linear Layers.

| Domain | Teacher → Student | Training | | LoRA | |
|---|---|---|---|---|---|
| | | LR | Epochs | Rank ($r$) | Alpha ($\alpha$) |
| General | Qwen2.5 (7B → {0.5, 1.5, 3}B) | 5e−5 | 2 | 16 | 32 |
| | Llama-3.1 8B → Llama-3.2 1B | 1e−4 | 2 | 16 | 32 |
| | Gemma-3 (4B → 1B) | 1e−4 | 2 | 16 | 32 |
| Math | Qwen2.5-Math (7B → 1.5B) | 5e−5 | 2 | 16 | 32 |
| Code | DeepSeek-Coder (6.7B → 1.3B) | 2e−5 | 1 | 128 | 128 |

We implement the on-policy KD baseline following the official setup of (Agarwal et al., 2024; Ko et al., 2024). We mix a fixed dataset with student-generated samples based on a threshold. In our experiments, we use student samples when the threshold is below 0.3; otherwise, we train on teacher outputs. For fair comparison, all experiments are run for the same number of epochs on four A100 GPUs, and the average training cost (s/step) is then calculated.

Finally, although the experiments presented in the main text have demonstrated the superior performance of the adaptive hybrid supervision approach, we still provide the fixed lambda values tested across different settings to facilitate reproducibility. We also hope this work will offer the community further insights for designing more principled and effective learning objectives. Additionally, in our curriculum scheduling strategy, the warm-up steps are set to 20% of the total training steps. The results are listed in Table 7.

### E.3. Evaluation Benchmarks

We evaluate the models on widely recognized benchmarks to assess their reasoning capabilities. To ensure reproducibility and stability, we report the average accuracy score across five independent runs using random seeds $\{10, 20, 30, 40, 50\}$.

**General Reasoning.** We set the maximum generation length to 2048 tokens. For other generation parameters, we adopt the official configurations recommended by the model providers to achieve a proper trade-off between diversity and accuracy. We include the following benchmarks in this category:

*Table 7.* Fixed lambda values used in distillation experiments.

| Domain | Teacher → Student | $\lambda$ |
|--------|-------------------|-----------|
| General | Qwen2.5-7B → Qwen2.5-0.5B | 0.95 |
|  | Qwen2.5-7B → Qwen2.5-1.5B | 0.9 |
|  | Qwen2.5-7B → Qwen2.5-3B | 0.05 |
|  | Llama-3.1-8B → Llama-3.2-1B | 0.95 |
|  | Gemma-3-4B → Gemma-3-1B | 0.95 |
| Math | Qwen2.5-Math-7B → Qwen2.5-1.5B | 0.9 |
| Code | DeepSeek-Coder-6.7B → DeepSeek-Coder-1.3B | 0.1 |

- *MMLU* (Hendrycks et al., 2020): Evaluates multitask accuracy across 57 subjects, ranging from elementary mathematics to professional law.
- *BBH* (Suzgun et al., 2022): A subset of 23 challenging tasks from BIG-Bench designed to test complex multi-step reasoning capabilities.
- *ARC-C* (Clark et al., 2018): The Challenge Set of the AI2 Reasoning Challenge, consisting of grade-school science questions that require deep reasoning.

The prompt template used for these evaluations is as follows:

---

**Prompt Template for Multiple-Choice**

**System Message:**
You are an expert AI assistant that solves multiple-choice questions. Follow these steps for your response:

1. First, provide a detailed, step-by-step reasoning process that explains how you arrived at the solution.

2. After your reasoning, conclude with the final answer.

3. The final answer must be formatted by enclosing ONLY the single capital letter of the correct option in \boxed{}. For example, if the correct answer is option B, your final line should be exactly: \boxed{B}

---

**User Message:**
```
<Input Question Text Here>

A. <Option A Text>
B. <Option B Text>
C. <Option C Text>
D. <Option D Text>
```

---

**Model Response:**
```
<Detailed step-by-step reasoning process explaining the solution...>

\boxed{<Correct Option Letter>}
```

---

*Figure 7.* The complete interaction template used for multiple-choice benchmarks. The model is instructed to first generate a reasoning chain and then output the final answer in a specific format to facilitate automatic parsing.

**Mathematical Reasoning.** We maintain consistent generation configurations with the general reasoning tasks described above. To elicit intermediate reasoning steps, we employ Chain-of-Thought prompting. For evaluation, we adopt the validation protocol from the `verl` framework[1]: the final answer is extracted from the generated \boxed{} content and

---

[1] https://github.com/volcengine/verl

compared against the ground truth using exact matching. The benchmarks included in this category are:

- *GSM8K* (Cobbe et al., 2021): A benchmark of high-quality grade school math word problems focusing on multi-step reasoning.
- *MATH* (Hendrycks et al., 2021): A dataset containing challenging competition-level mathematics problems spanning seven diverse disciplines.
- *Gaokao2023-En* (Liao et al., 2024): English-translated questions from the 2023 Chinese National College Entrance Examination, evaluating advanced high-school mathematical proficiency.

The prompt template used for these mathematical tasks is presented in Figure 8.

---

**Prompt Template for Mathematical Reasoning**

**System Message:**
Please reason step by step, and put your final answer WITHIN \boxed{}.

- - - - - - - - - - - - - - - - - - - - - - - - - - - - - - - - - - - - -

**User Message:**
`<Mathematical Problem Text>`

- - - - - - - - - - - - - - - - - - - - - - - - - - - - - - - - - - - - -

**Model Response:**
`<Step-by-step derivation and calculation process...>`

Therefore, the answer is \boxed{`<Final Numerical or Symbolic Answer>`}.

---

*Figure 8.* The zero-shot Chain-of-Thought prompt used for mathematical reasoning tasks. The concise instruction forces the model to generate intermediate steps before producing a boxed final answer for exact-match validation.

**Code Generation.** We employ the `EvalPlus` framework[2] to assess programming capabilities. This framework evaluates functional correctness using an augmented set of rigorous unit tests to prevent false positives. We report results under greedy decoding following the official setup to ensure high performance and reproducibility.

- *HumanEval* (Chen, 2021): Measures code generation capabilities on 164 hand-written Python problems via the Pass@1 accuracy.
- *MBPP* (Austin et al., 2021): Evaluates functional correctness on entry-level programming problems using the sanitized subset.

## F. Additional Experimental Results

### F.1. Extended Evaluation and Control Experiments

This section complements the main experiments by asking four concrete questions. First, does the hard–soft pattern hold beyond the primary model pairs and benchmark formats? Second, can hybrid supervision be combined with on-policy prefixes? Third, can the gains be explained by cheaper alternatives such as reversing the proxy, changing temperature, or adding global regularization? Fourth, in a setting where exact token-level $\kappa$ is computable, do high-sensitivity tokens coincide with the semantic decision states predicted by the Bridge–Garden decomposition?

**Broader model and generation settings.** Table 8 tests whether the main trend is tied to the original model pairs or to closed-form benchmark evaluation. Hybrid KD remains better than both pure hard and pure soft KD for a larger Qwen2.5 teacher–student capacity gap, an additional Qwen2.5-Coder pair, and open-ended AlpacaEval generation.

**Compatibility with on-policy training.** On-policy KD changes which prefixes are used for training, whereas Hybrid KD changes the supervision target at a given prefix. These two changes address different parts of the training procedure and can

---

[2]https://github.com/evalplus/evalplus

*Table 8.* Broader model and open-ended evaluations. For Qwen2.5-32B→3B, the four columns are BBH/MMLU/ARC-C/ThmQA. For Qwen2.5-Coder-7B→1.5B, they are HumanEval/HumanEval+/MBPP/MBPP+. AlpacaEval reports GPT-5.2-judged win rate against text-davinci-003.

| Setting | Method | BBH /HE | MMLU /HE+ | ARC-C /MBPP | ThmQA /MBPP+ | Avg./Win |
|---|---|---|---|---|---|---|
| Qwen2.5-32B→3B | Hard KD | 34.4 | **67.2** | 79.5 | 23.1 | 51.0 |
| | Soft KD | 44.0 | 65.6 | 78.1 | 22.9 | 52.6 |
| | Hybrid KD | **45.7** | 66.8 | **80.0** | **24.0** | **54.1** |
| Qwen2.5-Coder-7B→1.5B | Hard KD | 54.3 | 50.0 | 60.3 | 52.1 | 54.2 |
| | Soft KD | 52.7 | 47.9 | 59.6 | 52.4 | 53.1 |
| | Hybrid KD | **55.5** | **50.6** | **61.4** | **52.6** | **55.0** |
| AlpacaEval (Qwen2.5-7B→3B) | Hard KD | | Win rate | | | 57.5 |
| | Soft KD | | Win rate | | | 61.3 |
| | Hybrid KD | | Win rate | | | **64.4** |

be combined. Tables 9 and 10 show that adding hybrid supervision on top of on-policy soft KD improves the average score across all four settings.

*Table 9.* Hybrid supervision combined with on-policy training on reasoning benchmarks. The Qwen2.5-7B→3B setting is reported as benchmark average; Llama3.1-8B→1B reports the full BBH/MMLU/ARC-C/ThmQA breakdown.

| Setting | Method | BBH | MMLU | ARC-C | ThmQA | Avg. |
|---|---|---|---|---|---|---|
| Qwen2.5-7B→3B | Off-policy soft KD | | Avg. only | | | 51.9 |
| | Off-policy hybrid KD | | Avg. only | | | 53.5 |
| | On-policy soft KD | | Avg. only | | | 52.8 |
| | On-policy + hybrid KD | | Avg. only | | | **54.7** |
| Llama3.1-8B→1B | Off-policy soft KD | 22.1 | 33.1 | 33.4 | 4.4 | 23.3 |
| | Off-policy hybrid KD | **27.4** | 35.6 | 37.0 | 5.0 | 26.3 |
| | On-policy soft KD | 26.8 | 36.2 | 39.7 | 7.0 | 27.4 |
| | On-policy + hybrid KD | 27.2 | **36.7** | **40.2** | **7.1** | **27.8** |

*Table 10.* Hybrid supervision combined with on-policy training on code benchmarks. The columns are HumanEval (HE), HumanEval+ (HE+), MBPP, MBPP+, and their average.

| Setting | Method | HE | HE+ | MBPP | MBPP+ | Avg. |
|---|---|---|---|---|---|---|
| Qwen2.5-Coder-7B→1.5B | Off-policy soft KD | 52.7 | 47.9 | 59.6 | 52.4 | 53.1 |
| | Off-policy hybrid KD | 55.5 | 50.6 | **61.4** | 52.6 | 55.0 |
| | On-policy soft KD | 56.4 | 51.5 | **61.4** | **53.2** | 55.6 |
| | On-policy + hybrid KD | **56.8** | **52.9** | 61.1 | 52.9 | **55.9** |
| DeepSeek-Coder-6.7B→1.3B | Off-policy soft KD | 38.4 | 33.5 | 63.5 | 51.6 | 46.8 |
| | Off-policy hybrid KD | 41.5 | 36.6 | 63.2 | 50.5 | 48.0 |
| | On-policy soft KD | 43.3 | 39.0 | 63.8 | 52.1 | 49.5 |
| | On-policy + hybrid KD | **44.2** | **39.8** | **64.4** | **52.5** | **50.2** |

**Reverse-proxy ablation.** The practical algorithms use teacher confidence or entropy as local heuristics for allocating supervision at semantic decision states. If Hybrid KD only benefited from generic label mixing, then reversing the confidence or entropy rule should remain competitive. Table 11 shows the opposite: reverse-confidence and reverse-entropy both reduce the average score on Qwen2.5-7B→3B. This supports that the direction of the local hard/soft allocation matters.

**Training cost breakdown.** All methods in Table 12 use the same epochs, batch size, and hardware. Teacher logits for soft and hybrid supervision are computed on the fly rather than stored. Hybrid KD reuses the same teacher logits as soft KD

*Table 11.* Reverse-proxy ablation on Qwen2.5-7B→3B. Reversing confidence or entropy weighting reduces the average score.

| Method | BBH | MMLU | ARC-C | ThmQA | Avg. |
|---|---|---|---|---|---|
| Hard KD | 41.5 | 65.8 | 78.8 | 23.8 | 52.5 |
| Soft KD | 41.7 | 64.5 | 78.3 | 23.0 | 51.9 |
| Hybrid: confidence-based | 44.1 | **67.5** | **80.8** | 22.8 | 53.8 |
| Hybrid: reverse-confidence | 41.5 | 65.3 | 78.1 | 22.3 | 51.8 |
| Hybrid: entropy | **46.8** | 67.1 | 79.8 | 23.7 | **54.3** |
| Hybrid: reverse-entropy | 41.2 | 65.2 | 77.3 | 20.5 | 51.1 |

and only adds a lightweight per-token weighting step. By contrast, on-policy KD requires autoregressive student sampling, which accounts for most of its additional cost.

*Table 12.* Training cost on Qwen2.5-7B→3B with 4×A100 80GB.

| Method | Extra computation | s/step |
|---|---|---|
| Hard KD | none | 8.30 |
| Soft KD | none | 14.06 |
| Hybrid KD | one log-sum-exp per token | 15.24 |
| On-policy KD | autoregressive student sampling | 147.83 |

**Regularization and temperature controls.** We next test whether the gains can be reproduced without position-specific hard/soft allocation. Entropy regularization and temperature schedules provide global changes to the target distribution, and random-label mixing tests whether any hard-token signal is sufficient. Tables 13 and 14 show the full benchmark breakdowns. These alternatives can improve over pure soft KD, but they do not consistently match Hybrid KD. The random-label control performs especially poorly, indicating that the hard-label component must remain teacher-supported and context-dependent.

*Table 13.* Regularization and temperature controls on reasoning benchmarks. Avg. is computed over BBH, MMLU, ARC-C, and ThmQA. Hybrid KD is best by average in both settings.

| Setting | Method | BBH | MMLU | ARC-C | ThmQA | Avg. |
|---|---|---|---|---|---|---|
| Qwen2.5-7B→3B | Soft KD | 41.7 | 64.5 | 78.3 | 23.0 | 51.9 |
| | Entropy reg. | **46.6** | 67.0 | 80.5 | **23.8** | 54.5 |
| | T: high→low | 45.2 | 66.1 | 80.1 | 23.8 | 53.8 |
| | T: low→high | 46.4 | 66.5 | 80.6 | 23.3 | 54.2 |
| | Random labels | 40.1 | 61.2 | 74.2 | 20.3 | 49.0 |
| | Hybrid KD | 46.5 | **69.1** | **81.2** | **23.8** | **55.2** |
| Llama3.1-8B→1B | Soft KD | 22.1 | 33.1 | 33.4 | 4.4 | 23.3 |
| | Entropy reg. | 24.5 | **35.7** | 34.9 | **6.0** | 25.3 |
| | T: high→low | 26.0 | 35.1 | 34.7 | 5.8 | 25.4 |
| | T: low→high | 25.7 | 35.4 | 34.8 | 5.0 | 25.2 |
| | Random labels | 19.2 | 30.1 | 30.2 | 2.1 | 20.4 |
| | Hybrid KD | **27.4** | 35.6 | **37.0** | 5.0 | **26.3** |

## F.2. Controlled Synthetic Validation with Exact Token-Level $\kappa$

Exact $\kappa$ is intractable for real LLMs because it requires path-level intervention over future continuations. The synthetic experiment is designed to test the Bridge–Garden mechanism in a setting where this quantity is observable. We construct controlled domains where the semantic role of each token is known and exact token-level $\kappa$ can be computed. The synthetic experiment uses three domains, Dialogue, Math, and Code, with 4000 training examples, 500 validation examples, and 30 evaluation examples per domain. Each domain uses a vocabulary of 64 tokens; the evaluation action space has $|\mathcal{V}_{\text{eval}}| = 62$, includes EOS, and excludes only PAD and BOS. Each domain is generated by a scripted teacher oracle with a fixed template and typed decision states. Dialogue contains recipient, date/time, required-fact, forbidden-constraint, and tone/paraphrase

*Table 14.* Regularization and temperature controls on code benchmarks. Avg. is computed over HumanEval (HE), HumanEval+ (HE+), MBPP, and MBPP+. Hybrid KD is best or tied by average in both settings.

| Setting | Method | HE | HE+ | MBPP | MBPP+ | Avg. |
|---|---|---|---|---|---|---|
| Qwen2.5-Coder-7B→1.5B | Soft KD | 52.7 | 47.9 | 59.6 | 52.4 | 53.1 |
| | Entropy reg. | 53.0 | 48.2 | **63.5** | **55.3** | **55.0** |
| | T: high→low | 53.0 | 47.6 | 60.3 | 52.6 | 53.4 |
| | T: low→high | 54.0 | 48.7 | 60.9 | 51.9 | 53.9 |
| | Random labels | 51.8 | 48.2 | 60.6 | 53.2 | 53.5 |
| | Hybrid KD | **55.5** | **50.6** | 61.4 | 52.6 | **55.0** |
| DeepSeek-Coder-6.7B→1.3B | Soft KD | 38.4 | 33.5 | 63.5 | 51.6 | 46.8 |
| | Entropy reg. | **41.5** | **36.6** | 61.6 | 50.0 | 47.4 |
| | T: high→low | 40.1 | 35.4 | 62.7 | 49.9 | 47.0 |
| | T: low→high | 39.7 | 34.7 | 62.9 | 50.3 | 46.9 |
| | Random labels | 35.4 | 32.3 | 57.4 | 47.6 | 43.2 |
| | Hybrid KD | **41.5** | **36.6** | **63.2** | 50.5 | **48.0** |

choices; Math contains substitution, operator, computed-value, final-answer, and equivalent-representation choices; Code contains arithmetic/update operators, branch guards, return semantics, syntax/layout, and equivalent-implementation choices. The decision order is fixed before sampling: Dialogue interleaves tone, recipient, required fact, tone, date/time, and forbidden-constraint states; Math uses substitution, equivalent form, operator, computed value, final answer, and equivalent form; Code uses three equivalent-implementation states followed by operator, branch guard, operator, and return-semantics states inside a Python-like function template. High-risk states use four teacher-supported semantic choices: the main token probability starts at $0.98$, decays by $0.01$ across high-risk states, and is floored at $0.94$, with the remaining probability shared by the other three choices. Flexible states use eight equivalent choices with logits evenly spaced from $0$ to $-2.5$ before normalization, and ordinary context or layout tokens are deterministic. This design provides semantic labels that let us interpret Bridge-like and Garden-like behavior, while $\kappa$ and the reported states are computed and selected independently of those labels.

We train three student models under Hard KD, Soft KD, and Hybrid KD on samples from the same oracle. All students use a 2-layer Transformer with $d_{\text{model}} = 128$, 4 attention heads, feed-forward width 512, and dropout 0.1. Optimization uses AdamW, batch size 256, learning rate $2 \times 10^{-4}$, warmup-cosine scheduling with 6 warmup epochs, 24 maximum epochs, and early-stopping patience 6. Hybrid KD selects its mixing coefficient from $\{0.1, \ldots, 0.9\}$ on the held-out validation split: teacher-forced KL for Dialogue, validation rollout EB for Math, and, for Code, the smallest $\lambda$ whose validation rollout EB is within one standard error of the best validation value. The selected coefficients are $0.9$ for Dialogue and $0.1$ for Math and Code. For Code, high-risk operator, branch, and return states use hard-label supervision, while the remaining states use a local soft-label weight equal to the selected $\lambda$ times the normalized teacher ambiguity at that state.

For each trained student $m$, each evaluated prefix state $s$, and each token $a \in \mathcal{V}_{\text{eval}}$, we compute exact token-level sensitivity by forcing $a$ at $s$ and then following the teacher oracle for all future continuations:

$$\kappa_m(a \mid s) = Q_m(s, a) - \sum_{a'} \pi_T(a' \mid s) Q_m(s, a'),$$

where $Q_m(s, a)$ is the expected downstream student loss after the forced token. Overall EB is measured as the difference between the student's loss on its own rollout prefixes and teacher-forced prefixes, averaged over 30 rollout repeats.

For evaluation, Dialogue uses template length 35 and 1050 evaluated prefix states, Math uses template length 43 and 1290 evaluated prefix states, and Code uses template length 47 and 1410 evaluated prefix states, each evaluated over 62 actions. Table 15 reports the Bridge/Garden regional decomposition and overall EB. For this decomposition, Bridge and Garden regions are the top and bottom 20% of evaluated states ranked by computed $\kappa$; the columns report absolute $\kappa$-weighted contributions within those regions, where lower is better. The results match the predicted regional pattern: hard KD has lower Bridge contribution than soft KD, soft KD has lower Garden contribution than hard KD, and hybrid KD has the lowest overall EB. The token-level maps show the same pattern visually: large $\kappa$ contributions appear mainly at tokens that decide task semantics, while ordinary context and layout tokens have small contributions.

We also examine whether the local teacher-confidence signal aligns with exact $\kappa$ on the states where confidence-based

*Table 15.* Synthetic Bridge/Garden decomposition and overall exposure bias (EB). Lower is better. Bridge/Garden entries are absolute $\kappa$-weighted contributions under the common top/bottom 20% non-terminal split.

| Domain | $\lambda$ | Bridge Hard KD | Bridge Soft KD | Garden Hard KD | Garden Soft KD | Overall EB Hard KD | Overall EB Soft KD | Overall EB Hybrid KD |
|---|---|---|---|---|---|---|---|---|
| Dialogue | 0.9 | **0.1033** | 0.3005 | 0.0145 | **0.0051** | 0.0640 | 0.1745 | **0.0470** |
| Math | 0.1 | **0.1273** | 0.3479 | 0.0361 | **0.0186** | 0.0316 | 0.1478 | **0.0260** |
| Code | 0.1 | **0.2553** | 0.3254 | 0.0617 | **0.0372** | 0.0281 | 0.0821 | **0.0280** |

Hybrid KD is intended to act. For this analysis, we focus on states where the teacher has multiple valid next-token choices and define $c_T(s) = 1 - H_T(s)/\log K_T(s)$, where $K_T(s)$ is the number of teacher-defined candidate next-token choices at state $s$. Here $H_T(s)$ is the entropy of the teacher distribution over these candidate choices. Thus $c_T(s)$ is one minus normalized entropy: it is near 0 when the teacher is close to uniform over the candidates and larger when the teacher distribution is sharper. Table 16 shows that $c_T$ is strongly rank-aligned with exact state-level $\kappa$ across all three synthetic domains. The decomposition in Table 15 and the heatmaps also includes deterministic context and layout states.

*Table 16.* Teacher confidence and exact token-level sensitivity in the synthetic experiments. $\rho$ is Spearman correlation between $c_T(s)$ and exact state-level $\kappa(s)$ on states where the teacher has multiple valid next-token choices. Mean $c_T$ is reported separately for Bridge and Garden states under the same partition as Table 15.

| Domain | Choice states | Spearman $\rho$ $c_T$ vs. $\kappa$ | Mean $c_T$ Bridge | Mean $c_T$ Garden |
|---|---|---|---|---|
| Dialogue | 180 | 0.972 | 0.829 | 0.134 |
| Math | 180 | 0.972 | 0.864 | 0.134 |
| Code | 210 | 0.954 | 0.755 | 0.134 |

Figure 9 shows representative complete token-level $\kappa$ maps. Each row is one domain and includes all token states for the example, including deterministic context and layout positions. Light cells denote zero or near-zero contribution, while high-contribution cells concentrate on semantic decision tokens such as recipients, required facts, substitutions, operators, branch guards, and return values. Figures 10 and 11 provide additional representative examples.

**Complete token-level κ maps: representative examples A**

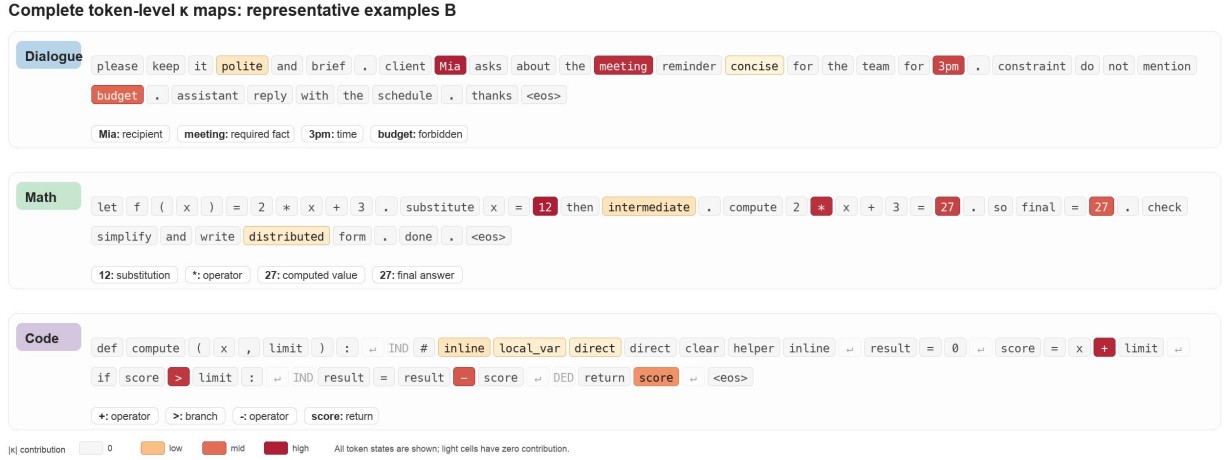

*Figure 9.* Representative complete token-level $\kappa$ maps across Dialogue, Math, and Code. Color intensity indicates $|\kappa|$; light cells have zero or near-zero contribution.

**Complete token-level κ maps: representative examples B**

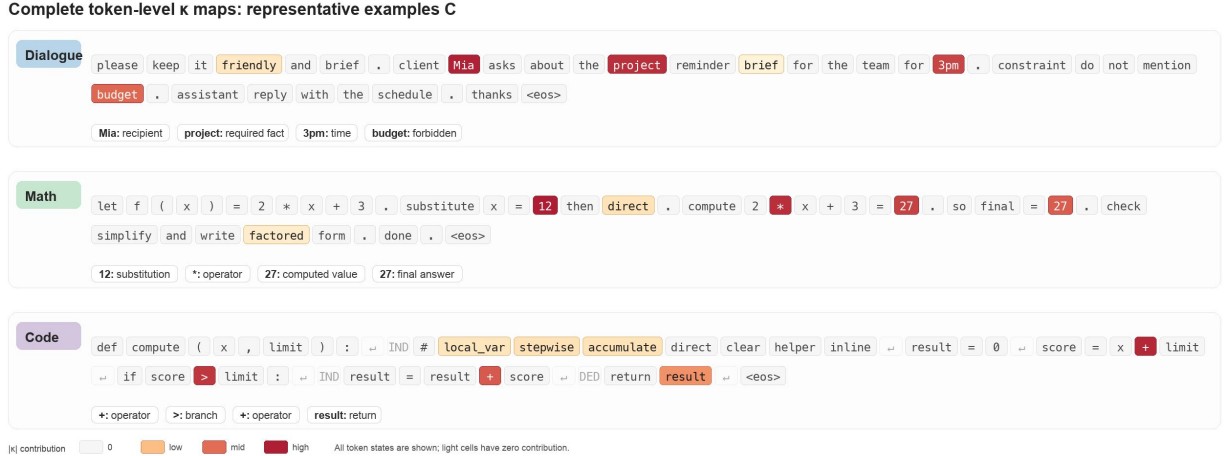

*Figure 10.* Additional representative complete token-level $\kappa$ maps across Dialogue, Math, and Code. Color intensity indicates $|\kappa|$; light cells have zero or near-zero contribution.

**Complete token-level κ maps: representative examples C**

*Figure 11.* Additional representative complete token-level $\kappa$ maps across Dialogue, Math, and Code. Color intensity indicates $|\kappa|$; light cells have zero or near-zero contribution.

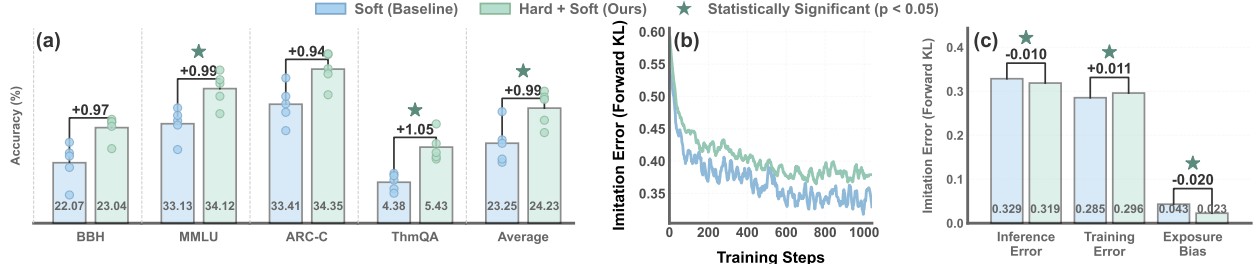

*Figure 12.* Comparative analysis (Qwen2.5 7B → 3B) of Hybrid KD ($\lambda\ell_{\text{soft}} + (1 - \lambda)\ell_{\text{hard}}$) vs. Soft KD ($\ell_{\text{soft}}$). (a) benchmark performance gains, (b) student-teacher imitation error during training (quantified by Reverse KL), and (c) inference imitation error decomposition based on the same metric.

*Figure 13.* Comparative analysis (Llama3.1-8B → 3.2-1B) of Hybrid KD ($\lambda\ell_{\text{soft}} + (1 - \lambda)\ell_{\text{hard}}$) vs. Soft KD ($\ell_{\text{soft}}$). (a) benchmark performance gains, (b) student-teacher imitation error during training (quantified by Forward KL), and (c) inference imitation error decomposition based on the same metric.

### F.3. Further Validation of the Hard-Soft Paradox

In this subsection, we provide additional empirical support to validate the universality of the Hard-Soft Paradox. We extend our analysis of training dynamics to a broader range of model families, including Llama, Qwen2.5-Math, DeepSeek-Coder, and Gemma.

As shown in Figures 12, 13, and 14, the hard-soft paradox remains consistent across different models, tasks, and divergence metrics. Moreover, Table 19, Table 20, and Figure 3 further confirm that these benefits hold true across various divergence measures. Collectively, these results suggest the paradox is robust across diverse model families and domains. This in turn underscores the importance of our work in addressing this puzzle in the existing literature.

### F.4. Further Analysis of Hybrid Strategies

We further examine the impact of different hybrid supervision methods on mathematical reasoning tasks. In Table 17, we present experimental results for distillation from Qwen2.5-Math-7B to Qwen2.5-1.5B under various approaches. These results consistently outperform those obtained using pure Hard KD or Soft KD alone. Moreover, as expected, the adaptive strategy designed under the inspiration of the Bridge–Garden theory achieves correspondingly better performance.

### F.5. Validation on Additional Teacher-Student Configurations

We also evaluate our framework on another teacher-student configuration, specifically distilling from Qwen2.5-7B to Qwen2.5-1.5B. As shown in Table 18, the hybrid distillation strategy consistently outperforms standard distillation baselines across all benchmarks. This further confirms the general effectiveness of the hybrid supervision method observed in our main experiments.

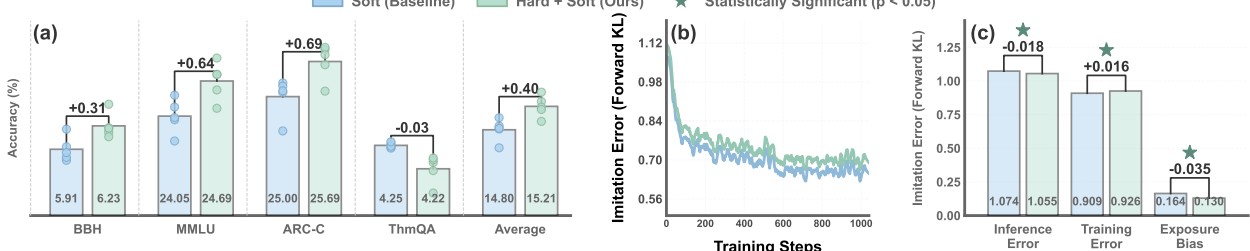

*Figure 14.* Comparative analysis (Gemma3-4B → 1B) of Hybrid KD ($\lambda\ell_{\text{soft}} + (1 - \lambda)\ell_{\text{hard}}$) vs. Soft KD ($\ell_{\text{soft}}$). (a) benchmark performance gains, (b) student-teacher imitation error during training (quantified by Forward KL), and (c) inference imitation error decomposition based on the same metric.

*Table 17.* Distillation results on Qwen2.5-Math models (↑). **Bold** with darker blue indicates Rank 1; light blue indicates Rank 2.

| Method | Qwen2.5-Math-7B → 1.5B | | | |
| --- | --- | --- | --- | --- |
| | **GSM8K** | **MATH** | **Gaokao23** | **Avg.** |
| Hard KD | $65.75_{\pm4.43}$ | $50.44_{\pm1.78}$ | $41.04_{\pm3.15}$ | 52.41 |
| Soft KD | $68.72_{\pm0.92}$ | $49.89_{\pm0.82}$ | $41.77_{\pm1.11}$ | 53.46 |
| Static Weighting | $70.42_{\pm1.89}$ | $\mathbf{51.96}_{\pm1.24}$ | $42.75_{\pm2.64}$ | 55.04 |
| Curriculum-based | $71.10_{\pm0.25}$ | $50.98_{\pm0.71}$ | $43.58_{\pm1.36}$ | 55.22 |
| Risk-Guided | $\mathbf{71.96}_{\pm0.46}$ | $51.46_{\pm0.64}$ | $42.91_{\pm1.76}$ | 55.45 |
| Entropy-based | $70.77_{\pm0.78}$ | $49.91_{\pm1.39}$ | $42.96_{\pm0.56}$ | 54.55 |
| **Confidence-based** | $71.65_{\pm1.20}$ | $51.37_{\pm0.71}$ | $\mathbf{44.10}_{\pm2.09}$ | **55.71** |

*Table 18.* Distillation results from Qwen2.5-7B (Teacher) to Qwen2.5-1.5B (Student) on general reasoning benchmarks. We use Forward KL for soft supervision and static weighting for hybrid supervision.

| Setting | Method | BBH | MMLU | ARC-Challenge | TheoremQA | Average |
| --- | --- | --- | --- | --- | --- | --- |
| **Teacher** | Qwen2.5-7B-Instruct | $64.66_{\pm0.69}$ | $78.22_{\pm0.22}$ | $89.90_{\pm0.24}$ | $32.47_{\pm0.32}$ | 66.31 |
| | Baseline (No Distill) | $25.87_{\pm0.06}$ | $58.99_{\pm0.11}$ | $71.11_{\pm0.07}$ | $9.90_{\pm0.27}$ | 41.47 |
| | Hard KD | $31.98_{\pm0.35}$ | $55.68_{\pm0.25}$ | $64.28_{\pm0.38}$ | $19.23_{\pm0.80}$ | 42.79 |
| **Student** | Forward KL divergence | $35.08_{\pm0.19}$ | $60.27_{\pm0.11}$ | $71.83_{\pm0.21}$ | $17.15_{\pm0.74}$ | 46.08 |
| **(Qwen2.5-1.5B)** | Reverse KL divergence | $\mathbf{35.88}_{\pm0.15}$ | $59.96_{\pm0.11}$ | $71.30_{\pm0.13}$ | $17.00_{\pm0.32}$ | 46.04 |
| | Skew Forward KL | $32.54_{\pm0.20}$ | $60.39_{\pm0.12}$ | $71.88_{\pm0.19}$ | $16.25_{\pm0.26}$ | 45.26 |
| | Skew Reverse KL | $32.99_{\pm0.24}$ | $60.31_{\pm0.12}$ | $71.62_{\pm0.12}$ | $17.62_{\pm0.76}$ | 45.64 |
| | Total Variation | $35.48_{\pm0.12}$ | $60.40_{\pm0.07}$ | $71.43_{\pm0.06}$ | $17.08_{\pm0.48}$ | 46.10 |
| | **HybKD (Ours)** | $35.11_{\pm0.26}$ | $\mathbf{61.68}_{\pm0.24}$ | $\mathbf{72.65}_{\pm0.33}$ | $\mathbf{18.65}_{\pm0.27}$ | **47.02** |

*Table 19.* DeepSeek-Coder distillation results (↑). We report pass@1 scores. **Bold** with darker background indicates the best performance; light background indicates the second-best performance.

| Method | DeepSeek-Coder-6.7B → 1.3B | | | | |
|---|---|---|---|---|---|
| | HumanEval | HumanEval+ | MBPP | MBPP+ | Avg. |
| **Reference & Baselines** | | | | | |
| *Teacher* | 76.83 | 71.34 | 75.32 | 66.75 | 72.56 |
| *Student (No Distill)* | 33.54 | 27.44 | 58.44 | 49.61 | 42.26 |
| Hard KD | 35.37 | 32.93 | 60.32 | 50.00 | 44.66 |
| **KL Divergence Family** | | | | | |
| Forward KL | 38.41 | 33.54 | 63.49 | 51.59 | 46.76 |
| + *Ours* | 41.46 | 36.59 | 63.12 | 50.39 | 47.89 |
| Reverse KL | 39.02 | 34.15 | 62.17 | 50.00 | 46.34 |
| + *Ours* | 37.80 | 34.15 | 63.38 | 51.95 | 46.82 |
| **Distance & Skewed Family** | | | | | |
| Total Variation | 39.63 | 35.37 | 61.56 | 49.61 | 46.54 |
| + *Ours* | 40.02 | 36.16 | 62.10 | 49.97 | 47.06 |
| Skew FKL | 42.07 | 35.98 | 60.85 | 50.26 | 47.29 |
| + *Ours* | 42.41 | 35.75 | 63.12 | 50.39 | 47.92 |
| Skew RKL | 42.07 | 35.98 | 60.78 | 50.39 | 47.31 |
| + *Ours* | 41.24 | 35.98 | 62.34 | 51.17 | 47.68 |
| **Unified Divergence Family** | | | | | |
| $\alpha$-$\beta$ divergence | 41.46 | 36.59 | 60.05 | 50.26 | 47.09 |
| **+ *Ours*** | **43.90** | **38.41** | **64.68** | **51.95** | **49.74** |

*Table 20.* Distillation results on Gemma-3 models (↑). **Bold** with darker background indicates the best performance; light background indicates the second-best performance.

| Method | Gemma3-4B → Gemma3-1B | | | | |
|---|---|---|---|---|---|
| | BBH | MMLU | ARC-C | ThmQA | Avg. |
| **Reference & Baselines** | | | | | |
| *Teacher* | 52.57 | 66.17 | 81.23 | 26.12 | 56.52 |
| *Student (No Distill)* | 6.47 | 15.94 | 15.70 | 0.95 | 9.76 |
| Hard KD | 5.28 | 18.11 | 18.09 | 3.15 | 11.16 |
| **KL Divergence Family** | | | | | |
| Forward KL | 5.91 | 24.05 | 25.00 | 4.25 | 14.80 |
| + *Ours* | 6.23 | 24.69 | 25.69 | 4.22 | 15.21 |
| Reverse KL | **9.52** | 24.77 | **27.58** | 1.26 | 15.78 |
| + *Ours* | 7.14 | 25.27 | 26.03 | 4.57 | 15.75 |
| **Distance & Skewed Family** | | | | | |
| Total Variation | 6.29 | 25.07 | 25.83 | 1.29 | 14.62 |
| + *Ours* | 5.02 | 24.13 | 25.39 | 4.00 | 14.64 |
| Skew FKL | 7.01 | 25.08 | 26.14 | 5.01 | 15.81 |
| + *Ours* | 7.78 | **25.94** | 26.07 | **5.24** | **16.26** |
| Skew RKL | 5.83 | 25.17 | 25.93 | 3.38 | 15.08 |
| + *Ours* | 5.72 | 25.73 | 26.92 | 3.41 | 15.45 |
| **Unified Divergence Family** | | | | | |
| $\alpha$-$\beta$ divergence | 7.26 | 25.49 | 26.07 | 4.34 | 15.79 |
| **+ *Ours*** | 7.56 | 25.69 | 25.90 | 4.54 | 15.92 |

*Table 21.* Performance comparison of Qwen2.5 on reasoning benchmarks. Avg. is the mean across all benchmarks. We evaluate our hybrid KD against recent soft KD methods.

| Method | Qwen2.5-7B → Qwen2.5-0.5B | | | | | Qwen2.5-7B → Qwen2.5-3B | | | | |
|---|---|---|---|---|---|---|---|---|---|---|
| | BBH | MMLU | ARC-C | ThmQA | Avg. | BBH | MMLU | ARC-C | ThmQA | Avg. |
| *Teacher* | $64.66_{\pm0.69}$ | $78.22_{\pm0.22}$ | $89.90_{\pm0.24}$ | $32.47_{\pm0.32}$ | 66.31 | $64.66_{\pm0.69}$ | $78.22_{\pm0.22}$ | $89.90_{\pm0.24}$ | $32.47_{\pm0.32}$ | 66.31 |
| *Student (No Distill)* | $3.81_{\pm0.07}$ | $44.67_{\pm0.12}$ | $43.72_{\pm0.12}$ | $7.78_{\pm0.39}$ | 24.99 | $22.34_{\pm0.06}$ | $64.61_{\pm0.06}$ | $78.40_{\pm0.03}$ | $12.22_{\pm0.25}$ | 44.39 |
| Hard KD (Kim & Rush, 2016) | $10.27_{\pm0.10}$ | $36.83_{\pm0.15}$ | $40.58_{\pm0.70}$ | $11.35_{\pm0.43}$ | 24.76 | $41.52_{\pm0.33}$ | $65.76_{\pm0.18}$ | $78.75_{\pm0.71}$ | $23.75_{\pm0.40}$ | 52.45 |
| Forward KL (Hinton et al., 2015) | $24.42_{\pm0.06}$ | $43.19_{\pm0.20}$ | $46.84_{\pm0.19}$ | $10.33_{\pm0.26}$ | 31.20 | $41.65_{\pm0.16}$ | $64.45_{\pm0.04}$ | $78.33_{\pm0.22}$ | $23.02_{\pm0.33}$ | 51.87 |
| Reverse KL (Gu et al., 2024) | $24.91_{\pm0.01}$ | $44.72_{\pm0.01}$ | $47.44_{\pm0.00}$ | $11.00_{\pm0.00}$ | 32.02 | $44.07_{\pm0.10}$ | $65.67_{\pm0.19}$ | $77.68_{\pm0.12}$ | $24.20_{\pm0.49}$ | 52.90 |
| Total Variation (Wen et al., 2023) | $26.74_{\pm0.38}$ | $44.35_{\pm0.12}$ | $46.76_{\pm0.51}$ | $10.20_{\pm0.56}$ | 32.01 | $40.50_{\pm0.16}$ | $64.52_{\pm0.03}$ | $78.11_{\pm0.18}$ | $22.83_{\pm0.56}$ | 51.49 |
| JS divergence (Agarwal et al., 2024) | $24.55_{\pm0.13}$ | $43.31_{\pm0.10}$ | $44.78_{\pm0.05}$ | $11.82_{\pm0.37}$ | 31.12 | $45.50_{\pm0.08}$ | $64.68_{\pm0.17}$ | $78.85_{\pm0.14}$ | $22.27_{\pm0.41}$ | 52.83 |
| Adaptive KL (Wu et al., 2025) | $26.04_{\pm0.23}$ | $41.85_{\pm0.11}$ | $45.89_{\pm0.32}$ | $11.33_{\pm0.61}$ | 31.28 | $44.71_{\pm0.13}$ | $64.69_{\pm0.07}$ | $79.23_{\pm0.22}$ | $22.25_{\pm0.33}$ | 52.72 |
| Skew FKL (Ko et al., 2024; 2025) | $25.87_{\pm0.24}$ | $44.12_{\pm0.24}$ | $47.12_{\pm0.65}$ | $10.65_{\pm0.22}$ | 31.94 | $41.39_{\pm0.17}$ | $64.67_{\pm0.15}$ | $77.75_{\pm0.23}$ | $23.77_{\pm0.71}$ | 51.89 |
| Skew RKL (Ko et al., 2024; 2025) | $28.16_{\pm0.16}$ | $45.03_{\pm0.05}$ | $47.51_{\pm0.20}$ | $11.37_{\pm0.66}$ | 33.02 | $41.22_{\pm0.08}$ | $63.95_{\pm0.08}$ | $76.91_{\pm0.18}$ | $23.67_{\pm0.20}$ | 51.44 |
| $\alpha$-$\beta$ divergence (Wang et al., 2025b) | $26.18_{\pm0.15}$ | $42.59_{\pm0.18}$ | $46.70_{\pm0.37}$ | $11.07_{\pm0.43}$ | 31.63 | $45.12_{\pm0.23}$ | $64.95_{\pm0.17}$ | $79.81_{\pm0.09}$ | $22.94_{\pm0.54}$ | 53.21 |
| **HybKD (Ours)** | $\mathbf{26.58_{\pm0.16}}$ | $\mathbf{49.08_{\pm0.22}}$ | $\mathbf{51.69_{\pm0.62}}$ | $\mathbf{10.50_{\pm0.54}}$ | **34.46** | $\mathbf{46.53_{\pm0.05}}$ | $\mathbf{69.05_{\pm0.07}}$ | $\mathbf{81.23_{\pm0.00}}$ | $\mathbf{23.82_{\pm0.58}}$ | **55.16** |

*Table 22.* Accuracy (↑) on Llama and Gemma models.

| Method | Llama3.1-8B → Llama3.2-1B | | | | | Gemma3-4B → Gemma3-1B | | | | |
|---|---|---|---|---|---|---|---|---|---|---|
| | BBH | MMLU | ARC-C | ThmQA | Avg. | BBH | MMLU | ARC-C | ThmQA | Avg. |
| *Teacher* | $57.72_{\pm0.07}$ | $70.90_{\pm0.03}$ | $83.58_{\pm0.10}$ | $18.10_{\pm0.40}$ | 57.58 | $52.57_{\pm0.00}$ | $66.17_{\pm0.00}$ | $81.23_{\pm0.00}$ | $26.12_{\pm0.00}$ | 56.52 |
| *Student (No Distill)* | $14.01_{\pm4.41}$ | $19.78_{\pm9.89}$ | $21.57_{\pm9.94}$ | $2.22_{\pm0.44}$ | 14.40 | $6.47_{\pm3.41}$ | $15.94_{\pm8.91}$ | $15.70_{\pm8.49}$ | $0.95_{\pm0.20}$ | 9.76 |
| Hard KD (Kim & Rush, 2016) | $15.29_{\pm2.75}$ | $22.54_{\pm1.38}$ | $23.98_{\pm1.96}$ | $3.88_{\pm0.73}$ | 16.42 | $5.28_{\pm1.89}$ | $18.11_{\pm4.83}$ | $18.09_{\pm5.25}$ | $3.15_{\pm0.15}$ | 11.16 |
| Forward KL (Hinton et al., 2015) | $22.07_{\pm2.11}$ | $33.13_{\pm1.69}$ | $33.41_{\pm1.78}$ | $4.37_{\pm0.43}$ | 23.25 | $5.91_{\pm1.43}$ | $24.05_{\pm2.01}$ | $25.00_{\pm2.83}$ | $4.25_{\pm0.33}$ | 14.80 |
| Reverse KL (Gu et al., 2024) | $23.69_{\pm0.77}$ | $31.63_{\pm1.88}$ | $32.08_{\pm1.20}$ | $3.60_{\pm0.09}$ | 22.75 | $9.52_{\pm1.03}$ | $24.77_{\pm0.68}$ | $\mathbf{27.58_{\pm0.60}}$ | $1.26_{\pm0.32}$ | 15.78 |
| Total Variation (Wen et al., 2023) | $23.55_{\pm1.51}$ | $27.41_{\pm4.96}$ | $28.99_{\pm4.47}$ | $2.68_{\pm0.56}$ | 20.66 | $6.29_{\pm0.86}$ | $25.07_{\pm0.31}$ | $25.83_{\pm0.43}$ | $1.29_{\pm0.41}$ | 14.62 |
| JS divergence (Agarwal et al., 2024) | $25.16_{\pm2.03}$ | $34.08_{\pm2.15}$ | $34.42_{\pm1.81}$ | $\mathbf{5.70_{\pm0.13}}$ | 24.84 | $7.14_{\pm0.74}$ | $25.27_{\pm0.56}$ | $26.03_{\pm0.33}$ | $4.57_{\pm0.97}$ | 15.75 |
| Adaptive KL (Wu et al., 2025) | $25.18_{\pm1.86}$ | $33.86_{\pm2.34}$ | $33.74_{\pm1.77}$ | $4.85_{\pm0.44}$ | 24.41 | $5.47_{\pm0.63}$ | $24.76_{\pm1.56}$ | $24.07_{\pm1.12}$ | $4.23_{\pm0.41}$ | 14.63 |
| Skew FKL (Ko et al., 2024; 2025) | $25.05_{\pm1.40}$ | $31.39_{\pm2.84}$ | $32.34_{\pm2.99}$ | $4.73_{\pm0.86}$ | 23.37 | $7.01_{\pm0.82}$ | $25.08_{\pm0.88}$ | $\underline{26.14_{\pm0.70}}$ | $\underline{5.01_{\pm0.29}}$ | $\underline{15.81}$ |
| Skew RKL (Ko et al., 2024; 2025) | $\underline{25.24_{\pm0.79}}$ | $33.00_{\pm1.40}$ | $32.24_{\pm1.36}$ | $4.27_{\pm0.44}$ | 23.69 | $5.83_{\pm0.25}$ | $25.17_{\pm0.22}$ | $25.93_{\pm0.29}$ | $3.38_{\pm0.26}$ | 15.08 |
| $\alpha$-$\beta$ divergence (Wang et al., 2025b) | $25.07_{\pm1.36}$ | $\underline{34.78_{\pm2.04}}$ | $\underline{34.90_{\pm1.45}}$ | $\underline{5.27_{\pm0.44}}$ | $\underline{25.01}$ | $\underline{7.26_{\pm0.78}}$ | $\underline{25.49_{\pm1.00}}$ | $26.07_{\pm0.66}$ | $4.34_{\pm0.89}$ | 15.79 |
| **HybKD (Ours)** | $\mathbf{27.44_{\pm2.85}}$ | $\mathbf{35.64_{\pm3.22}}$ | $\mathbf{37.01_{\pm3.98}}$ | $5.00_{\pm1.25}$ | **26.27** | $\mathbf{7.78_{\pm1.19}}$ | $\mathbf{25.94_{\pm0.55}}$ | $26.07_{\pm0.85}$ | $\mathbf{5.24_{\pm0.79}}$ | **16.26** |

