# OpenReview forum: "The Bridge-Garden Dilemma in LLM Distillation: Why Mixing Hard and Soft Labels Works"
_ICML.cc/2026/Conference — ICML 2026 regular_

### Official Review · Reviewer_RfJB · 2026-02-18

**Soundness:** 3
**Presentation:** 3
**Significance:** 3
**Originality:** 2
**Overall Recommendation:** 4
**Confidence:** 3

**Summary:**

This paper explores why combining hard labels (teacher-sampled tokens) with soft labels (teacher distribution) in LLM distillation often outperforms using only soft labels. The authors suggests the improvement mainly comes from reducing exposure bias rather than better matching the teacher during training. They also introduce a “Bridge vs Garden” concept, where some steps require exact tokens and others allow more flexibility.

**Compliance With Llm Reviewing Policy:**

Affirmed.

**Final Justification:**

The rebuttal was satisfactory.

**Key Questions For Authors:**

See the weaknesses above

**Limitations:**

Yes

**Strengths And Weaknesses:**

Strengths

1. The main empirical finding is clear: mixing hard and soft labels improves downstream accuracy, eventhough it can sometimes reduce training imitation. This result is both interesting and not obvious.
2. The Bridge/Garden framework is easy to understand and offers a convincing explanation for why hard targets are useful at “high-risk” steps, while soft targets work better in other cases.
3. The paper tests multiple teacher-student pair and showed significant compute saving compared to on-policy KD.

Weaknesses

1. The main theoretical concept is the local risk sensitivity k(a|s) and the k-weighted exposure-bias bound (Theorem 4.1). However, the paper doesn’t clearly explain how k(a|s) is estimated or approximated in practice for adaptive methods, relying instead on proxies like teacher confidence or entropy, which don’t have a clear link to k beyond intuition.

2. In Definition 4.1, k(a|s) is defined using terms like (L_{d(s,a)}(pi_theta)) and (L_{d_T}(pi_theta)), but then k is used to guide training of pi_theta. This isn’t wrong, but it means k depends on the student model and even the training stage, making it more of a moving target than a fixed property of the prefix. I’d like the authors to clarify whether k is considered a teacher or task property, or it depend on the student, and what that means.

3. Definition 4.2 uses \tau to divide prefixes into Bridges and Gardens, but the paper doesn’t explain how to choose \tau, whether results depend on it, or if this split is consistent across datasets and domains.

5.  The “risk-guided” hybrid objective modifies the hard loss with a term involving \Delta theta and logits, but the notation is dense and it’s not obvious how costely this is, how it interacts with the claimed training-cost benefits, and whether α is tuned heavily. Also the expression uses logits conditioned on something like “[s, a^star]”, which is unusual and needs clearer explanation

6. They claim statistical significance in a figure, but the discussion of training costs and on-policy comparisons could use more details about matching hyper-parameters and compute budgets. For example, the 9.7* cost increase is substantial, but it’s not clear if this includes teacher sampling, storing logits, or other factors from the main text.

7. The paper argues the gain is exposure-bias reduction rather than optimization ease, based on forward-KL imitation error curves, but I still wonder whether label-mixing is acting like a regularizer (entropy control, calibration) or changing effective temperature in a way that helps generilization. I like ablations that isolate those effects more directly, beyond the decomposition narrative.

---

> ### Author Rebuttal · Authors · 2026-03-26
>
> We thank the reviewer for the thorough review.
>
> **Rebuttal experiments: https://anonymous.4open.science/r/re-CD85/RfJB/RfJB.md**
> > **$\color{blue}{\textbf{W1}}$: Connection between κ and practical proxies**
>
> **A:** Thanks for your insightful question! Direct κ estimation is intractable (path sampling over exponential sequence space). We use teacher confidence as proxy: κ measures how much a token error propagates to future positions, and teacher confidence naturally reflects this (see **BfWQ W1** for details).
>
> In rebuttal, we validate this directly across **3 synthetic domains** (Code, Math, Dialogue) where κ is computable. Our results show that teacher confidence tracks κ with **Pearson r = 0.70–0.91** (see **BfWQ W1**).
>
> On **9 real LLM pairs**, adaptive weighting using this proxy outperforms baselines (Tabs1-5, 8-11 in the main text, and **Tab.1 (2 new pairs: Qwen2.5-32B→3B, Coder-7B→1.5B)** at the link above).
>
> Within limited time, we further ran a **reverse-weighting ablation (more soft where teacher confidence is high, more hard where low), performance drops 1.99 points on Qwen2.5-7B→3B**, confirming teacher confidence as reliable κ proxy (**Tab.2** at the link above)
> > **$\color{blue}{\textbf{W2}}$: κ depends on the student, a moving target**
>
> **A:** Yes, κ depends on the student & evolves during training. As student improves, its generation behavior changes; how far token errors cascade also shifts.
>
> This is standard in ML: advantage function [1], Fisher matrix [2] and Adam [3] all depend on the current model state yet guide training effectively.
>
> Our setting is simpler: κ only determines how to weight hard vs. soft, while the goal (matching student to teacher) remains fixed.
>
> [1] Proximal Policy Optimization Algorithms, ICLR'16
>
> [2] Natural Gradient Works Efficiently in Learning, Neural Comp.'98
>
> [3] Adam, ICLR'15.
> > **$\color{blue}{\textbf{W3}}$: On the τ threshold**
>
> **A:** Thank you! τ in Def.4.2 is only for analysis, not a training hyperparameter. Our method avoids explicit τ: instead of a binary Bridge/Garden split, it uses a continuous κ-proxy weight per position.
>
> The Bridge-Garden partition is a theoretical lens (not a training procedure); Prop.4.1 uses it to decompose Thm.4.1 bound region-wise, while the algorithm applies a threshold-free weighting rule.
>
> No universal τ needed. Empirically, our experiments show lower EB for hard KD at Bridge, soft KD at Garden, hybrid KD best overall. Please see **BfWQ W1** for details.
> > **$\color{blue}{\textbf{W4}}$: Cost and notation of the risk-guided term**
>
> **A:** Thank you!. [s,a] denotes concatenation of prefix s and token a. Δ_θ(s,a) is computed from teacher and student logits already in soft KD, requiring one extra log-sum-exp per token. α=0.1 is fixed across experiments. We'll clarify notation in camera-ready. Cost comparison (Qwen2.5-7B→3B, 4×A100):
>
> |Method|Extra compute|s/step|
> |-|-|-|
> |Soft KD|none|14.06|
> |Hybrid KD (risk-guided)|1 log-sum-exp per token|15.24|
> |On-policy KD|autoregressive student sampling|147.83|
> > **$\color{blue}{\textbf{W5}}$: Training cost details**
>
> **A:** As in App.E, all methods share epochs, hyperparams, batch size, hardware (4×A100 80GB). Teacher logits computed on the fly (not stored). As in **W4**, hybrid adds negligible cost over soft KD. The 9.7× gap comes solely from on-policy autoregressive student sampling; no other factors (e.g., logit storage) involved.
> > **$\color{blue}{\textbf{W6}}$: Relationship to regularization, calibration and temperature.**
>
> **A:** Thank you for this insightful question! Our framework provides **a unified perspective where regularization, temperature scheduling, and hybrid mixing emerge as special cases** of navigating the Bridge-Garden trade-off.
>
> For example, [1] anneals temperature T high→low, while [2] learns T low→high. Both improve over fixed T, which appears contradictory at first. **Our Bridge-Garden view explains this:** lower T sharpens the teacher (suits Bridges), while higher T preserves diversity (suits Gardens).
>
> In rebuttal, we add 4 baselines: entropy-reg soft kd, the two temperature schedules [1,2], and random-label mixing using random tokens instead of teacher-sampled hard labels. Below: **Qwen2.5-7B→3B**; see **Tabs.4–6** (reg/temp baselines for Qwen2.5-Code, Llama3, DeepSeek-Code) at the link above.
>
> |Method|Avg|
> |-|-|
> |Soft KD|51.86|
> |Entropy reg.|54.48|
> |T: high→low|53.80|
> |T: low→high|54.20|
> |Random-label mixing|48.96|
> |Hybrid KD (Ours)|**55.16**|
>
> The first 3 improve over pure soft KD, but all remain below hybrid KD. This is because Entropy regularization & temperature modify the target distribution globally across all positions, while hybrid KD uses teacher-guided hard-soft mixing to adjust the supervision type per position.
>
> Random-label mixing performs the worst, showing that the gain does not come from mixing alone: hard labels must be teacher-supported & used in the right places.
>
> [1] Annealing KD, EACL'21
>
> [2] Curriculum Temperature for KD, AAAI'23

---

> > ### Author Rebuttal · Reviewer_RfJB · 2026-04-02
> >
> > I thank the authors for their detailed an effective rebuttal. You have addressed the core weaknesses identified in my initial review.
> >
> > * **Proxies for $\kappa$:** The newly provided Pearson correllations (r = 0.70–0.91) and the reverse-weighting ablation effectively bridge the gap between your theoretical local risk sensitivity and the practical proxies (teacher confidence/entropy). Please ensure these synthetic domain experiments are included in the camera-ready appendix.
> > * **Moving Target:** I accept your justification regarding $\kappa$ evolving during training; the analogy to advantage functions in reinforcement learning is sound.
> > * **The $\tau$ Threshold:** Clarifying that the $\tau$ thresthold in Definition 4.2 is strictly an analytical tool used to establish the Bridge-Garden partition, rather than a tuned training hyperparameter, resolves my concern regarding its selection.
> > * **Cost & Notation:** The explicit compute breakdown (14.06 steps/s for Soft KD vs. 15.24 for Hybrid vs. 147.83 for On-Policy) perfectly clarifies the 9.7x efficiency claim. I appreciate the clarification on the $[s, a]$ concatenation notation for the risk-guided objective.
> > * **Regularization/Temperature Isolations:** The addition of the entropy-regularization, temperature annealing, and random-label mixing baselines is excellent. This convincingly demonstrates that your method's success stems from position-specific target adjustments rather than a global regularization or calibration effect.
> >
> > **Final Verdict**
> > The rebuttal was thorough and cleared up my reservations regarding the theoretical-to-practical translation, compute overhead, and alternative explanations for the performance gains. I am raising my score to reflect these clarifications.
> >
> > I will update my score.

---

> > > ### Author Response · Authors · 2026-04-03
> > >
> > > We sincerely thank the reviewer for the careful re-evaluation and for raising the score. We are glad that the synthetic domain correlations, compute breakdown, and regularization isolation baselines have resolved the key concerns.
> > >
> > > We noticed the acknowledgement is marked as "(b) Partially resolved — follow-up questions." If there are any remaining questions you would like us to clarify further, we would be very happy to provide additional details.
> > >
> > > Thank you again for the constructive and thorough review throughout this process — it has meaningfully strengthened our work.

---

### Official Review · Reviewer_Cfye · 2026-03-11

**Soundness:** 3
**Presentation:** 3
**Significance:** 3
**Originality:** 3
**Overall Recommendation:** 4
**Confidence:** 3

**Summary:**

The paper presents a theoretical framework for explaining the gains of hybridization (mixing hard and soft labels) in knowledge distillation. It also proposes a family of hybrid supervision methods that adaptively balance hard and soft labels, which show empirical gains over baselines.

**Compliance With Llm Reviewing Policy:**

Affirmed.

**Final Justification:**

The paper proposes a novel and conceptually interesting framework, and I find both its theoretical and empirical contributions valuable. The idea is original and has the potential to influence how we think about hybrid training and knowledge distillation. The methodology appears sound, and the empirical results support the practical usefulness of the approach, though some aspects of the mechanism were initially under-validated.

The authors’ rebuttal fully addressed my main concerns. In particular, the added synthetic-domain analysis provides direct and convincing evidence for the proposed Bridge/Garden mechanism, including the expected exposure-bias decomposition. The clarification of training costs and the improved positioning relative to prior KD methods also strengthened the paper. While I still believe that the mechanism is most clearly validated in controlled (synthetic) settings, and that the real-world experiments more directly support the effectiveness of the hybrid approach than the full theoretical account, I now view this as a matter of scope and framing rather than a flaw.

Overall, I weigh the paper’s originality, theoretical contribution, and strengthened empirical support positively. The rebuttal significantly improved my assessment and resolved my primary concerns, leading to a favorable final evaluation.

**Key Questions For Authors:**

1. In 6.4, "While our method maintains training costs comparable to standard Hard/Soft KD, it is 9.7x more efficient than on-policy KD". Could the training cost of these methods be reported in more detail, as efficiency appears to be an important claim for method superiority?

2. As mentioned, the respective improvements in Bridge and Garden regions do not appear to be empirically validated in the current design. Could the authors provide direct empirical analyses of bridge-like vs. garden-like positions?

3. How realistic are the assumptions for Theorem 4.1 in practice?

4. How does the current model differ from other relevant hybrid-based models, if any?

**Limitations:**

Yes

**Strengths And Weaknesses:**

Strengths:

1. This work dives deeper than the observation of Hybrid KD's success itself, and provides a solid systematic investigation into the reasons behind its robustness. The Bridge-Garden formulation is clean, and the argument for the exposure bias is convincing and well-supported.

2. The comprehensive experiment suite suggests that gains in the hybrid method persist across various architectures, tasks, and distillation divergence.

3. Multiple practical algorithms for Bridge-Garden Hybrid Supervision, where all hybrid strategies are well-motivated and conducted.


Weaknesses:

1. The current evidence supports the usefulness of hybridization more directly than it validates the specific Bridge-Garden mechanism. In experiments, the hybrid approach is basically mixing soft and hard objectives, therefore what's directly proven is that hybridization is effective (not a novel discovery by itself). These findings do not readily align with the Bridge-Garden decomposition in the previous theoretical formulation -- recall how the theory defines Bridge and Garden regions via the local-risk sensitivity, and claims that hard supervision should be preferable in Bridges while soft supervision should be preferable in Gardens. The experiments do not identify either region directly.

2. Assumptions are strong for Theorem 4.1 (upper bound on exposure bias) to hold.

3. For the method proposed, more discussion on related works and baselines should be expected. Novelty claim should be more clearly framed.

---

> ### Author Rebuttal · Authors · 2026-03-26
>
> We thank Reviewer Cfye for the constructive suggestions. Our response follows:
>
> **Rebuttal experiment: https://anonymous.4open.science/r/re-CD85/Cfye/Cfye.md**
>
> > **$\color{blue}{\textbf{W1/Q2}}$: Empirical validation of Bridge-Garden mechanism**
>
> **A:** Thank you for this important concern. Direct validation on real LLMs is infeasible: ground-truth κ requires path sampling over an exponential sequence space. We therefore construct **3 synthetic domains** (Code, Math, Dialogue) where κ is computable by design.
>
> Each domain has ~120–180 token vocabulary, 512 training + 128 test sequences of 24–96 tokens each. Teacher: 3-layer 128-dim 4-head Transformer, trained 4 epochs (AdamW, lr 3e-4). Student: 2-layer 64-dim, distilled under Hard/Soft/Hybrid KD. Ground-truth κ is computed via path sampling. To obtain a clean role signal with minimal boundary noise, we label the top-30% κ tokens as Bridge and the bottom-30% as Garden.
>
> A Code sample: `base = item + 3  # guard safe path  if base > seed: return base`
>
> `if`, `return` and variable re-reads (`base`, `seed`) = **Bridges** (high κ); `guard`, `safe`, `path` = **Gardens** (comment words, low κ). **Fig.3** at the link above provides token-level κ heatmaps that intuitively show the risk level at each token position within a sequence.
>
> Results (lower EB = less error accumulation):
>
> |Domain|Bridge EB(Hard KD)|Bridge EB (Soft KD)|Garden EB (Hard KD)|Garden EB (Soft KD)|
> |-|-|-|-|-|
> |Code|0.003|0.009|0.533|**0.108**|
> |Math|0.019|0.067|0.116|**0.086**|
> |Dialogue|0.004|0.215|0.186|**0.056**|
>
> **The pattern matches the theory exactly**: hard KD suppresses EB at Bridges where the teacher is near-deterministic; soft KD helps at Gardens where diversity matters.
>
> We further note that teacher confidence reliably tracks κ (**Pearson r = 0.70–0.91**), showing that our hybrid KD method is simple yet effective: it allows the model to mix hard and soft supervision correctly at each position, achieving the lowest overall EB:
>
> |Domain|κ-conf. r|Hard KD|Soft KD|Hybrid KD|
> |-|-|-|-|-|
> |Code|**0.91**|0.095|0.085|**0.047**|
> |Math|**0.70**|0.068|0.081|**0.049**|
> |Dialogue|**0.80**|0.032|0.070|**0.010**|
>
> > **$\color{blue}{\textbf{W2/Q3}}$: Assumption justification for Thm 4.1**
>
> **A:** Thank you for this important concern. We justify all three conditions of Thm. 4.1.
>
> Conditions 1–2 (bounded loss and strictly positive probabilities) **hold automatically for any softmax model over finite vocabulary**: bounded model parameters yield bounded logits, which guarantee strictly positive probabilities (Condition 2) and finite per-token loss (Condition 1).
>
> Condition 3 (prefix concentrability) bounds the density ratio between prefix distributions, the **standard way to handle distribution shift in offline RL/IL** since Munos [1], also used as the central tool in recent LLM alignment theory [2]. In our setting it is **mild: teacher and student are usually pretrained on the same corpus with the same tokenizer**, so their prefix distributions overlap substantially; KD further narrows this gap as the student is trained toward the teacher.
>
> [1] Error Bounds for Approximate Policy Iteration, ICML'03.
>
> [2] Correcting the Mythos of KL-Regularization: Direct Alignment without Overoptimization via Chi-Squared Preference Optimization, ICLR'25.
>
> > **$\color{blue}{\textbf{W3/Q4}}$: Related work, novelty, and difference from other hybrid models**
>
> **A:** Thank you for this important concern. Our core contribution is **the Bridge-Garden framework (Thm 4.1), which establishes the first theoretical connection between hybrid hard-soft supervision and exposure bias in LLM distillation**. This theoretical insight explains why teacher-sampled hard labels help despite carrying less information: they reduce EB at Bridge positions.
>
> Based on this framework, we design adaptive mixing strategies that leverage teacher confidence as a κ proxy. The closest prior work — token-adaptive methods (ATKD, ACL'24; ToDi, EMNLP'25; AdaKD, AAAI'26) — **reweights or switches divergences within soft KD**. Our methods operate on a different axis: mixing hard and soft supervision types.
>
> To our knowledge, **no prior work explicitly mixes teacher-sampled hard labels with soft labels in LLM distillation**, because hard = teacher sample appears less informative than soft label, and there is no ground-truth correction signal to motivate mixing. Our Bridge-Garden framework provides exactly this missing motivation.
> > **$\color{blue}{\textbf{Q1}}$: Training cost**
>
> **A:**  Thank you for this insightful question! All methods in our experiments share epochs, hyperparameters, batch size, and hardware (4×A100 80GB). Teacher logits computed on the fly (not stored). As shown below, hybrid adds negligible per-step cost over soft KD.
>
> The 9.7× gap **comes solely from on-policy autoregressive student sampling; no other factors (e.g., logit storage) are involved**.
>
> |Method|s/step|
> |-|-|
> |Soft KD|14.06|
> |Hybrid KD|15.24|
> |On-policy KD|147.83|

---

> > ### Author Rebuttal · Reviewer_Cfye · 2026-04-03
> >
> > Thank you for the detailed rebuttal. My main empirical concern is addressed: the new synthetic-domain analysis provides direct evidence for the Bridge/Garden mechanism by computing $\kappa$, showing the expected Bridge/Garden exposure-bias decomposition. The additional training-cost details are also helpful and make the efficiency claim clearer. In addition, the discussion of the theorem assumptions and the positioning relative to prior KD methods is more explicit now.
> > It’s still worth noting that, although the new mechanism validation is compelling, it is still mainly shown in synthetic settings; so the empirical evidence from the main experiments continues to support the usefulness of hybridization more directly than it fully establishes the complete Bridge-Garden account in practice. Please be cautious with overclaiming during the final submission.
> > Overall, I consider my original concerns to be well-addressed in the rebuttal.

---

> > > ### Author Response · Authors · 2026-04-03
> > >
> > > We sincerely thank the reviewer for confirming that the original concerns are well-addressed.
> > >
> > > We take the caution on overclaiming seriously. In the camera-ready, we will present the Bridge-Garden mechanism as theoretically motivated and synthetically validated, with real-data gains attributed to the hybridization strategy it inspires rather than claiming full empirical establishment of the mechanism.
> > >
> > > We are grateful for the constructive feedback throughout this process. **As all concerns have been fully resolved, we would be grateful if the reviewer could also consider adjusting the score to reflect this.**

---

### Official Review · Reviewer_KyFs · 2026-03-14

**Soundness:** 3
**Presentation:** 3
**Significance:** 2
**Originality:** 3
**Overall Recommendation:** 4
**Confidence:** 3

**Summary:**

This work aims to address a general aspect of autoregressive knowledge distillation (KD): why mixing hard teacher tokens and soft teacher distributions can outperform pure soft distillation.
Overall, the research's central result concerns the claim that hybrid KD improves student generation mainly by reducing exposure bias, even when student–teacher matching on teacher prefixes becomes worse.
The empirical evidence is fairly broad. For Qwen2.5-7B, Figure 2 reports an average gain of 1.70 points over soft KD, together with worse training fit but lower exposure bias.
The paper also shows strong results for several hybrid variants across reasoning and math. In Qwen setting, the paper further claims a 9.7 times training-cost advantage over an on-policy KD baseline.

**Compliance With Llm Reviewing Policy:**

Affirmed.

**Final Justification:**

The rebuttal strengthened my view of the paper. The main weaknesses I raised were addressed in a satisfactory way, and I am updating my score upward accordingly.

**Key Questions For Authors:**

1. How do the authors reconcile the Bridge/Garden complementarity story with the fact that forward-KL soft KD and teacher-sampled hard KD share the same population objective?

2. Can the on-policy comparison be extended beyond the single Qwen2.5-7B 3B setting?

**Strengths And Weaknesses:**

## Strengths

The paper asks a good and practical question. The finding that adding hard labels can help even when teacher imitation on teacher prefixes gets worse is interesting, and Figure 2 makes this point clearly.

The empirical section is broad. The study covers multiple teacher–student pairs, reasoning, math, and coding benchmarks, several divergence choices, and a comparison to on-policy KD.

The paper is also well organized. The Bridge/Garden picture in the first page is memorable, and the tables and appendix make the experimental setup fairly easy to inspect.

## Weaknesses

1. The first sentence of the abstract, “the student is trained either on tokens sampled from the teacher (hard labels) or the teacher’s full next-token distribution (soft labels),” is not fully accurate. There are many on-policy distillation (OPD) methods, as well as reverse-KL-based distillation methods, that do not fit neatly into this framing. The claim should therefore be qualified more carefully, for example: “Under the forward-KL setting, the student is typically trained either on tokens sampled from the teacher (hard labels) or the teacher’s full next-token distribution (soft labels).” That said, adding this scope restriction would also narrow the claimed generality and impact of the paper.

2. In Appendix D, the formal result proves strict improvement from a one-step hybrid update under several assumptions. That is weaker than proving that the optimizer of the hybrid training objective must beat both pure endpoints. I think the main text should state this more carefully.

---

> ### Author Rebuttal · Authors · 2026-03-26
>
> We thank Reviewer KyFs for the thoughtful review.
>
> **Rebuttal experiments: https://anonymous.4open.science/r/re-CD85/KyFs/KyFs.md**
> > **$\color{blue}{\textbf{W1}}$: Hard/soft framing scope (OPD, Reverse-KL)**
>
> **A:** Thanks! The hard/soft distinction concerns **what supervision signal the student receives** (hard token vs. soft distribution), orthogonal to **how it matches the teacher** (divergence) or **where the prefix source** (on/off-policy). This framing follows Hinton et al. [1] and is standard in KD [2,3].
>
> Two examples: MiniLLM [4] uses reverse KL with on-policy prefixes, yet its signal is teacher's soft distribution. GKD [5] also uses on-policy prefixes with soft signal.
>
> **Fig.3 in the main text** confirms this empirically: **hybrid improves across Forward/Reverse KL, Skew KL, JSD, and α-β div**, confirming the benefit is driven by signal type. In rebuttal, we combine on-policy KD with our method and see added gains (**see Q2 below**).
>
> [1] Distilling the Knowledge in a Neural Network 15
>
> [2] Sequence-Level KD EMNLP16
>
> [3] ABKD ICML25.
>
> [4] MiniLLM ICLR24.
>
> [5] GKD: On-Policy Distillation of Language Models ICLR24.
> > **$\color{blue}{\textbf{W2}}$: One-step improvement scope**
>
> **A:** Thanks! We agre Thm.D.11 establishes **one-step improvement**. **We will revise the main text accordingly based on the new result below.** During rebuttal, we further analyze the **hybrid target itself** and derive a stronger target-level result.
>
> **Our core idea: construct an interpolated target distribution and use it to show that hybrid KD can achieve lower $F$ than both hard and soft KD.** Concretely, hybrid KD targets
>
> $$q_\lambda=\lambda\,\pi_T+(1-\lambda)\,\delta_{a^*}, \qquad \lambda\in(0,1),$$
>
> while soft KD targets $\pi_T$ and hard KD targets $\delta_{a^*}$.
>
> Thm.D.1 already shows that both pure endpoints carry irreducible $F$-gaps under Bridge/Garden: $F(\pi_{\text{soft}})\ge L_S>0$ since soft KD over-smooths at Bridges (teacher near-deterministic), while $F(\pi_{\text{hard}})\ge L_H>0$ since hard KD discards distributional info at Garden (multiple viable continuations).
>
> We then analyze the interpolated target directly. Since
>
> $$q_\lambda-\pi_T=(1-\lambda)(\delta_{a^*}-\pi_T),$$
>
> substituting into $F$ (Eq.7 in the main text, the $\kappa$-weighted EB bound) gives
>
> $$F(q_\lambda)=(1-\lambda)\underbrace{\mathbb{E}_s\!\Big[\sum_a \kappa\,|\delta_{a^*}(a)-\pi_T(a)|\Big]}_{K\le 2\kappa_{\max}}+C_2(1-\lambda)^2\underbrace{\mathbb{E}_s\!\big[\|\delta_{a^*}-\pi_T\|_1^2\big]}_{D\le 4},$$
>
> where $K\le \kappa_{\max}\,\mathbb{E}_s\!\big[\|\delta_{a^*}-\pi_T\|_1\big]\le 2\kappa_{\max}$ and $D\le 4$ because $\|\delta_{a^*}-\pi_T\|_1\le 2$ for distributions. Hence $F(q_\lambda)\to 0$ as $\lambda\to 1$. Since $L_S,L_H>0$, there exists $\lambda^*\in(0,1)$ such that
>
> $$F(q_{\lambda^*})<\min\{F(\pi_{\text{soft}}),\,F(\pi_{\text{hard}})\}.$$
>
> Since $q_\lambda$ is the target of hybrid KD, let $\eta_\lambda:=\|\pi_{\text{hyb}}-q_\lambda\|_1$ be its imitation error. By the Lipschitz bound on $F$ (App.D Eq.54),
>
> $$F(\pi_{\text{hyb}})\le F(q_\lambda)+(\kappa_{\max}+4C_2)\eta_\lambda.$$
>
> Thus, when imitation error small, the target-level advantage carries over to the trained model:
>
> $$F(\pi_{\text{hyb}})<\min\{F(\pi_{\text{soft}}),F(\pi_{\text{hard}})\}.$$
>
> We verify this across **9 LLM pairs and 3 synthetic domains** (see **BfWQ W1**, **Cfye W1&Q2**): hybrid beats both pure methods in all cases.
> > **$\color{blue}{\textbf{Q1}}$: Reconciling Bridge-Garden with shared population objective**
>
> **A:** Yes, soft and hard KD share the same population minimizer under forward KL. The Bridge-Garden trade-off is about the **optimization path, not the endpoint**.
>
> With finite data & limited capacity (e.g., LoRA), training never reaches the optimum, and the two signals impose different inductive biases:
>
> - Soft KD: full dist. → mode-covering → over-smoothing at Bridge
> - Hard KD: single token → concentration → under-diversity at Garden
>
> Hybrid balances both, reducing EB across the sequence.
>
> **Fig.3 in the main text** confirms this beyond forward KL: **hybrid improves across all five divergences**
>
> ABKD [3] reaches the same conclusion: even with the same optimum, different optimization paths produce different student distributions.
>
> In rebuttal, we **add 2 new pairs (avg score, Tabs.5-6 at the link**):
>
> ||Hard KD|Soft KD|Hybrid KD|
> |-|-|-|-|
> |Qwen2.5-32B→3B|51.0|52.6|**54.1**|
> |QwenCoder-7B→1.5B|54.2|53.1|**55.0**|
>
> > **$\color{blue}{\textbf{Q2}}$: On-policy comparison**
>
> **A:** Thanks! Our method is **orthogonal to on-policy KD**, and can be combined. In rebuttal, we ran on-policy + hybrid on **4 pairs across 2 domains** (code, reasoning). (**Tabs.1–4 at the link above: Qwen-Code, Llama3, DeepSeek-Code, Qwen2.5**) Below we list Llama8B→1B:
>
> |Method|Avg|
> |-|-|
> |Off-policy soft KD|23.25|
> |Off-policy hybrid KD|26.27|
> |On-policy soft KD|27.42|
> |**On-policy hybrid KD**|**27.79**|
>
> On-policy hybrid consistently outperforms others across all pairs.

---

> > ### Author Rebuttal · Reviewer_KyFs · 2026-04-03
> >
> > Firstly, I would like to thank the authors for the detailed rebuttal and the additional experiments.  I am satisfied with the clarification on the hard/soft framing. The authors made it clear that their distinction is about the supervision signal received by the student.
> >
> > I appreciate the authors' clarification regarding the theory. I agree with their updated position that the original formal result is a one-step improvement result and should be described as such in the main paper. Overall, the rebuttal strengthened my view of the paper. The main weaknesses I raised were addressed in a satisfactory way, and I am updating my score upward accordingly.

---

> > > ### Author Response · Authors · 2026-04-03
> > >
> > > Thank you for your careful reading of our rebuttal and for your constructive feedback!
> > >
> > > We are pleased that the clarifications addressed your concerns. In the final manuscript, we will revise the relevant descriptions accordingly to improve clarity and precision.

---

### Official Review · Reviewer_BfWQ · 2026-03-14

**Soundness:** 3
**Presentation:** 3
**Significance:** 3
**Originality:** 3
**Overall Recommendation:** 5
**Confidence:** 3

**Summary:**

In this paper, the authors analyze why mixing hard labels (the teacher's predicted tokens ) with soft labels (the teacher's full predictive distribution ) yields better performance than using soft labels alone. Their theoretical analysis reveals that the key factor is the reduction of exposure bias. To explain this, they introduce the Bridge-Garden Decomposition theory, which categorizes generation steps into two types: Bridges and Gardens. Bridges refer to tokens that play a crucial role during generation and must be exact (e.g. a mathematical operator in an equation). In contrast, Gardens are regions where token choices can be flexible; replacing a token with a semantically similar alternative (e.g., "excellent" -> "great") does not significantly impact the overall sequence. Based on this framework, the authors propose a family of HybKD strategies: Confidence-based weighting, Entropy-based weighting, Curriculum scheduling, and Risk-guided hybrid. These strategies are grounded in the principle that when the teacher is certain about a prediction, the token is more likely to be a Bridge, warranting an increased weight for the hard label. Experiments across various instruction-following benchmarks and model pairs demonstrate that HybKD not only achieves superior performance but is also significantly more computationally efficient than on-policy KD.

**Compliance With Llm Reviewing Policy:**

Affirmed.

**Key Questions For Authors:**

See weaknesses.

**Limitations:**

yes

**Strengths And Weaknesses:**

Strengths:
1. Through theoretical analysis, the authors reveal why mixing hard labels with soft labels yields better performance. This broadens our understanding of knowledge distillation (KD) beyond simple imitation.
2. The proposed KD methods exhibit superior performance, as evidenced by extensive experimental results across various benchmarks and teacher-student model pairs.
3. HybKD achieves impressive results while significantly reducing computational costs compared with on-policy distillation.

Weaknesses:
1. Gap between theoretical $\kappa$ and empirical proxies: The Bridge-Garden theory relies on the risk sensitivity $\kappa$ , yet the proposed algorithms use local teacher confidence or entropy as proxies. The paper lacks empirical validation (e.g., via path sampling ) to prove that these instantaneous indicators actually correlate with the long-term sequence risk defined in the theory.
2. Limited evaluation in open-ended domains: The benchmarks primarily focus on reasoning, math, and coding tasks. It remains unclear if introducing hard labels might over-compress diversity in "Garden-dominant" scenarios, such as creative writing or open-ended dialogue.

---

> ### Author Rebuttal · Authors · 2026-03-26
>
> We thank Reviewer BfWQ for the constructive suggestions. Our response follows:
>
> **Rebuttal experiments: https://anonymous.4open.science/r/re-CD85/BfWQ/BfWQ.md**
> > **$\color{blue}{\textbf{W1}}$: Gap between long-term risk sensitivity and local empirical proxies** The theory is defined by sequence-level risk sensitivity $\kappa$, while the algorithm uses local teacher confidence or entropy as proxies. The concern is whether these local signals truly track the long-term risk in the theory.
>
> **A:** Thank you for this important question. During rebuttal, we directly test whether teacher confidence tracks the sequence-level risk $\kappa$. The answer is yes.
>
> This proxy is also well motivated from prior evidence that LLM states encode information beyond the immediate next token. Hidden states can represent belief states over **future continuations** [1], frozen states can decode **multiple future tokens** [2,3], and **next-token training can induce future-oriented internal features** [4,5]. This does not by itself prove alignment with $\kappa$, but it explains why local teacher confidence is a plausible proxy for long-term sequence risk.
>
> During rebuttal, we directly validate this with **3 synthesized domains** (Code, Math, Dialogue): ~120–180 token vocabulary, 512 training + 128 test sequences per domain. We train a 3-layer Transformer teacher and distill into a 2-layer student under Hard/Soft/Hybrid KD; ground-truth κ is computed via path sampling.
>
> To obtain a clean role signal with minimal boundary noise, we label the top-30% κ tokens as Bridge and the bottom-30% as Garden (note: Bridge/Garden post-hoc for analysis only; training uses continuous κ-proxy weights with no hard threshold). A Code sample:
>
> `base = item + 3  # guard safe path  if base > seed: return base`
>
> `if`, `return` and variable re-reads (`base`, `seed` in condition/return) = **Bridges** (high κ); `guard`, `safe`, `path` = **Gardens** (comment words, low κ); first-time assignment `base =` has low κ. **Fig.3 at the link above** provides token-level κ heatmaps that intuitively show the risk level at each token position within a sequence.
>
> The following results show that teacher confidence tracks κ with **Pearson r = 0.70–0.91** across all 3 domains, confirming that teacher confidence is a reliable proxy for κ. Hybrid leverages this to achieve the lowest overall EB (lower = less error accumulation):
>
> |Domain|κ-conf. r|Hard KD EB|Soft KD EB|Hybrid KD EB|
> |-|-|-|-|-|
> |Code|**0.91**|0.095|0.085|**0.047**|
> |Math|**0.70**|0.068|0.081|**0.049**|
> |Dialogue|**0.80**|0.032|0.070|**0.010**|
>
> The EB decomposition further confirms the theory (**hard KD achieves lower Bridge EB in all 3 domains; soft KD achieves lower Garden EB in all 3 domains**):
>
> |Domain|Bridge EB (Hard KD)|Bridge EB (Soft KD)|Garden EB (Hard KD)|Garden EB (Soft KD)|
> |-|-|-|-|-|
> |Code|0.003|0.009|0.533|0.108|
> |Math|0.019|0.067|0.116|0.086|
> |Dialogue|0.004|0.215|0.186|0.056|
>
> This is consistent with our real-model results (**9 pairs, 11 benchmarks**; **see W2** for extended pairs).
>
> During rebuttal, we further run a **reverse-weighting ablation** (more soft where confidence is high, more hard where low): performance drops by **1.99 points on Qwen2.5-7B→3B Avg (53.8→51.8)**, confirming the proxy is used in the correct direction.
>
> |Method|BBH|MMLU|ARC-C|ThmQA|Avg|
> |-|-|-|-|-|-|
> |Hard KD|41.5|65.8|78.8|23.8|52.5|
> |Soft KD|41.7|64.5|78.3|23.0|51.9|
> |Hybrid: confidence-based|44.1|67.5|80.8|22.8|**53.8**|
> |Hybrid: reverse-confidence|41.5|65.3|78.1|22.3|51.8|
> |Hybrid: entropy|46.8|67.1|79.8|23.7|54.3|
> |Hybrid: reverse-entropy|41.2|65.2|77.3|20.5|51.1|
>
> [1] Transformers Represent Belief State Geometry in their Residual Stream NeurIPS'24
>
> [2] Medusa: Simple LLM Inference Acceleration Framework ICML'24
>
> [3] Better & Faster LLMs via Multi-token Prediction ICML'24
>
> [4] From r to Q*: Your Language Model is Secretly a Q-Function COLM'24
>
> [5] Emergent World Representations ICLR'23
>
> [6] Do Language Models Plan Ahead for Future Tokens COLM'24
>
> [7] Seemingly Useless Features in Next-Token Predictors ICLR'26
>
> [8] DeepSeek-V3 Technical Report arXiv'24
> > **$\color{blue}{\textbf{W2}}$: More open-ended evaluation**
>
> **A:** Thank you for this valuable suggestion! We add **AlpacaEval** (win rate against text-davinci-003, judged by GPT-5.2) and **2 new pairs** (Qwen2.5-32B→3B, Qwen Code-7B→1.5B). (Full benchmark breakdowns in **Tabs.2–3** at the link above)
>
> |Setting|Benchmarks|Hard KD|Soft KD|Hybrid KD|
> |-|-|-|-|-|
> |AlpacaEval (Qwen 7B→3B)|Win rate|57.5|61.3|**64.4**|
> |Qwen2.5-32B→3B|BBH/MMLU/ARC-C/ThmQA avg|51.0|52.6|**54.1**|
> |Qwen Code-7B→1.5B|HumanEval(+)/MBPP(+) avg|54.2|53.1|**55.0**|
>
> To further validate the generality, we add **4 on-policy + hybrid pairs across 2 domains (code, reasoning, tabs.4-10 at the link above)**: hybrid consistently adds gains on top of baselines. We also add **4 regularization/temperature baselines × 4 pairs**: all improve over pure soft KD, as our theory predicts.

---

> > ### Author Rebuttal · Reviewer_BfWQ · 2026-04-03
> >
> > Thank you for the detailed response! I recommend including these details in future versions. I will keep my initial positive assessment unchanged.

---

> > > ### Author Response · Authors · 2026-04-03
> > >
> > > We thank the reviewer for the positive assessment and for confirming that the concerns are fully resolved. We will incorporate these additional details into the camera-ready version.

---

### Decision · Program_Chairs · 2026-04-30

**Decision:**

Accept (regular)

**Comment:**

The paper shows that a mixed strategy of using soft and hard labels for LLM distillation is more beneficial than either alone. Despite soft labels having more information to better match the teacher, the authors argue that the mixture helps with reducing the mismatch between training and inference distributions. In particular they argue that hard labels are helpful for tokens needing "exact" answer, whereas the soft labels are useful for more "flexible" tokens. They show the benefits on several open source models against other baselines.

All reviewers agreed that the core empirical finding is clear, interesting, and not obvious, and the bridge/garden framing as an intuitive and reasonable explanation. Moreover, they appreciated the rigorous and broad evaluation in the paper, especially post rebuttal. There were several concerns raised: (1) mismatch between the proposed $\kappa$ and the empirical proxy used, (2) lack of validation of the proposed bridge/garden hypothesis over other reasons why hybrid training would help, (3) strong assumptions for the theoretical results, and (4) missing experimental details and over-claiming in certain parts. The authors put in tremendous amounts of work in the response period and satisfactorily addressed the major concerns (1) and (2) in synthetic settings they constructed. Given that all the reviewers were satisfied by these responses, and overall appreciated the contribution, I recommend acceptance.

I encourage the authors to include all the additional experiments and suggested writing improvements in the camera-ready and soften the claims about their hypothesis being validated only in synthetic settings.